# Improved Bounds for Swap Multicalibration and Swap Omniprediction

**Haipeng Luo**[*]
USC
haipengl@usc.edu

**Spandan Senapati**[*]
USC
ssenapat@usc.edu

**Vatsal Sharan**[*]
USC
vsharan@usc.edu

## Abstract

In this paper, we consider the related problems of *multicalibration* — a multigroup fairness notion and *omniprediction* — a simultaneous loss minimization paradigm, both in the distributional and online settings. The recent work of Garg et al. (2024) raised the open problem of whether it is possible to efficiently achieve $\tilde{\mathcal{O}}(\sqrt{T})$ $\ell_2$-multicalibration error against bounded linear functions. In this paper, we answer this question in a strongly affirmative sense. We propose an efficient algorithm that achieves $\tilde{\mathcal{O}}(T^{\frac{1}{3}})$ $\ell_2$-swap multicalibration error (both in high probability and expectation). On propagating this bound onward, we obtain significantly improved rates for $\ell_1$-swap multicalibration and swap omniprediction for a loss class of convex Lipschitz functions. In particular, we show that our algorithm achieves $\tilde{\mathcal{O}}(T^{\frac{2}{3}})$ $\ell_1$-swap multicalibration and swap omniprediction errors, thereby improving upon the previous best-known bound of $\tilde{\mathcal{O}}(T^{\frac{7}{8}})$. As a consequence of our improved online results, we further obtain several improved sample complexity rates in the distributional setting. In particular, we establish a $\tilde{\mathcal{O}}(\varepsilon^{-3})$ sample complexity of efficiently learning an $\varepsilon$-swap omnipredictor for the class of convex and Lipschitz functions, $\tilde{\mathcal{O}}(\varepsilon^{-2.5})$ sample complexity of efficiently learning an $\varepsilon$-swap agnostic learner for the squared loss, and $\tilde{\mathcal{O}}(\varepsilon^{-5}), \tilde{\mathcal{O}}(\varepsilon^{-2.5})$ sample complexities of learning $\ell_1, \ell_2$-swap multicalibrated predictors against linear functions, all of which significantly improve on the previous best-known bounds.

## 1 Introduction

Recent years have witnessed surprising connections between *multicalibration* — a multigroup fairness perspective (Hébert-Johnson et al., 2018) and *omniprediction* — a simultaneous loss minimization paradigm, first introduced by Gopalan et al. (2022a). In this paper, we consider multicalibration and omniprediction in both the distributional (offline) and online settings. We begin by introducing these two notions, starting with some notation. Let the instance space be $\mathcal{X} \subset \mathbb{R}^d$, label set be $\mathcal{Y} = \{0, 1\}$, $\mathcal{D}$ be an unknown distribution over $\mathcal{X} \times \mathcal{Y}$, $\ell : [0, 1] \times \mathcal{Y} \to \mathbb{R}$ be a loss function, $\mathcal{L}$ be a class of loss functions, $\mathcal{F}$ be a collection of hypotheses over $\mathcal{X}$, and $p : \mathcal{X} \to [0, 1]$ be a predictor. Multicalibration (for Boolean functions, e.g., Boolean circuits, decision trees $\mathcal{F} \subset \{0, 1\}^{\mathcal{X}}$) was introduced by Hébert-Johnson et al. (2018) as a mechanism to incentivize fair predictions. For Boolean functions, multicalibration can be interpreted as calibration (the property that the predictions of $p$ are correct conditional on themselves, i.e., $v = \mathbb{E}[y|p(x) = v]$ for all $v \in \mathsf{Range}(p)$), which is additionally conditioned on set membership. In its most general form, for an arbitrary bounded hypothesis class $\mathcal{F} \subset \mathbb{R}^{\mathcal{X}}$, multicalibration translates to the understanding that the hypotheses in $\mathcal{F}$ do not have any correlation with the residual error $y - p(x)$ when conditioned on the level sets of $p$. On the other hand, omnipredictors are sufficient statistics that simultaneously encode loss-minimizing predictions for a

---

[*]Author ordering is alphabetical.

39th Conference on Neural Information Processing Systems (NeurIPS 2025).

broad class of loss functions $\mathcal{L}$. Notably, omniprediction generalizes loss minimization for a fixed loss function (*agnostic learning* (Haussler, 1992)) to simultaneous loss minimization. Since different losses expect different "types" of optimal predictions (e.g., for the squared loss $\ell(p, y) = (p - y)^2$, the optimal predictor $p$ that minimizes the expected risk $\mathbb{E}[\ell(p(x), y)]$ is the Bayes optimal predictor $\mathbb{E}[y|x]$, where as for the $\ell_1$-loss $\ell(p, y) = |p - y|$ the optimal predictor is $\mathbb{I}[\mathbb{E}[y|x] \geq 0.5]$), such a simultaneous guarantee is made possible by a "post-processing" or "type-checking" of the predictions (a univariate data free minimization problem) via a loss specific function $k_\ell$, i.e., for each $\ell \in \mathcal{L}$, $k_\ell \circ p$ incurs expected loss that is comparable to the best hypothesis in $\mathcal{F}$.

Even though multicalibration is not stated in the context of loss minimization, the first construction of an efficient omnipredictor for the class of convex and Lipschitz functions $\mathcal{L}^{\mathsf{cvx}}$ in Gopalan et al. (2022a) was achieved via multicalibration, thereby representing a surprising connection between the above notions. However, multicalibration is not necessary for omniprediction (Gopalan et al., 2022a), thereby raising an immediate question related to the characterization of omniprediction in terms of a sufficient and necessary condition. Motivated by the role of *swap regret* in online learning and to explore the interplay between multicalibration and omniprediction, Gopalan et al. (2023b) introduced the concepts of *swap multicalibration*, *swap omniprediction*, and an accompanying notion *swap agnostic learning*, and also established a computational equivalence between the above notions. Informally, each of the above notions is a stronger version of its non swap variant, and requires a particular swap-like guarantee (specific to the considered notion) to hold at the scale of the level sets of the predictor, e.g., for swap agnostic learning, the predictor $p$ is required to have a loss that is comparable to the best hypothesis in $\mathcal{F}$ not just overall but also when conditioned on the level sets of $p$.

Despite the qualitative progress in understanding the interplay between swap omniprediction and swap multicalibration in both the distributional (Gopalan et al., 2023b) and online settings (Garg et al., 2024), a quantitative statistical treatment for the above measures has a huge scope of improvement even for the quintessential setting when the hypothesis class comprises of bounded linear functions $\mathcal{F}_1^{\mathsf{lin}}$. In particular, the existing bounds for swap omniprediction and $\ell_2$-swap multicalibration in the online setting as derived by Garg et al. (2024) are much worse than the corresponding bounds for online omniprediction (Okoroafor et al., 2025) and $\ell_2$-calibration (Luo et al., 2025; Fishelson et al., 2025; Foster and Hart, 2023; Foster and Vohra, 1998) respectively. Even more, the sample complexity of learning an efficient swap omnipredictor for the class of convex Lipschitz functions $\mathcal{L}^{\mathsf{cvx}}$ with error at most $\varepsilon$ is $\approx \varepsilon^{-10}$ (Gopalan et al., 2023b), which is prohibitively large. Along the lines of the above concern, Garg et al. (2024) devised an efficient algorithm with $\tilde{\mathcal{O}}(T^{\frac{3}{4}})$ $\ell_2$-swap multicalibration error after $T$ rounds of interaction between a forecaster and an adversary, and raised the problem of whether it is possible to efficiently achieve $\tilde{\mathcal{O}}(\sqrt{T})$ $\ell_2$-multicalibration error against $\mathcal{F}_1^{\mathsf{lin}}$. In this paper, we answer this question in a strongly affirmative sense by proposing an efficient algorithm that achieves $\tilde{\mathcal{O}}(T^{\frac{1}{3}})$ $\ell_2$-swap multicalibration error against $\mathcal{F}_1^{\mathsf{lin}}$. On propagating the above bound onward, we obtain a significantly improved rate for swap omniprediction for $\mathcal{L}^{\mathsf{cvx}}$. Subsequently, using our improved rates in the online setting, we construct efficient randomized predictors for swap omniprediction and swap agnostic learning in the distributional setting and derive explicit sample complexity rates, which significantly improve upon the previous best-known bounds.

**Contributions and Overview of Results.** Throughout the paper, we consider predictors $p$ such that $\mathsf{Range}(p) \subseteq \mathcal{Z}$, where $\mathcal{Z} = \{0, \frac{1}{N}, \ldots, \frac{N-1}{N}, 1\}$ is a finite discretization of $[0, 1]$ and $N$ is a parameter to be specified later. Similarly, in the online setting, we consider forecasters that make predictions that lie in $\mathcal{Z}$. Our contributions are as follows.

- In Section 2, we propose an efficient algorithm that achieves $\tilde{\mathcal{O}}(T^{\frac{1}{3}} d^{\frac{2}{3}})$ $\ell_2$-swap multicalibration error (both in high probability and expectation) against the class of $d$-dimensional linear functions $\mathcal{F}_1^{\mathsf{lin}}$. In contrast to our result, Garg et al. (2024) achieved a $\tilde{\mathcal{O}}(T^{\frac{3}{4}} d)$ bound by reducing $\ell_2$-swap multicalibration to the problem of minimizing *contextual swap regret* — an extension of swap agnostic learning to the online setting. Towards achieving this improved rate, we proceed in the following manner:

  1. We introduce the notions of *pseudo swap multicalibration* and *pseudo contextual swap regret*, where the swap multicalibration error (swap regret respectively) is measured via the the conditional distributions $\mathcal{P}_1, \ldots, \mathcal{P}_T$ rather than the true realizations $p_1, \ldots, p_T$. Subsequently,

in Section 2.2, in a similar spirit to Garg et al. (2024), we establish a reduction from pseudo swap multicalibration to pseudo contextual swap regret.

2. By not taking into account the randomness in the predictions, the pseudo variants are often easier to optimize. Indeed, in Section 2.3, we propose a deterministic algorithm that achieves $\tilde{\mathcal{O}}(T^{\frac{1}{3}}d^{\frac{2}{3}})$ pseudo contextual swap regret. In contrast, for contextual swap regret, Garg et al. (2024) established a $\tilde{\mathcal{O}}(T^{\frac{3}{4}}d)$ bound by using the reduction from swap regret to external regret as proposed by Ito (2020). To achieve the desired bound for pseudo contextual swap regret, we use the Blum-Mansour (BM) reduction (Blum and Mansour, 2007) instead. On a high level, the key to the improvement from $T^{\frac{3}{4}}$ (contextual swap regret) to $T^{\frac{1}{3}}$ (pseudo contextual swap regret) is the following: Garg et al. (2024) derived a $\tilde{\mathcal{O}}(T/N + N\sqrt{T})$ bound on the contextual swap regret, where the $\tilde{\mathcal{O}}(1/N)$ term is accounted due to a rounding operation (additively accounted for $T$ rounds) and the $\tilde{\mathcal{O}}(N\sqrt{T})$ term is due to a concentration argument, which is inevitable since the reduction by Ito (2020) is randomized. However, by using the BM reduction and an improved rounding procedure due to Fishelson et al. (2025), we propose a deterministic algorithm that achieves $\tilde{\mathcal{O}}(T/N^2 + N)$ bound on the pseudo contextual swap regret, where the $\tilde{\mathcal{O}}(1/N^2)$ term is due to the rounding operation and the $\mathcal{O}(N)$ term is because each of the $N$ external regret algorithms in the BM reduction can be instantiated to guarantee $\tilde{\mathcal{O}}(1)$ external regret against bounded linear functions. The desired result follows by choosing $N = \tilde{\Theta}(T^{\frac{1}{3}})$.

3. Using the reduction from pseudo swap multicalibration to pseudo contextual swap regret, we obtain a $\tilde{\mathcal{O}}(T^{\frac{1}{3}}d^{\frac{2}{3}})$ bound for the former. Noticeably, since the swap multicalibration error is possibly random and our algorithm for minimizing pseudo swap multicalibration is deterministic, we derive a concentration bound in going from pseudo swap multicalibration to swap multicalibration. By performing a martingale analysis using Freedman's inequality (Lemma 6 in Appendix B), we show that this only accounts for a $\tilde{\mathcal{O}}(N)$ deviation term, which does not change the final rate.

- In Section C in the appendix, we explore several consequences of our improved rate for $\ell_2$-swap multicalibration. Particularly, in Section C.1, we show that our algorithm from Section 2.3 achieves $\tilde{\mathcal{O}}(T^{\frac{2}{3}}d^{\frac{1}{3}})$ swap omniprediction error for $\mathcal{L}^{\text{cvx}}$ against $\mathcal{F}_{\text{res}}^{\text{aff}}$, where $\mathcal{F}_{\text{res}}^{\text{aff}}$ is a class of appropriately scaled and shifted linear functions. This significantly improves upon the $\tilde{\mathcal{O}}(T^{\frac{7}{8}}d^{\frac{1}{2}})$ rate of Garg et al. (2024). In Section C.2, we show that the same algorithm (with a different choice of the discretization parameter $N$) achieves $\tilde{\mathcal{O}}(T^{\frac{3}{5}}d^{\frac{2}{5}})$ contextual swap regret, which improves upon the $\tilde{\mathcal{O}}(T^{\frac{3}{4}}d)$ bound of Garg et al. (2024). We summarize a comparison of our results to the ones derived by Garg et al. (2024) in Table 1.

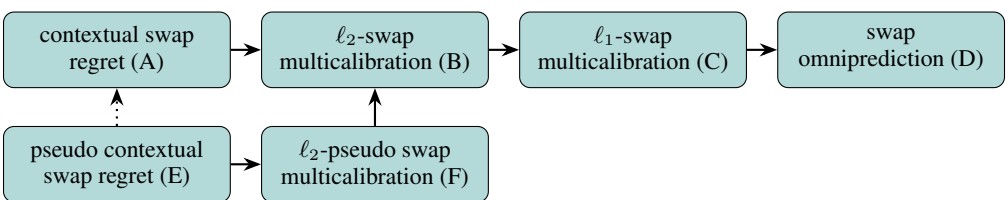

Figure 1: Path A → B → C → D represents the sequence of reductions followed by Garg et al. (2024), whereas path E → F → B → C → D represents our road-map. To derive an improved guarantee for B, we establish: (i) a $\tilde{\mathcal{O}}(T^{\frac{1}{3}})$ bound for E; (ii) a reduction from E to F; and (c) a concentration bound from F to B. The improved guarantees for C and D follow as a consequence of the improvement in B. The improved guarantee for A follows due to E and a concentration bound from E to A.

We remark that although relevant ideas for the above steps have appeared in the literature, our paper successfully unifies them in a novel framework to achieve state-of-the-art bounds for swap multicalibration and swap omniprediction. Refer to Figure 1 for a summary of our framework. For a detailed comparison with prior work, refer to the section on related work.

As a consequence of our improvements in the online setting, we establish several improved sample complexity rates in the distributional setting. Towards achieving so, we perform an online-to-batch conversion (Cesa-Bianchi et al., 2004) using our online algorithm in Section 2.3 to obtain a randomized predictor $p$ that mixes uniformly over the $T$ predictors output by the online algorithm.

Table 1: Comparison of our rates with the previous best-known bounds. For simplicity, we only tabulate the leading dependence on $\varepsilon, T$. The first 4 rows correspond to regret/error bounds (as a function of $T$) in the online setting, whereas the last 4 rows correspond to sample complexity bounds (as a function of $\varepsilon$) in the distributional setting.

| ONLINE (REGRET/ERROR) AND DISTRIBUTIONAL (SAMPLE COMPLEXITY) BOUNDS | | |
| --- | --- | --- |
| Quantity | Previous bound | Our bound |
| Contextual swap regret | $\tilde{\mathcal{O}}(T^{\frac{3}{4}})$ (Garg et al., 2024) | $\tilde{\mathcal{O}}(T^{\frac{3}{5}})$ (Theorem 3) |
| $\ell_2$-swap multicalibration | $\tilde{\mathcal{O}}(T^{\frac{3}{4}})$ (Garg et al., 2024) | $\tilde{\mathcal{O}}(T^{\frac{1}{3}})$ (Theorem 1) |
| $\ell_1$-swap multicalibration | $\tilde{\mathcal{O}}(T^{\frac{7}{8}})$ (Garg et al., 2024) | $\tilde{\mathcal{O}}(T^{\frac{2}{3}})$ (Corollary 1) |
| Swap omniprediction | $\tilde{\mathcal{O}}(T^{\frac{7}{8}})$ (Garg et al., 2024) | $\tilde{\mathcal{O}}(T^{\frac{2}{3}})$ (Theorem 2) |
| Swap agnostic error | $\tilde{\mathcal{O}}(\varepsilon^{-5})$ (Globus-Harris et al., 2023) | $\tilde{\mathcal{O}}(\varepsilon^{-2.5})$ (Theorem 5) |
| $\ell_2$-swap multicalibration | $\tilde{\mathcal{O}}(\varepsilon^{-5})$ (Globus-Harris et al., 2023) | $\tilde{\mathcal{O}}(\varepsilon^{-2.5})$ (Theorem 6) |
| $\ell_1$-swap multicalibration | $\tilde{\mathcal{O}}(\varepsilon^{-10})$ (Hébert-Johnson et al., 2018; Globus-Harris et al., 2023) | $\tilde{\mathcal{O}}(\varepsilon^{-5})$ (Theorem 6) |
| Swap omniprediction | $\tilde{\mathcal{O}}(\varepsilon^{-10})$ (Hébert-Johnson et al., 2018; Globus-Harris et al., 2023) | $\tilde{\mathcal{O}}(\varepsilon^{-3})$ (Theorem 4) |

- In Section D.1 in the appendix, we show that $\gtrsim \varepsilon^{-3}$ samples are sufficient for $p$ to be a swap omnipredictor for $\mathcal{L}^{\mathsf{cvx}}$ against $\mathcal{F}^{\mathsf{aff}}_{\mathsf{res}}$ with error at most $\varepsilon$. To prove this, we obtain a tight concentration bound that relates the swap omniprediction errors in the distributional and online settings. Our arguments in deriving the concentration bound are motivated by a recent work by Okoroafor et al. (2025), who proposed a similar online-to-batch conversion for omniprediction. However, compared to their result, an online-to-batch conversion for swap omniprediction poses several other technical nuances. In particular, unlike Okoroafor et al. (2025), we cannot merely use Azuma-Hoeffding's inequality or related concentration inequalities that guarantee concentration to a $\sqrt{n}$ factor ($n$ is the number of random variables). By performing a careful martingale analysis using Freedman's inequality on a geometric partition of the interval $[0, 1]$, we finally establish the desired concentration bound.

- In Section D.2, we specialize to the squared loss and show that $\gtrsim \varepsilon^{-2.5}$ samples are sufficient for $p$ to achieve swap agnostic error $\varepsilon$. Notably, since the squared loss is convex and Lipschitz, the result of Section D.1 already gives a $\tilde{\mathcal{O}}(\varepsilon^{-3})$ sample complexity. However, specifically for the squared loss, we derive a concentration bound that relates the contextual swap regret with the swap agnostic error. As we show, this bound is tighter than the corresponding deviation for swap omniprediction. Combining this with an improved $\tilde{\mathcal{O}}(T^{\frac{3}{5}})$ bound for contextual swap regret compared to swap omniprediction ($\tilde{\mathcal{O}}(T^{\frac{2}{3}})$), we obtain the improved sample complexity. Finally, in Section D.3, by using a characterization of $\ell_2$-swap multicalibration in terms of swap agnostic learning (Globus-Harris et al., 2023; Gopalan et al., 2023b), we establish a $\tilde{\mathcal{O}}(\varepsilon^{-2.5})$ sample complexity for $\ell_2$-swap multicalibration, and thus $\tilde{\mathcal{O}}(\varepsilon^{-5})$ for $\ell_1$-swap multicalibration against $\mathcal{F}^{\mathsf{lin}}_1$. We summarize our results and the previous best-known bounds in Table 1. For discussion regarding the previously best-known bounds, refer to the section on additional related work (Appendix A).

## 1.1 Preliminaries

For simplicity, we give formal definitions of several notions considered in the paper in the online setting, and defer definitions for the distributional setting to Appendix D.

**Online (Swap) Multicalibration.** Following Garg et al. (2024), we model online (swap) multicalibration as a sequential decision making problem over binary outcomes that lasts for $T$ time steps. At each time $t \in [T]$: (a) the adversary presents a context $x_t \in \mathcal{X}$; (b) the forecaster randomly

predicts $p_t \sim \mathcal{P}_t$, where $\mathcal{P}_t \in \Delta_{\mathcal{Z}}$ represents the conditional distribution of $p_t$; and (c) the adversary reveals the true label $y_t \in \mathcal{Y}$. For simplicity in the analysis, throughout the paper we assume that the adversary is oblivious, i.e., it decides the sequence $(x_1, y_1), \ldots, (x_T, y_T)$ at time $t = 0$ with complete knowledge about the forecaster's algorithm, although our results readily generalize to an adaptive adversary. For each $p \in \mathcal{Z}$, letting $\rho_{p,f} := \frac{\sum_{t=1}^{T} \mathbb{I}[p_t=p] f(x_t)(y_t-p)}{\sum_{t=1}^{T} \mathbb{I}[p_t=p]}$ be the empirical average of the correlation of $f$ with the residual sequence $\{y_t - p_t\}_{t=1}^{T}$ (conditioned on the rounds when the prediction made is $p_t = p$), the $\ell_q$-swap multicalibration error incurred by the forecaster is defined as

$$\mathsf{SMCal}_{\mathcal{F},q} := \sup_{\{f_p \in \mathcal{F}\}_{p \in \mathcal{Z}}} \sum_{p \in \mathcal{Z}} \left( \sum_{t=1}^{T} \mathbb{I}[p_t = p] \right) \left| \rho_{p,f_p} \right|^q = \sum_{p \in \mathcal{Z}} \left( \sum_{t=1}^{T} \mathbb{I}[p_t = p] \right) \sup_{f \in \mathcal{F}} \left| \rho_{p,f} \right|^q. \quad (1)$$

We also introduce a new notion — pseudo swap multicalibration, where the swap multicalibration error is measured via the conditional distributions $\mathcal{P}_1, \ldots, \mathcal{P}_T$ rather than the true realizations $p_1, \ldots, p_T$. Particularly, letting $\tilde{\rho}_{p,f} := \frac{\sum_{t=1}^{T} \mathcal{P}_t(p) f(x_t)(y_t-p)}{\sum_{t=1}^{T} \mathcal{P}_t(p)}$, we define the $\ell_q$-pseudo swap multicalibration error incurred by the forecaster as

$$\mathsf{PSMCal}_{\mathcal{F},q} := \sup_{\{f_p \in \mathcal{F}\}_{p \in \mathcal{Z}}} \sum_{p \in \mathcal{Z}} \left( \sum_{t=1}^{T} \mathcal{P}_t(p) \right) \left| \tilde{\rho}_{p,f_p} \right|^q = \sum_{p \in \mathcal{Z}} \left( \sum_{t=1}^{T} \mathcal{P}_t(p) \right) \sup_{f \in \mathcal{F}} \left| \tilde{\rho}_{p,f} \right|^q. \quad (2)$$

As we shall see, by not taking into account the randomness in the predictions, pseudo (swap) multicalibration is often easier to optimize. Note that, (pseudo) multicalibration ((P)MCal$_{\mathcal{F},q}$) is a special case of (pseudo) swap multicalibration where the comparator profile is $f_p = f$ for all $p \in \mathcal{Z}$:

$$\mathsf{MCal}_{\mathcal{F},q} := \sup_{f \in \mathcal{F}} \sum_{p \in \mathcal{Z}} \left( \sum_{t=1}^{T} \mathbb{I}[p_t = p] \right) \left| \rho_{p,f} \right|^q, \quad \mathsf{PMCal}_{\mathcal{F},q} := \sup_{f \in \mathcal{F}} \sum_{p \in \mathcal{Z}} \left( \sum_{t=1}^{T} \mathcal{P}_t(p) \right) \left| \tilde{\rho}_{p,f} \right|^q, \quad (3)$$

and calibration (Cal$_q$) is a further restriction of multicalibration to $\mathcal{F} = \{1\}$, where $1$ denotes the constant function that always outputs $1$. In this paper, we shall be primarily concerned with the $\ell_1, \ell_2$-(pseudo) (swap) multicalibration errors, which are related as $\mathsf{PSMCal}_{\mathcal{F},1} \leq \sqrt{T \cdot \mathsf{PSMCal}_{\mathcal{F},2}}$, $\mathsf{PMCal}_{\mathcal{F},1} \leq \sqrt{T \cdot \mathsf{PMCal}_{\mathcal{F},2}}$, $\mathsf{SMCal}_{\mathcal{F},1} \leq \sqrt{T \cdot \mathsf{SMCal}_{\mathcal{F},2}}$, $\mathsf{MCal}_{\mathcal{F},1} \leq \sqrt{T \cdot \mathsf{MCal}_{\mathcal{F},2}}$. The proof (skipped for brevity) follows trivially via the Cauchy-Schwartz inequality.

**Online (Swap) Omniprediction.** Omnipredictors are sufficient statistics that simultaneously encode loss-minimizing predictions for a broad class of loss functions $\mathcal{L}$. Since different losses expect different "types" of optimal predictions, the output of an omnipredictor $p$ has to be "post-processed" or "type-checked" via a loss specific function $k_\ell : [0, 1] \to [0, 1]$ to approximately minimize $\ell$ relative to the hypotheses in $\mathcal{F}$. The post-processing function $k_\ell$ is chosen to be a best-response function, i.e., $k_\ell(q) := \arg\min_{p \in [0,1]} \mathbb{E}_{y \sim \mathsf{Ber}(q)}[\ell(p, y)]$ denotes the prediction that minimizes the expected loss for $y \sim \mathsf{Ber}(q)$. Online omniprediction follows a similar learning protocol as that of online multicalibration described above, however, we equip our protocol with learners (parametrized by loss functions) who utilize the forecaster's predictions to choose actions. In particular, after the forecaster predicts $p_t$, each $\ell$-learner ($\ell \in \mathcal{L}$) chooses action $k_\ell(p_t)$ and incurs loss $\ell(k_\ell(p_t), y_t)$. The swap omniprediction error against a loss profile $\{\ell_p\}_{p \in \mathcal{Z}}$, comparator profile $\{f_p\}_{p \in \mathcal{Z}}$ is defined as

$$\mathsf{SOmni}\left( \{\ell_p\}_{p \in \mathcal{Z}}, \{f_p\}_{p \in \mathcal{Z}} \right) := \sum_{t=1}^{T} \ell_{p_t}(k_{\ell_{p_t}}(p_t), y_t) - \ell_{p_t}(f_{p_t}(x_t), y_t). \quad (4)$$

The swap omniprediction error incurred by the forecaster is then defined as a supremum over all loss, comparator profiles, i.e., $\mathsf{SOmni}_{\mathcal{L},\mathcal{F}} = \sup_{\{\ell_p \in \mathcal{L}, f_p \in \mathcal{F}\}_{p \in \mathcal{Z}}} \mathsf{SOmni}\left( \{\ell_p\}_{p \in \mathcal{Z}}, \{f_p\}_{p \in \mathcal{Z}} \right)$. Omniprediction is a special case of swap omniprediction where the loss, comparator profiles are fixed and independent of $p$.

In the distributional setting, a computational equivalence between swap multicalibration and swap omniprediction was established in Gopalan et al. (2023b) via an intermediate notion — swap agnostic learning, which was extended to the online setting by Garg et al. (2024) as contextual swap regret.

**Contextual Swap Regret.** Throughout the paper, we consider contextual swap regret for the squared loss. Contextual swap regret is a special case of online swap omniprediction when $\mathcal{L} = \{\ell\}$ and $\ell(p, y) = (p - y)^2$, so that $k_\ell(p) = p$. Similar to pseudo swap multicalibration, we also introduce a new notion — pseudo contextual swap regret:

$$\text{PSReg}_{\mathcal{F}} \coloneqq \sup_{\{f_p \in \mathcal{F}\}_{p \in \mathcal{Z}}} \sum_{t=1}^{T} \mathbb{E}_{p_t \sim \mathcal{P}_t} \left[ (p_t - y_t)^2 - (f_{p_t}(x_t) - y_t)^2 \right]. \tag{5}$$

**Other Notation.** For any $m \in \mathbb{N}$, let $[m] \coloneqq \{1, 2, \ldots, m\}$ denote the index set. We use bold lowercase letters to represent vectors and bold uppercase letters for matrices. The indicator function is denoted by $\mathbb{I}[\cdot]$, which evaluates to 1 if the condition inside the braces is true, and 0 otherwise. For any $k \in \mathbb{N}$, we define $\Delta_k$ as the $(k-1)$-dimensional probability simplex: $\Delta_k \coloneqq \{x \in \mathbb{R}^k; x_i \geq 0 \text{ for all } i \in [k], \sum_{i=1}^{k} x_i = 1\}$. More generally, for a set $\Omega$, let $\Delta_\Omega$ denote the set of all probability distributions over $\Omega$. A function $f : \mathcal{W} \to \mathbb{R}$ is said to be $\alpha$-*exp-concave* over a convex set $\mathcal{W}$ if the function $\exp(-\alpha f(w))$ is concave on $\mathcal{W}$. The following notation is used extensively throughout the paper: given a hypothesis class $\mathcal{F}$ and $\beta > 0$, we define the subset $\mathcal{F}_\beta \coloneqq \{f \in \mathcal{F}; f^2(x) \leq \beta \text{ for all } x \in \mathcal{X}\}$. Finally, we use the notations $\tilde{\Omega}(.), \tilde{\mathcal{O}}(.)$ to hide lower-order logarithmic terms.

**A note on the organization.** Due to the numerous concepts considered in this paper and the limited space available, we focus our detailed discussion on our improved bound for $\ell_2$-swap multicalibration and the path E → F → B in Figure 1, deferring the discussion of other results to the appendix. Readers are encouraged to keep the broader context presented in Figure 1 in mind and to note that our improved sample complexity bounds in the offline setting arise from an online-to-batch conversion.

## 1.2 Comparison with Related Work

We discuss comparison with the most relevant work, deferring additional discussions to Appendix A.

**Calibration.** For $\ell_2$-calibration ($\text{Cal}_2$), Foster and Hart (2023); Luo et al. (2025) showed that there exists an efficient algorithm that achieves $\text{Cal}_2 = \tilde{\mathcal{O}}(T^{\frac{1}{3}})$. The result of Luo et al. (2025) builds on a recent work by Fishelson et al. (2025), who showed that it is possible to achieve $\tilde{\mathcal{O}}(T^{\frac{1}{3}})$ $\ell_2$-pseudo calibration ($\text{PCal}_2$) by minimizing pseudo (non-contextual) swap regret of the squared loss. Building on the works of Luo et al. (2025); Fishelson et al. (2025), we observe that it is possible to achieve $\text{SMCal}_{\mathcal{F}_1^{\text{lin}}, 2} = \tilde{\mathcal{O}}(T^{\frac{1}{3}})$. In particular, our introduction of pseudo swap multicalibration and pseudo contextual swap regret is reminiscent of pseudo swap regret by Fishelson et al. (2025). Furthermore, our Freedman-based martingale analysis in going from $\ell_2$-pseudo swap multicalibration to $\ell_2$-swap multicalibration is largely motivated by a similar analysis by Luo et al. (2025) in going from $\ell_2$-pseudo calibration to $\ell_2$-calibration. Arguably, our result shows that $\ell_2$-swap multicalibration and $\ell_2$-calibration share the same upper bounds, despite the former being a stronger notion.

**Omniprediction.** Very recently, Okoroafor et al. (2025) have shown that it is possible to achieve oracle-efficient omniprediction, given access to an offline ERM oracle, with $\tilde{\mathcal{O}}(\varepsilon^{-2})$ sample complexity (matching the lower bound for the minimization of a fixed loss function) for the class of bounded variation loss functions $\mathcal{L}_{\text{BV}}$ against an arbitrary hypothesis class $\mathcal{F}$ with bounded statistical complexity. Notably, the class $\mathcal{L}_{\text{BV}}$ is quite broad and includes all convex functions, Lipschitz functions, proper losses, etc. At a high level, Okoroafor et al. (2025) obtain $\tilde{\mathcal{O}}(\sqrt{T})$ online omniprediction error by identifying proper calibration and multiaccuracy as sufficient conditions for omniprediction and proposing an algorithm based on the celebrated Blackwell approachability theorem (Blackwell, 1956; Abernethy et al., 2011) that simultaneously guarantees $\tilde{\mathcal{O}}(\sqrt{T})$ proper calibration and multiaccuracy errors. The result for (offline) omniprediction then follows from an online-to-batch conversion along with other technical subtleties to make the algorithm efficient (with respect to the offline ERM oracle). Next, we highlight our comparisons with Okoroafor et al. (2025)'s result and techniques. While Okoroafor et al. (2025) analyze omniprediction in a more general setting (for the class $\mathcal{L}_{\text{BV}}$ against an arbitrary $\mathcal{F}$), we study the harder notion of *swap* omniprediction but in a specialized setting (for $\mathcal{L}^{\text{cvx}} \subset \mathcal{L}_{\text{BV}}$, against $\mathcal{F}_{\text{res}}^{\text{aff}}$). Finally, as mentioned in the introduction, although our online-to-batch conversion in Section D.1 follows in a similar spirit to Okoroafor et al. (2025), it is substantially different due to a more involved martingale analysis via Freedman's inequality.

# 2 Achieving $\tilde{\mathcal{O}}(T^{\frac{1}{3}})$ $\ell_2$-Swap Multicalibration

In this section, we propose an efficient algorithm to minimize the $\ell_2$-swap multicalibration error. As per Figure 1, we shall reduce $\ell_2$-swap multicalibration to $\ell_2$-pseudo swap multicalibration, followed by a reduction to pseudo contextual swap regret. Finally, we shall propose an efficient algorithm to minimize the pseudo contextual swap regret.

## 2.1 From swap multicalibration to pseudo swap multicalibration

**Lemma 1.** *Let $\mathcal{F} \subset [-1, 1]^{\mathcal{X}}$ be a finite hypothesis class. For any algorithm $\mathcal{A}_{\mathsf{SMCal}_{\mathcal{F}}}$ such that for each $t \in [T]$ the conditional distribution $\mathcal{P}_t$ is deterministic, with probability at least $1 - \delta$ over $\mathcal{A}_{\mathsf{SMCal}_{\mathcal{F}}}$'s predictions, we have $\mathsf{SMCal}_{\mathcal{F},2} \leq 384(N+1) \log \left( \frac{6(N+1)|\mathcal{F}|}{\delta} \right) + 6 \cdot \mathsf{PSMCal}_{\mathcal{F},2}$.*

The proof of Lemma 1 is deferred to Appendix B.1 and is motivated by a recent result due to Luo et al. (2025) who derive a high probability bound (via Freedman's inequality) that relates the $\ell_2$-calibration error ($\mathsf{Cal}_2$) with the $\ell_2$-pseudo calibration error ($\mathsf{PCal}_2$). Particularly, our proof is an adaptation of the analysis provided by Luo et al. (2025) to the contextual setting. Note that the assumption that the conditional distribution $\mathcal{P}_t$ is deterministic is because the algorithm we propose for minimizing pseudo contextual swap regret is deterministic, therefore the only randomness lies in the sampling $p_t \sim \mathcal{P}_t$. In a way, our proposed algorithm in Section 2.3 correctly aligns with the assumption. However, we remark that this assumption can be relaxed to account for the randomness in $\mathcal{P}_t$.

Next, we instantiate our result for the class of linear functions with bounded norm, defined over the set $\mathcal{X} = \{x \in \mathbb{B}_2^d; x_1 = \frac{1}{2}\}$. The requirement that each $x \in \mathcal{X}$ has a constant first coordinate (equal to $1/2$) is without any loss of generality. More generally, we only require that each $x \in \mathcal{X}$ has a constant $i$-th coordinate (equal to $\Gamma$ for some $\Gamma \in (0,1)$) to ensure that the class $\mathcal{F}^{\mathsf{lin}} = \{f_\theta(x) = \langle \theta, x \rangle ; \theta \in \mathbb{R}^d\}$ is closed under affine transformations — a property that shall be required later. Although not explicitly mentioned in Garg et al. (2024), we realize that they also invoke this requirement. By definition, $\mathcal{F}_1^{\mathsf{lin}} = \{f \in \mathcal{F}^{\mathsf{lin}}; |f(x)| \leq 1 \text{ for all } x \in \mathcal{X}\}$ and the set of all $\theta$'s characterizing $\mathcal{F}_1^{\mathsf{lin}}$ satisfies $\mathbb{B}_2^d \subset \{\theta \in \mathbb{R}^d; |\langle \theta, x \rangle| \leq 1 \text{ for all } x \in \mathcal{X}\} \subset 2 \cdot \mathbb{B}_2^d$. Since $\mathcal{F}_1^{\mathsf{lin}}$ is infinite, the result of Lemma 1 does not immediately apply for the choice $\mathcal{F} = \mathcal{F}_1^{\mathsf{lin}}$. Instead, to bound $\mathsf{SMCal}_{\mathcal{F}_1^{\mathsf{lin}},2}$, we form a finite sized cover $\mathcal{C}_\varepsilon$ of $\mathcal{F}_1^{\mathsf{lin}}$, bound $\mathsf{SMCal}_{\mathcal{F}_1^{\mathsf{lin}},2}$ in terms of $\mathsf{SMCal}_{\mathcal{C}_\varepsilon,2}$, and use the result of Lemma 1 to bound $\mathsf{SMCal}_{\mathcal{C}_\varepsilon,2}$ in terms of $\mathsf{PSMCal}_{\mathcal{C}_\varepsilon,2}$. Recall that for a function class $\mathcal{F}$, a function class $\mathcal{G}$ is an $\varepsilon$-cover if for each $f \in \mathcal{F}$ there exists a $g \in \mathcal{G}$ such that $|f(x) - g(x)| \leq \varepsilon$ for all $x \in \mathcal{X}$. For each $f \in \mathcal{F}$, we refer to the corresponding function $g \in \mathcal{G}$ that realizes the condition as a *representative*. We use the following standard result (refer to (Luo, 2024, Proposition 2)) to bound $|\mathcal{C}_\varepsilon|$ in the proof of Lemma 2, whose proof is deferred to Appendix B.2.

**Proposition 1.** *There exists a cover $\mathcal{C}_\varepsilon \subseteq \mathcal{F}_1^{\mathsf{lin}}$ of $\mathcal{F}_1^{\mathsf{lin}}$ with $|\mathcal{C}_\varepsilon| = \mathcal{O}\left(\frac{1}{\varepsilon}\right)^d$.*

**Lemma 2.** *For the class $\mathcal{F}_1^{\mathsf{lin}}$ and any $\varepsilon > 0$, with probability at least $1 - \delta$, we have $\mathsf{SMCal}_{\mathcal{F}_1^{\mathsf{lin}},2} = \mathcal{O}\left(N \log \frac{N}{\delta} + Nd \log \frac{1}{\varepsilon} + \mathsf{PSMCal}_{\mathcal{F}_1^{\mathsf{lin}},2} + \varepsilon^2 T\right).$*

## 2.2 From pseudo swap multicalibration to pseudo contextual swap regret

In this section, we show that the $\ell_2$-pseudo swap multicalibration error is bounded by the pseudo contextual swap regret. Notably, our result is an adaptation of a similar result proved by Garg et al. (2024) for contextual swap regret (see also Globus-Harris et al. (2023); Gopalan et al. (2023b) for a similar characterization of swap multicalibration in terms of swap agnostic learning) to pseudo contextual swap regret. Our result holds for any arbitrary hypothesis class $\mathcal{F}$ satisfying the following mild assumption:

**Assumption 1.** *$\mathcal{F}$ is closed under affine transformations, i.e., for each $f \in \mathcal{F}$, the function $g(x) = af(x) + b \in \mathcal{F}$ for all $a, b \in \mathbb{R}$.*

We remark that the above is quite standard and has been explicitly assumed in Garg et al. (2024); Globus-Harris et al. (2023), and is implicit in Gopalan et al. (2023b). Clearly, $\mathcal{F}^{\mathsf{lin}}$ satisfies Assumption 1. This is because, for a $f \in \mathcal{F}$ that is determined by $\theta \in \mathbb{R}^d$, we have

$a \langle \theta, x \rangle + b = \langle a \cdot \theta + 2b \cdot e_1, x \rangle \in \mathcal{F}^{\mathsf{lin}}$. Before proving the main implication, we prove the following converse result.

**Lemma 3.** *Assume that there exists a $p \in \mathcal{Z}$, $f \in \mathcal{F}_1$ such that $\tilde{\rho}_{p,f} \geq \alpha$ for some $\alpha > 0$. Then, there exists a $f' \in \mathcal{F}_4$ such that $\frac{\sum_{t=1}^{T} \mathcal{P}_t(p)\left((p-y_t)^2 - (f'(x_t)-y_t)^2\right)}{\sum_{t=1}^{T} \mathcal{P}_t(p)} \geq \alpha^2$.*

The proof of Lemma 3 can be found in Appendix B.3 and is similar to (Garg et al., 2024, Theorem 3.2). In particular, we consider the function $f'(x) = p + \eta f(x)$, where $\eta = \min\left(1, \frac{\alpha}{\mu}\right)$, $\mu = \frac{\sum_{t=1}^{T} \mathcal{P}_t(p)(f(x_t))^2}{\sum_{t=1}^{T} \mathcal{P}_t(p)}$. Under Assumption 1, $f' \in \mathcal{F}$. Furthermore, since $f \in \mathcal{F}_1$, we have $f' \in \mathcal{F}_4$. It then follows from direct computation that $\frac{\sum_{t=1}^{T} \mathcal{P}_t(p)\left((p-y_t)^2 - (f'(x_t)-y_t)^2\right)}{\sum_{t=1}^{T} \mathcal{P}_t(p)} \geq 2\eta\alpha - \eta^2\mu$. Finally, we analyze the cases $\alpha \geq \mu$ and $\alpha < \mu$ and derive a common lower bound $2\eta\alpha - \eta^2\mu \geq \alpha^2$, thereby finishing the proof. Equipped with Lemma 3, we prove the main result of this section.

**Lemma 4.** *Assume that there exists $\alpha > 0$ such that $\mathsf{PSReg}_{\mathcal{F}_4} \leq \alpha$. Then, $\mathsf{PSMCal}_{\mathcal{F}_{1,2}} \leq \alpha$.*

The proof of Lemma 4 can be found in Appendix B.4 and follows by the method of contradiction, using the result of Lemma 3. Particularly, assuming that $\mathsf{PSMCal}_{\mathcal{F}_{1,2}} > \alpha$, we conclude that there exists a comparator profile $\{f_p\}_{p \in \mathcal{Z}}$ that realizes $\sum_{p \in \mathcal{Z}} \alpha_p > \alpha$, where $\alpha_p = \sum_{t=1}^{T} \mathcal{P}_t(p)(\tilde{\rho}_{p,f_p})^2$. By definition of $\alpha_p$, there exists a function $f_p^\star(x) \in \{f_p(x), -f_p(x)\}$ such that $\tilde{\rho}_{p,f_p^\star} = \sqrt{\frac{\alpha_p}{\sum_{t=1}^{T} \mathcal{P}_t(p)}}$ for all $p \in \mathcal{Z}$. By Lemma 3, for each $p \in \mathcal{Z}$, there exists a $f_p' \in \mathcal{F}_4$ that satisfies $\sum_{t=1}^{T} \mathcal{P}_t(p)((p-y_t)^2 - (f_p'(x_t)-y_t)^2) \geq \alpha_p$. Summing over all $p \in \mathcal{Z}$ we obtain a contradiction to the assumption that $\mathsf{PSReg}_{\mathcal{F}_4} \leq \alpha$.

Combining the result of Lemma 2 and 4, we observe that to bound $\mathsf{SMCal}_{\mathcal{F}_1^{\mathsf{lin}}}$, we only require a bound on $\mathsf{PSReg}_{\mathcal{F}_4^{\mathsf{lin}}, 2}$. In the next section, we propose an efficient algorithm to minimize $\mathsf{PSReg}_{\mathcal{F}_4^{\mathsf{lin}}}$.

### 2.3 Bound on the pseudo contextual swap regret

Now, we give an algorithm to minimize the pseudo contextual swap regret of the squared loss $\ell(p, y) = (p - y)^2$ against the hypothesis class $\mathcal{F}_4^{\mathsf{lin}}$. Recall that for our choice of $\mathcal{X}$, the set of $\theta$'s that determine $\mathcal{F}_4^{\mathsf{lin}}$ satisfies $2 \cdot \mathbb{B}_2^d \subset \{\theta \in \mathbb{R}^d; |\langle \theta, x \rangle| \leq 2 \text{ for all } x \in \mathcal{X}\} \subset 4 \cdot \mathbb{B}_2^d$. We consider the more general setting when $\mathcal{F}$ is arbitrary and then instantiate our result for $\mathcal{F}_4^{\mathsf{lin}}$. Our general algorithm (Algorithm 3) is based on the well-known Blum-Mansour (BM) reduction (Blum and Mansour, 2007).

Before proceeding further, we first recall the BM reduction (Algorithm 3 in Appendix B.5). Let $\mathcal{Z}$ be enumerated as $\mathcal{Z} = \{z_0, \ldots, z_N\}$, where $z_i = i/N$ for all $i \in \{0, \ldots, N\}$. The reduction maintains $N + 1$ external regret algorithms $\mathcal{A}_0, \ldots, \mathcal{A}_N$. At each time $t$, let $\boldsymbol{q}_{t,i} \in \Delta_{N+1}$ denote the probability distribution over $\mathcal{Z}$ produced by $\mathcal{A}_i$. Let $\boldsymbol{Q}_t = [\boldsymbol{q}_{t,0}, \ldots, \boldsymbol{q}_{t,N}]$ be the matrix formed by concatenating $\boldsymbol{q}_{t,0}, \ldots, \boldsymbol{q}_{t,N}$ as columns. Upon receiving the context $x_t$, we compute the stationary distribution of $\boldsymbol{Q}_t$, i.e., a distribution $\boldsymbol{p}_t \in \Delta_{N+1}$ satisfying $\boldsymbol{Q}_t \boldsymbol{p}_t = \boldsymbol{p}_t$. With $\boldsymbol{p}_t$ being our final distribution of predictions, i.e., $\mathcal{P}_t(z_i) = p_{t,i}$, we sample $p_t \sim \mathcal{P}_t$ and observe the outcome $y_t$. Thereafter, we feed the scaled loss function $p_{t,i}\ell(., y_t)$ to $\mathcal{A}_i$. Let $\boldsymbol{\ell}_{t,i}^{\mathsf{sc}} = p_{t,i}\boldsymbol{\ell}_t \in \mathbb{R}^{N+1}$ be a scaled loss vector, where $\boldsymbol{\ell}_t(j) = \ell(z_j, y_t)$. It immediately follows from Proposition 2 (Appendix B.5) that $\mathsf{PSReg}_{\mathcal{F}} \leq \sum_{i=0}^{N} \mathsf{Reg}_i(\mathcal{F})$, where $\mathsf{Reg}_i(\mathcal{F}) \coloneqq \sup_{f \in \mathcal{F}} \sum_{t=1}^{T} \langle \boldsymbol{q}_{t,i}, \boldsymbol{\ell}_{t,i}^{\mathsf{sc}} \rangle - p_{t,i}(f(x_t) - y_t)^2$, i.e., the pseudo swap regret is bounded by the sum of the (scaled) external regrets of the $N + 1$ algorithms. It remains to design the $i$-th external regret algorithm $\mathcal{A}_i$ that minimizes $\mathsf{Reg}_i(\mathcal{F})$. We emphasize that $\mathcal{A}_i$ is required to predict a distribution $\boldsymbol{q}_{t,i}$ over $\mathcal{Z}$ and is subsequently fed a scaled loss function $p_{t,i}\ell(., y_t)$ at each time $t$. To implement $\mathcal{A}_i$, we assume the following oracle ALG that solves a (scaled) external regret minimization problem for the squared loss (which will be later instantiated by a concrete algorithm for the linear class).

**Assumption 2.** *Let ALG be an algorithm for minimizing the (scaled) external regret against $\mathcal{F}$ in the following learning protocol: at every time $t \in [T]$, (a) an adversary selects context $x_t \in \mathcal{X}$, a scaling parameter $\alpha_t \in [0, 1]$, and the true label $y_t$; (b) the adversary reveals $x_t$; (c) ALG predicts $w_t \in [0, 1]$, observes $\alpha_t, y_t$, and incurs the scaled loss $\alpha_t\ell(w_t, y_t)$. We assume that ALG achieves the following external regret guarantee: $\mathbb{E}\left[\sup_{f \in \mathcal{F}} \sum_{t=1}^{T} \alpha_t(w_t - y_t)^2 - \alpha_t(f(x_t) - y_t)^2\right] \leq r(T, \mathcal{F})$, where*

$r(T, \mathcal{F})$ is a non-negative function that captures the regret bound of ALG, and the expectation is taken over the joint randomness in the adversary and the algorithm.

To instantiate $\mathcal{A}_i$ (Algorithm 1), we propose using $\mathsf{ALG}_i$ (an instance of ALG for the $i$-th external regret algorithm) along with the randomized rounding procedure of Fishelson et al. (2025). The rounding procedure is required because at each time $t$, $\mathsf{ALG}_i$ outputs $w_{t,i} \in [0, 1]$, however, $\mathcal{A}_i$ is required to predict a distribution $\boldsymbol{q}_{t,i} \in \Delta_{N+1}$ over $\mathcal{Z}$. Towards this end, Fishelson et al. (2025) proposed the randomized rounding scheme summarized in Algorithm 2. In Proposition 3 (Appendix B.6), we show that the change in the expected loss due to rounding, i.e., the quantity $\sum_{j=0}^N q_j \ell(z_j, y) - \ell(p, y)$ is at most $\frac{1}{N^2}$.

---

**Algorithm 1** The $i$-th external regret algorithm ($\mathcal{A}_i$)

---

1: **for** $t = 1, \ldots, T,$
2:      Set $w_{t,i} \in [0, 1]$ as the output of $\mathsf{ALG}_i$ at time $t$ (where $\mathsf{ALG}_i$ satisfies Assumption 2);
3:      Predict $\boldsymbol{q}_{t,i} = \mathsf{RRound}(w_{t,i})$ (Algorithm 2);
4:      Receive the scaled loss function $f_{t,i}(w) = p_{t,i} \ell(w, y_t)$ and feed it to $\mathsf{ALG}_i$.

---

---

**Algorithm 2** Randomized rounding ($\mathsf{RRound}(p)$)

---

**Input:** $p \in [0, 1]$, **Output:** Probability distribution $\boldsymbol{q} \in \Delta_{N+1}$;
**Scheme:** Let $p_i, p_{i+1}$ for some $i \in \{0, \ldots, N-1\}$ be two neighboring points in $\mathcal{Z}$ such that $p_i \leq p < p_{i+1}$; output $\boldsymbol{q} \in \Delta_{N+1}$, where $q_i = \frac{p_{i+1}-p}{p_{i+1}-p_i}$, $q_{i+1} = \frac{p-p_i}{p_{i+1}-p_i}$, and $q_j = 0$ otherwise.

---

Combining everything, we derive the regret guarantee $\mathsf{Reg}_i(\mathcal{F})$ of $\mathcal{A}_i$. It follows from Proposition 3 that at any time $t$, the distribution $\boldsymbol{q}_{t,i} = \mathsf{RRound}(w_{t,i})$ satisfies $\langle \boldsymbol{q}_{t,i}, \boldsymbol{\ell}_t \rangle \leq \ell(w_{t,i}, y_t) + \frac{1}{N^2}$. Multiplying with $p_{t,i}$ and summing over all $t$, we obtain

$$\mathsf{Reg}_i(\mathcal{F}) \leq \sup_{f \in \mathcal{F}} \left( \sum_{t=1}^T p_{t,i}(w_{t,i} - y_t)^2 - p_{t,i}(f(x_t) - y_t)^2 \right) + \frac{1}{N^2} \sum_{t=1}^T p_{t,i} \leq r_i(T, \mathcal{F}) + \frac{1}{N^2} \sum_{t=1}^T p_{t,i},$$

where $r_i(T, \mathcal{F})$ denotes the external regret bound of $\mathsf{ALG}_i$ (Assumption 2). It then follows from Proposition 2 that

$$\mathsf{PSReg}_{\mathcal{F}} \leq \sum_{i=0}^N \mathsf{Reg}_i(\mathcal{F}) \leq \sum_{i=0}^N r_i(T, \mathcal{F}) + \frac{1}{N^2} \sum_{i=0}^N \sum_{t=1}^T p_{t,i} = \sum_{i=0}^N r_i(T, \mathcal{F}) + \frac{T}{N^2}.$$

When $\mathcal{F} = \mathcal{F}_4^{\mathsf{lin}}$, we can instantiate each $\mathsf{ALG}_i$ with the Online Newton Step algorithm (ONS) (Hazan et al., 2007) corresponding to the scaled loss $\phi_{t,i}(\theta) \coloneqq p_{t,i}(\langle \theta, x_t \rangle - y_t)^2$, which is $\frac{1}{50}$-exp-concave and 10-Lipschitz over $4 \cdot \mathbb{B}_2^d$ (Proposition 4 in Appendix B.7). We propose to employ ONS over the set $4 \cdot \mathbb{B}_2^d$ since the set of all $\theta$'s characterizing $\mathcal{F}_4^{\mathsf{lin}}$ is a subset of $4 \cdot \mathbb{B}_2^d$. $\mathsf{ONS}_i$ (Algorithm 4 in Appendix B.5) represents an instance of ONS for $\mathsf{ALG}_i$. On updating $\theta_{t,i}$ via $\mathsf{ONS}_i$, $\mathsf{ALG}_i$ (Algorithm 5) simply predicts $w_{t,i} = \mathsf{Proj}_{[0,1]}(\langle \theta_{t,i}, x_t \rangle)$, where $\mathsf{Proj}_{[0,1]}$ denotes the projection to $[0, 1]$, i.e., $\mathsf{Proj}_{[0,1]}(x) \coloneqq \operatorname{argmin}_{y \in [0,1]} |x - y|$.

The following lemma (due to Hazan et al. (2007)) bounds the regret of $\mathsf{ONS}_i$.

**Lemma 5.** *The regret of $\mathsf{ONS}_i$ can be bounded as* $\sup_{\theta \in 4 \cdot \mathbb{B}_2^d} \phi_{t,i}(\theta_{t,i}) - \phi_{t,i}(\theta) = \mathcal{O}(d \log T)$.

The regret of $\mathsf{ALG}_i$ can then be bounded as $\sup_{\theta \in 4 \cdot \mathbb{B}_2^d} \sum_{t=1}^T p_{t,i}(w_{t,i} - y_t)^2 - p_{t,i}(\langle \theta, x_t \rangle - y_t)^2 \leq \sup_{\theta \in 4 \cdot \mathbb{B}_2^d} \sum_{t=1}^T \phi_{t,i}(\theta_{t,i}) - \phi_{t,i}(\theta) = \mathcal{O}(d \log T)$, where the first inequality follows since $(\mathsf{Proj}(\langle \theta_{t,i}, x_t \rangle) - y_t)^2 \leq (\langle \theta_{t,i}, x_t \rangle - y_t)^2$. The above bound implies that $r_i(T, \mathcal{F}_4^{\mathsf{lin}}) = \mathcal{O}(d \log T)$ and $\mathsf{PSReg}_{\mathcal{F}_4^{\mathsf{lin}}} \leq \mathcal{O}\left(Nd \log T + \frac{T}{N^2}\right)$. Combining the result of Lemma 2 and Lemma 4 with the bound on $\mathsf{PSReg}_{\mathcal{F}_4^{\mathsf{lin}}}$, we obtain the following theorem.

**Theorem 1.** *There exists an efficient algorithm that achieves the following bound:* $\mathsf{SMCal}_{\mathcal{F}_1^{\mathsf{lin}}, 2} = \mathcal{O}(T^{\frac{1}{3}} d^{\frac{2}{3}} (\log T)^{\frac{2}{3}} + (\frac{T}{d \log T})^{\frac{1}{3}} \log \frac{1}{\delta})$ *with probability* $\geq 1 - \delta$. *Furthermore,* $\mathbb{E}\left[\mathsf{SMCal}_{\mathcal{F}_1^{\mathsf{lin}}, 2}\right] = \mathcal{O}(T^{\frac{1}{3}} d^{\frac{2}{3}} (\log T)^{\frac{2}{3}})$, *where the expectation is taken over the internal randomness of the algorithm.*

Theorem 1 answers the open problem raised by Garg et al. (2024). Compared to our result, Garg et al. (2024) showed that $\mathsf{SMCal}_{\mathcal{F}_1^{\mathsf{lin}},2} = \tilde{\mathcal{O}}\left(dT^{\frac{3}{4}}\sqrt{\log\frac{1}{\delta}}\right)$ with probability at least $1 - \delta$. An immediate corollary of Theorem 1 is an improved $\tilde{\mathcal{O}}(T^{\frac{2}{3}})$ bound on $\mathsf{SMCal}_{\mathcal{F}_1^{\mathsf{lin}},1}$ (refer Corollary 1 in Appendix B.8).

## 3  Conclusion and Future Directions

In this paper, we obtained state-of-the-art regret/error bounds (in the online setting) and sample complexity bounds (in the distributional setting) for swap multicalibration, swap omniprediction, and swap agnostic learning. Crucially, our bound for swap multicalibration is derived in the context of linear functions, and that for swap omniprediction is obtained specifically for convex Lipschitz functions against a hypothesis class that comprises linear functions. A natural question is whether it is possible to extend our framework (Figure 1) to arbitrary hypothesis, loss classes and obtain oracle-efficient algorithms (similar to Garg et al. (2024); Okoroafor et al. (2025)). Moreover, recall that we obtained $\tilde{\mathcal{O}}(T^{\frac{1}{3}})$ and $\tilde{\mathcal{O}}(T^{\frac{3}{5}})$ bounds for pseudo contextual swap regret and contextual swap regret, respectively, which is in contrast to the non-contextual setting, where both quantities enjoy the favorable $\tilde{\mathcal{O}}(T^{\frac{1}{3}})$ rate (Foster and Hart, 2023; Luo et al., 2025; Fishelson et al., 2025). The $\tilde{\mathcal{O}}(T^{\frac{3}{5}})$ bound for contextual swap regret is a limitation of our analysis in Section C.2; we suspect this can be improved by a more sophisticated analysis which can also improve the sample complexity of swap agnostic learning as a by product. This improvement shall manifest in further improvements in the sample complexity of swap multicalibration as per the discussion in Section D.3.

## Acknowledgements

We thank Princewill Okoroafor and Michael P. Kim for several helpful discussions. This work was supported in part by NSF CAREER Awards CCF-2239265 and IIS-1943607 and an Amazon Research Award (2024). Any opinions, findings, and conclusions or recommendations expressed in this material are those of the author(s) and do not reflect the views of sponsors such as Amazon or NSF.

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

# Contents

# A   Additional Related Work and Notations

**Downstream decision making.**   The seminal work of Foster and Vohra (1998) proposed the first algorithm for online calibration that achieved $\mathbb{E}[\mathsf{Cal}_1] = \tilde{\mathcal{O}}(T^{\frac{2}{3}})$. For $\ell_1$-calibration ($\mathsf{Cal}_1$), a lower bound of $\Omega(\sqrt{T})$ was folklore, and recent breakthroughs by Qiao and Valiant (2021); Dagan et al. (2024) have made progress towards closing the gap between the lower and upper bounds. In particular, Dagan et al. (2024) proved that for any online algorithm, there exists an adversary such that $\mathbb{E}[\mathsf{Cal}_1] = \Omega(T^{0.54389})$, and also proved the existence of an algorithm (a non-constructive proof) that achieves $\mathbb{E}[\mathsf{Cal}_1] = \mathcal{O}(T^{\frac{2}{3}-\varepsilon})$, where $\varepsilon > 0$. Understanding the limitations of $\ell_1$-calibration, i.e., it is impossible to achieve $\sqrt{T}$ $\ell_1$-calibration, has led to the introduction of several weaker notions of calibration, e.g., continuous calibration (Foster and Hart, 2021), decision calibration (Zhao et al., 2021; Noarov et al., 2023; Tang et al., 2025), U-calibration (Kleinberg et al., 2023; Luo et al., 2024), distance to calibration (Błasiok et al., 2023; Qiao and Zheng, 2024; Arunachaleswaran et al., 2025), maximum swap regret (Roth and Shi, 2024; Hu and Wu, 2024), subsampled smooth calibration error (Haghtalab et al., 2024), etc. The above measures are still meaningful for downstream tasks and can be achieved at more favorable rates. On the contrary, the work by Luo et al. (2025) introduced the notion of KL-calibration, which is arguably a stronger measure for studying upper bounds than $\ell_2$-calibration and showed that KL-calibration simultaneously bounds swap regret for several important subclasses of proper losses while being achievable at $\tilde{\mathcal{O}}(T^{\frac{1}{3}})$ rate. A recent work by Collina et al. (2025) (see also Collina et al. (2024)) considered the problem of online collaborative prediction, where over a sequence of $T$ days, two parties, Alice and Bob, each with their context (the context of Alice is unknown to Bob and vice-versa), engage in a communication protocol to solve a squared error regression problem defined over the joint context space. Using the bound for contextual swap regret as derived by Garg et al. (2024), Collina et al. (2025) propose an online collaboration protocol that achieves $\tilde{\mathcal{O}}(T^{\frac{55}{56}})$ external regret against the class of bounded linear functions. Not surprisingly, this bound can be improved using our algorithm in Section 2.3 instead.

**(Swap) Multicalibration.**   Since its inception, starting with the work of Hébert-Johnson et al. (2018), multicalibration has found surprising connections with several domains, e.g., computational complexity (Casacuberta et al., 2024), algorithmic fairness (Hébert-Johnson et al., 2018; Obermeyer et al., 2019; Devic et al., 2024; Gopalan et al., 2022b), learning theory (Gopalan et al., 2022a, 2023a; Gollakota et al., 2023; Globus-Harris et al., 2023), conformal prediction (Bastani et al., 2022), online learning (Gupta et al., 2022; Jung et al., 2021; Haghtalab et al., 2023), cryptography (Dwork et al., 2023), etc. However, the literature on multicalibration has differed in the concrete definition, thereby leading to some confusion. In this paper, we primarily adopt the definition given by Globus-Harris et al. (2023); Gopalan et al. (2023b) in the distributional setting and its extension by Garg et al. (2024) to the online setting. The previously best-known sample complexity bounds as mentioned in Table 1 are with respect to these definitions.

For $\ell_\infty$-multicalibration, Haghtalab et al. (2023) derived a $\tilde{\mathcal{O}}(\varepsilon^{-2})$ sample complexity when randomized predictors are allowed, and a $\tilde{\mathcal{O}}(\varepsilon^{-4})$ sample complexity for deterministic predictors. Notably, since $\ell_1, \ell_\infty$-multicalibration errors are related as $\ell_1 \leq |\mathcal{Z}| \cdot \ell_\infty$, their result implies a $\tilde{\mathcal{O}}(\varepsilon^{-2})$ sample complexity for $\ell_1$-multicalibration (the bound for $\ell_\infty$-multicalibration has a logarithmic dependence on $|\mathcal{Z}|$, therefore $|\mathcal{Z}|$ can be chosen to be $\mathcal{O}(1)$). However, Haghtalab et al. (2023) use a bucketed definition of multicalibration which is different from that considered in this paper, where we enforce our predictor to predict among $|\mathcal{Z}|$ possible values. For swap-multicalibration, the multicalibration algorithms of Hébert-Johnson et al. (2018); Gopalan et al. (2022a) were shown to be swap-multicalibrated in Gopalan et al. (2023b), thereby establishing a $\tilde{\mathcal{O}}(\varepsilon^{-10})$ sample complexity for $\ell_1$-swap multicalibration. For $\ell_2$-swap multicalibration, Globus-Harris et al. (2023) proposed an algorithm that achieved $\ell_2$-multicalibration error at most $\varepsilon$ given $\gtrsim \varepsilon^{-5}$ samples. However, we realize that their algorithm is in fact swap multicalibrated, thereby establishing a $\tilde{\mathcal{O}}(\varepsilon^{-5})$ sample complexity for $\ell_2$-swap multicalibration. Since $\ell_1, \ell_2$-swap multicalibration errors are related as $\ell_1 \leq \sqrt{\ell_2}$, their result also implies a $\tilde{\mathcal{O}}(\varepsilon^{-10})$ sample complexity for $\ell_1$-swap multicalibration matching that of Gopalan et al. (2023b), albeit with a remarkably simpler algorithm. Since $\varepsilon$ $\ell_1$-swap multicalibration error implies $\mathcal{O}(\varepsilon)$ swap omniprediction error for $\mathcal{L}^{\mathsf{cvx}}$ (Gopalan et al., 2023b), the above discussion implies a $\tilde{\mathcal{O}}(\varepsilon^{-10})$ sample complexity for swap omniprediction for $\mathcal{L}^{\mathsf{cvx}}$. As mentioned, although the multicalibration algorithm of Haghtalab et al. (2023) requires fewer samples, their considered

definition of multicalibration is different and it is not clear whether they achieve the stronger swap multicalibration guarantee, therefore, we do not portray a comparison to their results in Table 1.

**Omniprediction.** For the class of convex and Lipschitz functions, Gopalan et al. (2022a) proposed the first construction of an efficient omnipredictor via $\ell_1$-multicalibration. However, as shown by Gopalan et al. (2022a), multicalibration is not necessary for omniprediction. Several follow-up works (Gopalan et al., 2023a; Okoroafor et al., 2025) have investigated weaker notions of multicalibration that suffice for omniprediction. Particularly, Gopalan et al. (2023a) identified calibrated multiaccuracy to imply omniprediction for more general classes of loss functions (beyond convexity) and proposed an oracle-efficient algorithm (given access to an offline weak agnostic learning oracle) that required $\gtrsim \varepsilon^{-10}$ samples. Subsequent work by Hu et al. (2024) proposed an efficient construction of omnipredictors for single index models, requiring $\gtrsim \varepsilon^{-4}$ samples. Very recently, Okoroafor et al. (2025) have shown that it is possible to achieve oracle-efficient omniprediction, given access to an offline ERM oracle with $\tilde{\mathcal{O}}(\varepsilon^{-2})$ sample complexity (matching the lower bound for the minimization of a fixed loss function) for the class of bounded variation loss functions $\mathcal{L}_{\mathsf{BV}}$ against an arbitrary hypothesis class $\mathcal{F}$ with bounded statistical complexity, thereby settling the sample complexity of omniprediction.

In the context of swap omniprediction, a recent work by Lu et al. (2025) proposed a notion of decision swap regret for high-dimensional predictions in the regression setting, i.e., $\mathcal{Y} = [0, 1]$. However, compared to swap omniprediction, the loss function is not indexed by the forecaster's prediction, and notably, the techniques in our paper are considerably different than those proposed by Lu et al. (2025) who impose a relaxation of calibration called decision calibration (Zhao et al., 2021; Noarov et al., 2023; Tang et al., 2025) to achieve low decision swap regret.

**Additional Notations.** For a set $\mathcal{I}$, its complement is denoted by $\bar{\mathcal{I}}$, representing all elements not in $\mathcal{I}$. We denote conditional probability and expectation given the history up to time $t - 1$ (inclusive) by $\mathbb{P}_t$ and $\mathbb{E}_t$, respectively. A loss $\ell : [0, 1] \times \{0, 1\} \to \mathbb{R}$ is called *proper* if $\mathbb{E}_{y \sim \mathsf{Ber}(p)}[\ell(p, y)] \leq \mathbb{E}_{y \sim \mathsf{Ber}(p)}[\ell(p', y)]$ for all $p, p' \in [0, 1]$, e.g., the squared loss $\ell(p, y) = (p - y)^2$, log loss $\ell(p, y) = -y \log p - (1 - y) \log(1 - p)$, etc.

# B Deferred Proofs and Discussion in Section 2

## B.1 Proof of Lemma 1

**Lemma 6** (Freedman's Inequality). *(Beygelzimer et al., 2011, Theorem 1) Let $X_1, \ldots, X_n$ be a martingale difference sequence where $|X_i| \leq B$ for all $i = 1, \ldots, n$, and $B$ is a fixed constant. Define $V := \sum_{i=1}^{n} \mathbb{E}_i[X_i^2]$. Then, for any fixed $\mu \in \left[0, \frac{1}{B}\right], \delta \in [0, 1]$, with probability at least $1 - \delta$, we have*

$$\left| \sum_{i=1}^{n} X_i \right| \leq \mu V + \frac{\log \frac{2}{\delta}}{\mu}.$$

*Proof of Lemma 1.* Fix a $p \in \mathcal{Z}, f \in \mathcal{F}$. We first bound $|\rho_{p,f} - \tilde{\rho}_{p,f}|$. To achieve so, we consider the martingale difference sequences $\{X_t\}, \{Y_t\}, \{Z_t\}$, where

$$X_t := y_t f(x_t)(\mathcal{P}_t(p) - \mathbb{I}[p_t = p]), \quad Y_t := f(x_t)(\mathcal{P}_t(p) - \mathbb{I}[p_t = p]), \quad Z_t := \mathcal{P}_t(p) - \mathbb{I}[p_t = p].$$

Clearly, $|Z_t| \leq 1$ for all $t \in [T]$. Furthermore, since $f \in \mathcal{F} \subset [-1, 1]^{\mathcal{X}}$, we have $|X_t| \leq 1, |Y_t| \leq 1$ for all $t \in [T]$. Fix a $\mu_p \in [0, 1]$. Applying Lemma 6 to the sequences $X, Y, Z$ and taking a union bound (over $X, Y, Z$), we obtain that with probability at least $1 - \delta$ (simultaneously over $X, Y, Z$),

$$\left| \sum_{t=1}^{T} y_t f(x_t)(\mathcal{P}_t(p) - \mathbb{I}[p_t = p]) \right| \leq \mu_p V_X + \frac{\log \frac{6}{\delta}}{\mu_p}, \tag{6}$$

$$\left| \sum_{t=1}^{T} f(x_t)(\mathcal{P}_t(p) - \mathbb{I}[p_t = p]) \right| \leq \mu_p V_Y + \frac{\log \frac{6}{\delta}}{\mu_p}, \tag{7}$$

$$\left| \sum_{t=1}^{T} \mathcal{P}_t(p) - \mathbb{I}[p_t = p] \right| \leq \mu_p V_Z + \frac{\log \frac{6}{\delta}}{\mu_p}, \tag{8}$$

where $V_X, V_Y, V_Z$ are given by

$$V_X = \sum_{t=1}^{T} \mathbb{E}_t\left[X_t^2\right] = \sum_{t=1}^{T} y_t(f(x_t))^2 \mathcal{P}_t(p)(1 - \mathcal{P}_t(p)) \le \sum_{t=1}^{T} \mathcal{P}_t(p),$$

$$V_Y = \sum_{t=1}^{T} \mathbb{E}_t\left[Y_t^2\right] = \sum_{t=1}^{T} (f(x_t))^2 \mathcal{P}_t(p)(1 - \mathcal{P}_t(p)) \le \sum_{t=1}^{T} \mathcal{P}_t(p),$$

$$V_Z = \sum_{t=1}^{T} \mathbb{E}_t\left[Z_t^2\right] = \sum_{t=1}^{T} \mathcal{P}_t(p)(1 - \mathcal{P}_t(p)) \le \sum_{t=1}^{T} \mathcal{P}_t(p).$$

To bound $|\rho_{p,f} - \tilde{\rho}_{p,f}|$, we first upper bound $\tilde{\rho}_{p,f} - \rho_{p,f}$ as per the following steps:

$$\tilde{\rho}_{p,f} - \rho_{p,f} =$$

$$\frac{\sum_{t=1}^{T} \mathcal{P}_t(p)f(x_t)y_t}{\sum_{t=1}^{T} \mathcal{P}_t(p)} - \frac{\sum_{t=1}^{T} \mathbb{I}[p_t = p]f(x_t)y_t}{\sum_{t=1}^{T} \mathbb{I}[p_t = p]} +$$

$$p\left(\frac{\sum_{t=1}^{T} \mathbb{I}[p_t = p]f(x_t)}{\sum_{t=1}^{T} \mathbb{I}[p_t = p]} - \frac{\sum_{t=1}^{T} \mathcal{P}_t(p)f(x_t)}{\sum_{t=1}^{T} \mathcal{P}_t(p)}\right)$$

$$\le \frac{\mu_p \sum_{t=1}^{T} \mathcal{P}_t(p) + \frac{1}{\mu_p}\log\frac{6}{\delta} + \sum_{t=1}^{T} \mathbb{I}[p_t = p]f(x_t)y_t}{\sum_{t=1}^{T} \mathcal{P}_t(p)} - \frac{\sum_{t=1}^{T} \mathbb{I}[p_t = p]f(x_t)y_t}{\sum_{t=1}^{T} \mathbb{I}[p_t = p]} +$$

$$p\left(\frac{\sum_{t=1}^{T} \mathbb{I}[p_t = p]f(x_t)}{\sum_{t=1}^{T} \mathbb{I}[p_t = p]} + \frac{\mu_p \sum_{t=1}^{T} \mathcal{P}_t(p) + \frac{1}{\mu_p}\log\frac{6}{\delta} - \sum_{t=1}^{T} \mathbb{I}[p_t = p]f(x_t)}{\sum_{t=1}^{T} \mathcal{P}_t(p)}\right)$$

$$= \mu_p + \frac{\log\frac{6}{\delta}}{\mu_p \sum_{t=1}^{T} \mathcal{P}_t(p)} + \frac{\sum_{t=1}^{T} \mathbb{I}[p_t = p]f(x_t)y_t}{\sum_{t=1}^{T} \mathbb{I}[p_t = p] \cdot \sum_{t=1}^{T} \mathcal{P}_t(p)}\left(\sum_{t=1}^{T} \mathbb{I}[p_t = p] - \mathcal{P}_t(p)\right) +$$

$$p\left(\mu_p + \frac{\log\frac{6}{\delta}}{\mu_p \sum_{t=1}^{T} \mathcal{P}_t(p)} + \frac{\sum_{t=1}^{T} \mathbb{I}[p_t = p]f(x_t)}{\sum_{t=1}^{T} \mathbb{I}[p_t = p] \cdot \sum_{t=1}^{T} \mathcal{P}_t(p)}\left(\sum_{t=1}^{T} \mathcal{P}_t(p) - \mathbb{I}[p_t = p]\right)\right)$$

$$\le \mu_p + \frac{\log\frac{6}{\delta}}{\mu_p \sum_{t=1}^{T} \mathcal{P}_t(p)} + \left|\frac{\sum_{t=1}^{T} \mathbb{I}[p_t = p]f(x_t)y_t}{\sum_{t=1}^{T} \mathbb{I}[p_t = p] \cdot \sum_{t=1}^{T} \mathcal{P}_t(p)}\right|\left(\mu_p \sum_{t=1}^{T} \mathcal{P}_t(p) + \frac{1}{\mu_p}\log\frac{6}{\delta}\right) +$$

$$p\left(\mu_p + \frac{\log\frac{6}{\delta}}{\mu_p \sum_{t=1}^{T} \mathcal{P}_t(p)} + \left|\frac{\sum_{t=1}^{T} \mathbb{I}[p_t = p]f(x_t)}{\sum_{t=1}^{T} \mathbb{I}[p_t = p] \cdot \sum_{t=1}^{T} \mathcal{P}_t(p)}\right|\left(\mu_p \sum_{t=1}^{T} \mathcal{P}_t(p) + \frac{1}{\mu_p}\log\frac{6}{\delta}\right)\right)$$

$$\le 4\mu_p + \frac{4\log\frac{6}{\delta}}{\mu_p \sum_{t=1}^{T} \mathcal{P}_t(p)},$$

where the first inequality follows from (6), (7); the second inequality follows from (8); the final inequality follows since $f \in \mathcal{F}$.

Proceeding in a similar manner, we can upper bound $\rho_{p,f} - \tilde{\rho}_{p,f}$. We have,

$$\rho_{p,f} - \tilde{\rho}_{p,f}$$

$$= \frac{\sum_{t=1}^{T} \mathbb{I}[p_t = p]f(x_t)y_t}{\sum_{t=1}^{T} \mathbb{I}[p_t = p]} - \frac{\sum_{t=1}^{T} \mathcal{P}_t(p)f(x_t)y_t}{\sum_{t=1}^{T} \mathcal{P}_t(p)} +$$

$$p\left(\frac{\sum_{t=1}^{T} \mathcal{P}_t(p)f(x_t)}{\sum_{t=1}^{T} \mathcal{P}_t(p)} - \frac{\sum_{t=1}^{T} \mathbb{I}[p_t = p]f(x_t)}{\sum_{t=1}^{T} \mathbb{I}[p_t = p]}\right)$$

$$\le \frac{\sum_{t=1}^{T} \mathbb{I}[p_t = p]f(x_t)y_t}{\sum_{t=1}^{T} \mathbb{I}[p_t = p]} + \frac{\mu_p \sum_{t=1}^{T} \mathcal{P}_t(p) + \frac{1}{\mu_p}\log\frac{6}{\delta} - \sum_{t=1}^{T} \mathbb{I}[p_t = p]f(x_t)y_t}{\sum_{t=1}^{T} \mathcal{P}_t(p)} +$$

$$p\left(\frac{\mu_p \sum_{t=1}^{T} \mathcal{P}_t(p) + \frac{1}{\mu_p}\log\frac{6}{\delta} + \sum_{t=1}^{T} \mathbb{I}[p_t = p]f(x_t)}{\sum_{t=1}^{T} \mathcal{P}_t(p)} - \frac{\sum_{t=1}^{T} \mathbb{I}[p_t = p]f(x_t)}{\sum_{t=1}^{T} \mathbb{I}[p_t = p]}\right)$$

$$= \mu_p + \frac{\log \frac{6}{\delta}}{\mu_p \sum_{t=1}^T \mathcal{P}_t(p)} + \frac{\sum_{t=1}^T \mathbb{I}[p_t = p] f(x_t) y_t}{\sum_{t=1}^T \mathbb{I}[p_t = p] \cdot \sum_{t=1}^T \mathcal{P}_t(p)} \left( \sum_{t=1}^T \mathcal{P}_t(p) - \mathbb{I}[p_t = p] \right) +$$

$$p \left( \mu_p + \frac{\log \frac{6}{\delta}}{\mu_p \sum_{t=1}^T \mathcal{P}_t(p)} + \frac{\sum_{t=1}^T \mathbb{I}[p_t = p] f(x_t)}{\sum_{t=1}^T \mathbb{I}[p_t = p] \cdot \sum_{t=1}^T \mathcal{P}_t(p)} \left( \sum_{t=1}^T \mathbb{I}[p_t = p] - \mathcal{P}_t(p) \right) \right)$$

$$\leq \mu_p + \frac{\log \frac{6}{\delta}}{\mu_p \sum_{t=1}^T \mathcal{P}_t(p)} + \left| \frac{\sum_{t=1}^T \mathbb{I}[p_t = p] f(x_t) y_t}{\sum_{t=1}^T \mathbb{I}[p_t = p] \cdot \sum_{t=1}^T \mathcal{P}_t(p)} \right| \left( \mu_p \sum_{t=1}^T \mathcal{P}_t(p) + \frac{1}{\mu_p} \log \frac{6}{\delta} \right) +$$

$$p \left( \mu_p + \frac{\log \frac{6}{\delta}}{\mu_p \sum_{t=1}^T \mathcal{P}_t(p)} + \left| \frac{\sum_{t=1}^T \mathbb{I}[p_t = p] f(x_t)}{\sum_{t=1}^T \mathbb{I}[p_t = p] \cdot \sum_{t=1}^T \mathcal{P}_t(p)} \right| \left( \mu_p \sum_{t=1}^T \mathcal{P}_t(p) + \frac{1}{\mu_p} \log \frac{6}{\delta} \right) \right)$$

$$\leq 4\mu_p + \frac{4 \log \frac{6}{\delta}}{\mu_p \sum_{t=1}^T \mathcal{P}_t(p)}.$$

Combining both the bounds obtained above, we have shown that $|\rho_{p,f} - \tilde{\rho}_{p,f}| \leq 4\mu_p + \frac{4 \log \frac{6}{\delta}}{\mu_p \sum_{t=1}^T \mathcal{P}_t(p)}$. Taking a union bound over all $p \in \mathcal{Z}, f \in \mathcal{F}$, with probability at least $1 - \delta$, we have (simultaneously for all $p \in \mathcal{Z}, f \in \mathcal{F}$)

$$|\rho_{p,f} - \tilde{\rho}_{p,f}| \leq 4\mu_p + \frac{4 \log \frac{6(N+1)|\mathcal{F}|}{\delta}}{\mu_p \sum_{t=1}^T \mathcal{P}_t(p)}, \quad \left| \sum_{t=1}^T \mathcal{P}_t(p) - \mathbb{I}[p_t = p] \right| \leq \mu_p \sum_{t=1}^T \mathcal{P}_t(p) + \frac{\log \frac{6(N+1)|\mathcal{F}|}{\delta}}{\mu_p}.$$
$$(9)$$

Consider the function $g(\mu) := \mu + \frac{a}{\mu}$, where $a \geq 0$ is a fixed constant. Clearly, $\min_{\mu \in [0,1]} g(\mu) = 2\sqrt{a}$ when $a \leq 1$, and $1 + a$ otherwise. Minimizing the bound in (9) with respect to $\mu_p$, we obtain

$$|\rho_{p,f} - \tilde{\rho}_{p,f}| \leq 8 \sqrt{\frac{\log \frac{6(N+1)|\mathcal{F}|}{\delta}}{\sum_{t=1}^T \mathcal{P}_t(p)}} \text{ if } \log \frac{6(N+1)|\mathcal{F}|}{\delta} \leq \sum_{t=1}^T \mathcal{P}_t(p).$$

Moreover, since $f \in \mathcal{F}$, by definition we have $|\rho_{p,f}| \leq 1, |\tilde{\rho}_{p,f}| \leq 1$ and thus $|\rho_{p,f} - \tilde{\rho}_{p,f}| \leq 2$. However, when $\sum_{t=1}^T \mathcal{P}_t(p)$ is quite small (e.g., $\to 0$), the bound on $|\rho_{p,f} - \tilde{\rho}_{p,f}|$ obtained above is much worse than the trivial bound $|\rho_{p,f} - \tilde{\rho}_{p,f}| \leq 2$. Therefore, we define the set

$$\mathcal{I} := \left\{ p \in \mathcal{Z} \text{ s.t. } \log \frac{6(N+1)|\mathcal{F}|}{\delta} \leq \sum_{t=1}^T \mathcal{P}_t(p) \right\}$$

and bound $|\rho_{p,f} - \tilde{\rho}_{p,f}|$ as

$$|\rho_{p,f} - \tilde{\rho}_{p,f}| \leq \begin{cases} 8 \sqrt{\frac{\log \frac{6(N+1)|\mathcal{F}|}{\delta}}{\sum_{t=1}^T \mathcal{P}_t(p)}} & \text{if } p \in \mathcal{I}, \\ 2 & \text{otherwise.} \end{cases} \tag{10}$$

Similarly, by minimizing (9) with respect to $\mu_p$, we obtain the following bound:

$$\left| \sum_{t=1}^T \mathcal{P}_t(p) - \mathbb{I}[p_t = p] \right| \leq \begin{cases} 2 \sqrt{\sum_{t=1}^T \mathcal{P}_t(p) \log \frac{6(N+1)|\mathcal{F}|}{\delta}} & \text{if } p \in \mathcal{I}, \\ \sum_{t=1}^T \mathcal{P}_t(p) + \log \frac{6(N+1)|\mathcal{F}|}{\delta} & \text{otherwise.} \end{cases} \tag{11}$$

Our next goal is to bound $\mathsf{SMCal}_{\mathcal{F},2}$ in terms of $\mathsf{PSMCal}_{\mathcal{F},2}$. By definition,

$$\mathsf{SMCal}_{\mathcal{F},2} = \sup_{\{f_p \in \mathcal{F}\}_{p \in \mathcal{Z}}} \sum_{p \in \mathcal{Z}} \left( \sum_{t=1}^T \mathbb{I}[p_t = p] \right) (\rho_{p,f_p})^2 = \sum_{p \in \mathcal{Z}} \left( \sum_{t=1}^T \mathbb{I}[p_t = p] \right) \sup_{f \in \mathcal{F}} \rho_{p,f}^2,$$

$$\mathsf{PSMCal}_{\mathcal{F},2} = \sup_{\{f_p \in \mathcal{F}\}_{p \in \mathcal{Z}}} \sum_{p \in \mathcal{Z}} \left( \sum_{t=1}^T \mathcal{P}_t(p) \right) \tilde{\rho}_{p,f_p}^2 = \sum_{p \in \mathcal{Z}} \left( \sum_{t=1}^T \mathcal{P}_t(p) \right) \sup_{f \in \mathcal{F}} \tilde{\rho}_{p,f}^2.$$

Therefore,

$$\mathsf{SMCal}_{\mathcal{F},2} \leq 2 \sum_{p \in \mathcal{Z}} \left( \sum_{t=1}^{T} \mathbb{I}[p_t = p] \right) \left( \sup_{f \in \mathcal{F}} (\rho_{p,f} - \tilde{\rho}_{p,f})^2 + \sup_{f \in \mathcal{F}} \tilde{\rho}_{p,f}^2 \right),$$

where the inequality follows by the sub-additivity of the supremum function and since $(u + v)^2 \leq 2u^2 + 2v^2$. Equipped with (10) and (11), it is easy to express $\mathsf{SMCal}_{\mathcal{F},2}$ in terms of $\mathsf{PSMCal}_{\mathcal{F},2}$. To achieve so, we define two terms $\mathrm{TERM}_1, \mathrm{TERM}_2$ as

$$\mathrm{TERM}_1 := \sum_{p \in \mathcal{I}} \left( \sum_{t=1}^{T} \mathbb{I}[p_t = p] \right) \left( \sup_{f \in \mathcal{F}} (\rho_{p,f} - \tilde{\rho}_{p,f})^2 + \sup_{f \in \mathcal{F}} \tilde{\rho}_{p,f}^2 \right),$$

$$\mathrm{TERM}_2 := \sum_{p \in \bar{\mathcal{I}}} \left( \sum_{t=1}^{T} \mathbb{I}[p_t = p] \right) \left( \sup_{f \in \mathcal{F}} (\rho_{p,f} - \tilde{\rho}_{p,f})^2 + \sup_{f \in \mathcal{F}} \tilde{\rho}_{p,f}^2 \right)$$

and bound $\mathrm{TERM}_1, \mathrm{TERM}_2$ separately. We begin by bounding $\mathrm{TERM}_1$ as

$$\mathrm{TERM}_1 \leq \sum_{p \in \mathcal{I}} \left( \sum_{t=1}^{T} \mathcal{P}_t(p) + 2 \sqrt{\left( \sum_{t=1}^{T} \mathcal{P}_t(p) \right) \log \frac{6(N+1)|\mathcal{F}|}{\delta}} \right) \left( \sup_{f \in \mathcal{F}} (\rho_{p,f} - \tilde{\rho}_{p,f})^2 + \sup_{f \in \mathcal{F}} \tilde{\rho}_{p,f}^2 \right)$$

$$\leq \sum_{p \in \mathcal{I}} \left( \sum_{t=1}^{T} \mathcal{P}_t(p) + 2 \sqrt{\left( \sum_{t=1}^{T} \mathcal{P}_t(p) \right) \log \frac{6(N+1)|\mathcal{F}|}{\delta}} \right) \left( \frac{64 \log \frac{6(N+1)|\mathcal{F}|}{\delta}}{\sum_{t=1}^{T} \mathcal{P}_t(p)} + \sup_{f \in \mathcal{F}} \tilde{\rho}_{p,f}^2 \right)$$

$$\leq 3 \sum_{p \in \mathcal{I}} \left( \sum_{t=1}^{T} \mathcal{P}_t(p) \right) \left( \frac{64 \log \frac{6(N+1)|\mathcal{F}|}{\delta}}{\sum_{t=1}^{T} \mathcal{P}_t(p)} + \sup_{f \in \mathcal{F}} \tilde{\rho}_{p,f}^2 \right)$$

$$= 192 |\mathcal{I}| \log \frac{6(N+1)|\mathcal{F}|}{\delta} + 3 \sum_{p \in \mathcal{I}} \left( \sum_{t=1}^{T} \mathcal{P}_t(p) \right) \sup_{f \in \mathcal{F}} \tilde{\rho}_{p,f}^2,$$

where the first inequality follows from (11); the second inequality follows from (10); the third inequality follows since $\log \frac{6(N+1)|\mathcal{F}|}{\delta} \leq \sum_{t=1}^{T} \mathcal{P}_t(p)$ as $p \in \mathcal{I}$. Next, we bound $\mathrm{TERM}_2$ as

$$\mathrm{TERM}_2 \leq \sum_{p \in \bar{\mathcal{I}}} \left( 2 \sum_{t=1}^{T} \mathcal{P}_t(p) + \log \frac{6(N+1)|\mathcal{F}|}{\delta} \right) \left( \sup_{f \in \mathcal{F}} (\rho_{p,f} - \tilde{\rho}_{p,f})^2 + \sup_{f \in \mathcal{F}} \tilde{\rho}_{p,f}^2 \right)$$

$$\leq \sum_{p \in \bar{\mathcal{I}}} \left( 2 \sum_{t=1}^{T} \mathcal{P}_t(p) + \log \frac{6(N+1)|\mathcal{F}|}{\delta} \right) \left( 4 + \sup_{f \in \mathcal{F}} \tilde{\rho}_{p,f}^2 \right)$$

$$\leq 13 |\bar{\mathcal{I}}| \log \frac{6(N+1)|\mathcal{F}|}{\delta} + 2 \sum_{p \in \bar{\mathcal{I}}} \left( \sum_{t=1}^{T} \mathcal{P}_t(p) \right) \sup_{f \in \mathcal{F}} \tilde{\rho}_{p,f}^2,$$

where the first inequality follows from (11); the second inequality follows from (10); the final inequality follows from the definition of $\bar{\mathcal{I}}$ and since $|\tilde{\rho}_{p,f}| \leq 1$. Combining the bounds on $\mathrm{TERM}_1, \mathrm{TERM}_2$ to $\mathsf{SMCal}_{\mathcal{F},2} \leq 2(\mathrm{TERM}_1 + \mathrm{TERM}_2)$, we obtain

$$\mathsf{SMCal}_{\mathcal{F},2} \leq 384 |\mathcal{I}| \log \frac{6(N+1)|\mathcal{F}|}{\delta} + 6 \sum_{p \in \mathcal{I}} \left( \sum_{t=1}^{T} \mathcal{P}_t(p) \right) \sup_{f \in \mathcal{F}} \tilde{\rho}_{p,f}^2 +$$

$$26 |\bar{\mathcal{I}}| \log \frac{6(N+1)|\mathcal{F}|}{\delta} + 4 \sum_{p \in \bar{\mathcal{I}}} \left( \sum_{t=1}^{T} \mathcal{P}_t(p) \right) \sup_{f \in \mathcal{F}} \tilde{\rho}_{p,f}^2$$

$$\leq 384(N+1) \log \frac{6(N+1)|\mathcal{F}|}{\delta} + 6 \sum_{p \in \mathcal{Z}} \left( \sum_{t=1}^{T} \mathcal{P}_t(p) \right) \sup_{f \in \mathcal{F}} \tilde{\rho}_{p,f}^2$$

$$= 384(N+1) \log \frac{6(N+1)|\mathcal{F}|}{\delta} + 6 \cdot \mathsf{PSMCal}_{\mathcal{F},2}.$$

This completes the proof. $\qquad \square$

## B.2 Proof of Lemma 2

*Proof.* To bound $\mathsf{SMCal}_{\mathcal{F}_1^{\mathsf{lin}},2}$ in terms of $\mathsf{SMCal}_{\mathcal{C}_\varepsilon,2}$, we realize that for each $f \in \mathcal{F}_1^{\mathsf{lin}}$, letting $f_\varepsilon \in \mathcal{C}_\varepsilon$ be the representative of $f$, we have

$$|\rho_{p,f} - \rho_{p,f_\varepsilon}| = \left| \frac{\sum_{t=1}^T \mathbb{I}[p_t = p](f(x_t) - f_\varepsilon(x_t))(y_t - p)}{\sum_{t=1}^T \mathbb{I}[p_t = p]} \right| \le \varepsilon.$$

Therefore, $\sup_{f \in \mathcal{F}_1^{\mathsf{lin}}} \rho_{p,f}^2 \le 2 \sup_{f \in \mathcal{C}_\varepsilon} \rho_{p,f}^2 + 2\varepsilon^2$ and

$$\mathsf{SMCal}_{\mathcal{F}_1^{\mathsf{lin}},2} = \sum_{p \in \mathcal{Z}} \left( \sum_{t=1}^T \mathbb{I}[p_t = p] \right) \sup_{f \in \mathcal{F}_1^{\mathsf{lin}}} \rho_{p,f}^2 \le 2 \sum_{p \in \mathcal{Z}} \left( \sum_{t=1}^T \mathbb{I}[p_t = p] \right) \sup_{f \in \mathcal{C}_\varepsilon} \rho_{p,f}^2 + 2\varepsilon^2 T$$

$$= 2\mathsf{SMCal}_{\mathcal{C}_\varepsilon,2} + 2\varepsilon^2 T.$$

Using the result of Lemma 1 to bound $\mathsf{SMCal}_{\mathcal{C}_\varepsilon,2}$, we obtain

$$\mathsf{SMCal}_{\mathcal{F}_1^{\mathsf{lin}},2} \le 768(N+1) \log \frac{6(N+1)|\mathcal{C}_\varepsilon|}{\delta} + 12\mathsf{PSMCal}_{\mathcal{F}_1^{\mathsf{lin}},2} + 2\varepsilon^2 T,$$

where we have also used the inequality $\mathsf{PSMCal}_{\mathcal{C}_\varepsilon,2} \le \mathsf{PSMCal}_{\mathcal{F}_1^{\mathsf{lin}},2}$ since $\mathcal{C}_\varepsilon \subseteq \mathcal{F}_1^{\mathsf{lin}}$. Using Proposition 1 to bound $|\mathcal{C}_\varepsilon|$ finishes the proof. $\qquad\square$

## B.3 Proof of Lemma 3

*Proof.* Consider the function $f'(x) := p + \eta f(x)$, where

$$\eta := \min \left( 1, \frac{\alpha}{\frac{\sum_{t=1}^T \mathcal{P}_t(p)(f(x_t))^2}{\sum_{t=1}^T \mathcal{P}_t(p)}} \right).$$

Under Assumption 1, $f' \in \mathcal{F}$. Furthermore, since $f \in \mathcal{F}_1$, we have $(f'(x))^2 \le 2p^2 + 2\eta^2(f(x))^2 \le 4$, therefore, $f' \in \mathcal{F}_4$. For convinience, we define $\Delta := \sum_{t=1}^T \mathcal{P}_t(p) \left( (p - y_t)^2 - (f'(x_t) - y_t)^2 \right)$. By direct computation, we obtain

$$\Delta = \sum_{t=1}^T \mathcal{P}_t(p) \left( (p - y_t)^2 - (p + \eta f(x_t) - y_t)^2 \right) = \sum_{t=1}^T \mathcal{P}_t(p) \left( -\eta^2 (f(x_t))^2 + 2\eta f(x_t) \cdot (y_t - p) \right).$$

Therefore, the desired quantity can be lower bounded as

$$\frac{\Delta}{\sum_{t=1}^T \mathcal{P}_t(p)} = 2\eta \cdot \tilde{\rho}_{p,f} - \eta^2 \cdot \frac{\sum_{t=1}^T \mathcal{P}_t(p)(f(x_t))^2}{\sum_{t=1}^T \mathcal{P}_t(p)} \ge 2\eta\alpha - \eta^2 \frac{\sum_{t=1}^T \mathcal{P}_t(p)(f(x_t))^2}{\sum_{t=1}^T \mathcal{P}_t(p)} = 2\eta\alpha - \eta^2 \mu,$$

where $\mu := \frac{\sum_{t=1}^T \mathcal{P}_t(p)(f(x_t))^2}{\sum_{t=1}^T \mathcal{P}_t(p)}$. Next, we consider two cases depending on whether or not 1 realizes the minimum in the expression defining $\eta$. If $\alpha \ge \mu$, $\eta = \min(1, \frac{\alpha}{\mu}) = 1$. Therefore, $2\eta\alpha - \eta^2 \mu = 2\alpha - \mu \ge \alpha \ge \alpha^2$, where the last inequality follows since $\tilde{\rho}_{p,f} \le 1$ as $f \in \mathcal{F}_1$, and $\tilde{\rho}_{p,f} \ge \alpha$ by assumption, thus $\alpha \le 1$. Otherwise, if $\alpha < \mu$, we have $\eta = \frac{\alpha}{\mu}$ and $2\eta\alpha - \eta^2 \mu = \frac{\alpha^2}{\mu} \ge \alpha^2$ since $\mu \le 1$ as $f \in \mathcal{F}_1$. Combining both cases, we have shown that $\frac{\Delta}{\sum_{t=1}^T \mathcal{P}_t(p)} \ge \alpha^2$, which completes the proof. $\qquad\square$

## B.4 Proof of Lemma 4

*Proof.* We shall prove the desired result by contradiction. Assume that $\mathsf{PSMCal}_{\mathcal{F}_1,2} > \alpha$. Therefore, there exists a comparator profile $\{f_p \in \mathcal{F}_1\}_{p \in \mathcal{Z}}$ such that

$$\sum_{p \in \mathcal{Z}} \left( \sum_{t=1}^T \mathcal{P}_t(p) \right) \left( \tilde{\rho}_{p,f_p} \right)^2 > \alpha. \tag{12}$$

For each $p \in \mathcal{Z}$, define $\alpha_p := \left(\sum_{t=1}^{T} \mathcal{P}_t(p)\right) \tilde{\rho}_{p,f_p}^2$. Thus, there exists a function $f_p^\star(x)$ which is either $f_p(x)$ or $-f_p(x)$ such that $\tilde{\rho}_{p,f_p^\star} = \sqrt{\frac{\alpha_p}{\sum_{t=1}^{T} \mathcal{P}_t(p)}}$. Clearly, $f_p^\star \in \mathcal{F}_1$. It follows from Lemma 3 that for each $p \in \mathcal{P}$, there exists a $f_p' \in \mathcal{F}_4$ such that

$$\sum_{t=1}^{T} \mathcal{P}_t(p) \left((p - y_t)^2 - (f_p'(x_t) - y_t)^2\right) \geq \alpha_p.$$

Summing over $p \in \mathcal{Z}$, we obtain that the comparator profile $\{f_p' \in \mathcal{F}_4\}_{p \in \mathcal{Z}}$ realizes

$$\sum_{p \in \mathcal{Z}} \sum_{t=1}^{T} \mathcal{P}_t(p) \left((p - y_t)^2 - (f_p'(x_t) - y_t)^2\right) \geq \sum_{p \in \mathcal{Z}} \alpha_p > \alpha,$$

where the last inequality follows from (12). This is a contradiction to the assumption that $\mathsf{PSReg}_{\mathcal{F}_4} \leq \alpha$. This completes the proof. $\qquad\square$

### B.5 Deferred algorithms

---
**Algorithm 3** Generic BM algorithmic template

---
**Initialize:** $\mathcal{A}_i$ for $i \in \{0, \ldots, N\}$ and set $q_{1,i} = \left[\frac{1}{N+1}, \ldots, \frac{1}{N+1}\right]$;

1: **for** $t = 1, \ldots, T$,
2:     Receive context $x_t$;
3:     Set $Q_t = [q_{t,0}, \ldots, q_{t,N}]$;
4:     Compute the stationary distribution of $Q_t$, i.e., $p_t \in \Delta_{N+1}$ that satisfies $Q_t p_t = p_t$;
5:     Output conditional distribution $\mathcal{P}_t$, where $\mathcal{P}_t(z_i) = p_t(i)$ and observe $y_t$;
6:     **for** $i = 0, \ldots, N$
7:         Feed the scaled loss function $\phi_{t,i}(w) = p_{t,i}\ell(w, y_t)$ to $\mathcal{A}_i$ (Algorithm 1) and obtain $q_{t+1,i}$.

---

---
**Algorithm 4** Online Newton Step ($\mathsf{ONS}_i$) with scaled losses

---
1: Set $\beta = \frac{1}{640}, \omega = \frac{1}{4\beta^2}$, and initialize $\theta_{1,i} \in 4 \cdot \mathbb{B}_2^d$ arbitrarily;
2: **for** $t = 2, \ldots, T$,
3:     Update $\theta_{t,i}$ as

$$\theta_{t,i} = \Pi_{4 \cdot \mathbb{B}_2^d}^{A_{t-1,i}}\left(\theta_{t-1,i} - \frac{1}{\beta}A_{t-1,i}^{-1}\nabla_{t-1,i}\right),$$

where $\nabla_{\tau,i} = \nabla\phi_{\tau,i}(\theta_{\tau,i}) = 2p_{\tau,i}\left(\langle\theta_{\tau,i}, x_\tau\rangle - y_\tau\right), A_{t-1,i} = \sum_{\tau=1}^{t-1}\nabla_{\tau,i}\nabla_{\tau,i}^\mathsf{T} + \omega I_d$, and $\Pi_{4 \cdot \mathbb{B}_2^d}^{A_{t-1,i}}$ is the projection operator with respect to the norm induced by $A_{t-1,i}$, i.e.,

$$\Pi_{4 \cdot \mathbb{B}_2^d}^{A_{t-1,i}}(\theta) = \operatorname*{argmin}_{\tilde{\theta} \in 4 \cdot \mathbb{B}_2^d}(\theta - \tilde{\theta})^\mathsf{T} A_{t-1,i}(\theta - \tilde{\theta}).$$

---

---
**Algorithm 5** $\mathsf{ALG}_i$

---
1: **for** $t = 1, \ldots, T$,
2:     Obtain the output $\theta_{t,i}$ of $\mathsf{ONS}_i$ and predict $w_{t,i} = \mathsf{Proj}_{[0,1]}(\langle\theta_{t,i}, x_t\rangle)$.

---

**Proposition 2.** *For Algorithm 3, we have* $\mathsf{PSReg}_{\mathcal{F}} \leq \sum_{i=0}^{N} \mathsf{Reg}_i(\mathcal{F})$.

*Proof.* For each $i \in \{0, \ldots, N\}$, fix a $f_i \in \mathcal{F}$. By definition of $\mathsf{Reg}_i(\mathcal{F})$, we have

$$\sum_{t=1}^{T} p_{t,i}\left(\sum_{j=0}^{N} q_{t,i,j}(z_j - y_t)^2\right) - \sum_{t=1}^{T} p_{t,i}(f_i(x_t) - y_t)^2 \leq \mathsf{Reg}_i(\mathcal{F}).$$

Summing the equation above for all $i \in \{0, \ldots, N\}$, we obtain

$$\sum_{t=1}^{T}\sum_{i=0}^{N}\sum_{j=0}^{N} p_{t,i} q_{t,i,j}(z_j - y_t)^2 - \sum_{t=1}^{T}\sum_{i=0}^{N} p_{t,i}(f_i(x_t) - y_t)^2 \leq \sum_{i=0}^{N} \mathsf{Reg}_i(\mathcal{F}). \qquad (13)$$

Simplifying the first term on the left-hand side of the equation above, we have

$$\sum_{t=1}^{T}\sum_{i=0}^{N}\sum_{j=0}^{N} p_{t,i} q_{t,i,j}(z_j - y_t)^2 = \sum_{t=1}^{T}\sum_{i=0}^{N} p_{t,i} \langle \boldsymbol{q}_{t,i}, \boldsymbol{\ell}_t \rangle = \sum_{t=1}^{T} \boldsymbol{p}_t^{\mathsf{T}} \boldsymbol{Q}_t^{\mathsf{T}} \boldsymbol{\ell}_t = \sum_{t=1}^{T} \boldsymbol{p}_t^{\mathsf{T}} \boldsymbol{\ell}_t,$$

where the last equality follows since $\boldsymbol{p}_t$ is chosen such that $\boldsymbol{Q}_t \boldsymbol{p}_t = \boldsymbol{p}_t$. Therefore, (13) simplifies to

$$\sum_{t=1}^{T}\sum_{i=0}^{N} p_{t,i}\left((z_i - y_t)^2 - (f_i(x_t) - y_t)^2\right) \leq \sum_{i=0}^{N} \mathsf{Reg}_i(\mathcal{F}).$$

Taking the supremum over all $f_i$'s completes the proof. $\qquad\square$

### B.6   Expected loss of randomized rounding

**Proposition 3.** *Let $p \in [0, 1]$ and $p^-, p^+ \in \mathcal{Z}$ be neighbouring points in $\mathcal{Z}$ such that $p^- \leq p < p^+$. Let $q$ be the random variable that takes value $p^-$ with probability $\frac{p^+ - p}{p^+ - p^-}$ and $p^+$ with probability $\frac{p - p^-}{p^+ - p^-}$. Then, for all $y \in \{0, 1\}$, we have $\mathbb{E}[\ell(q, y)] - \ell(p, y) \leq \frac{1}{N^2}$.*

*Proof.* Let $\Delta := \mathbb{E}_q[(q - y)^2]$. Substituting $p_i = \frac{i}{N}, p_{i+1} = \frac{i+1}{N}$ and by direct computation, we obtain

$$\Delta = \frac{\frac{i+1}{N} - p}{\frac{1}{N}}\left(\frac{i}{N} - y\right)^2 + \frac{p - \frac{i}{N}}{\frac{1}{N}}\left(\frac{i+1}{N} - y\right)^2$$

$$= \frac{\frac{i}{N} - p}{\frac{1}{N}}\left(\frac{i}{N} - y\right)^2 + \left(\frac{i}{N} - y\right)^2 + \frac{p - \frac{i}{N}}{\frac{1}{N}}\left(\frac{i}{N} - y\right)^2 + \frac{p - \frac{i}{N}}{\frac{1}{N}} \cdot \frac{1}{N^2} + 2\left(p - \frac{i}{N}\right)\left(\frac{i}{N} - y\right)$$

$$= \left(\frac{i}{N} - y\right)^2 + \frac{1}{N}\left(p - \frac{i}{N}\right) + 2\left(p - \frac{i}{N}\right)\left(\frac{i}{N} - y\right)$$

$$= \left(\frac{i}{N} - p\right)^2 + (p - y)^2 + 2\left(\frac{i}{N} - p\right)(p - y) + \frac{1}{N}\left(p - \frac{i}{N}\right) + 2\left(p - \frac{i}{N}\right)\left(\frac{i}{N} - y\right)$$

$$= \left(\frac{i}{N} - p\right)^2 + (p - y)^2 + \frac{1}{N}\left(p - \frac{i}{N}\right) - 2\left(p - \frac{i}{N}\right)^2$$

$$= (p - y)^2 + \frac{1}{N}\left(p - \frac{i}{N}\right) - \left(p - \frac{i}{N}\right)^2$$

$$\leq (p - y)^2 + \frac{1}{N^2},$$

where the inequality follows by dropping the negative term, and since $p < \frac{i+1}{N}$. This completes the proof. $\qquad\square$

### B.7   Exp-concavity parameter of the scaled loss

**Proposition 4.** *Let $x \in \mathbb{B}_2^d, y \in \{0, 1\}$. The function $\phi : 4 \cdot \mathbb{B}_2^d \to \mathbb{R}$ defined as $\phi(\theta) := \alpha(\langle \theta, x \rangle - y)^2$ for some $\alpha \in [0, 1]$ is $\frac{1}{50}$-exp-concave and 10-Lipschitz.*

*Proof.* For a $\gamma > 0$, let $g(\theta) := \exp\left(-\gamma \phi(\theta)\right)$. The first derivative of $g$ is given by

$$\nabla g(\theta) = -2\gamma \alpha(\langle \theta, x \rangle - y) \exp\left(-\gamma \alpha(\langle \theta, x \rangle - y)^2\right) x.$$

Differentiating with respect to $\theta$ again, we obtain

$$\nabla^2 g(\theta) = -2\gamma \alpha \exp\left(-\gamma \alpha(\langle \theta, x \rangle - y)^2\right) \cdot \left(1 - 2\gamma \alpha(\langle \theta, x \rangle - y)^2\right) \cdot x x^{\mathsf{T}}.$$

Choosing $\gamma = \frac{1}{50}$, the expression above simplifies to

$$\nabla^2 g(\theta) = -\frac{\alpha}{25} \exp\left(-\frac{\alpha}{50}(\langle \theta, x \rangle - y)^2\right) \cdot \left(1 - \frac{\alpha}{25}(\langle \theta, x \rangle - y)^2\right) \cdot xx^\mathsf{T} \preceq 0,$$

where the inequality is because $(\langle \theta, x \rangle - y)^2 \leq 25$ since $|\langle \theta, x \rangle| \leq \|\theta\| \|x\| \leq 4$ by the Cauchy-Schwartz inequality, therefore $|\langle \theta, x \rangle - y| \leq 5$; this implies that $\frac{\alpha}{25}(\langle \theta, x \rangle - y)^2 \leq 1$. Hence, the function $\exp\left(-\frac{1}{50}\phi(\theta)\right)$ is concave, thus $\phi$ is $\frac{1}{50}$-exp-concave by definition. To bound the Lipschitzness parameter, we note that since $\nabla\phi(\theta) = 2\alpha(\langle \theta, x \rangle - y)x$, we have $\|\nabla\phi(\theta)\| = 2\alpha |\langle \theta, x \rangle - y| \cdot \|x\| \leq 10$. Therefore, $\phi$ is 10-Lipschitz. This completes the proof. $\qquad\square$

## B.8   Proof of Theorem 1

*Proof.* We have,

$$\mathsf{SMCal}_{\mathcal{F}_1^{\mathsf{lin}},2} = \mathcal{O}\left(N \log \frac{N}{\delta} + Nd \log \frac{1}{\varepsilon} + Nd \log T + \frac{T}{N^2} + \varepsilon^2 T\right)$$

$$= \mathcal{O}\left(N \log \frac{N}{\delta} + Nd \log T + \frac{T}{N^2}\right)$$

$$= \mathcal{O}\left(T^{\frac{1}{3}} d^{\frac{2}{3}} (\log T)^{\frac{2}{3}} + \left(\frac{T}{d \log T}\right)^{\frac{1}{3}} \log \frac{1}{\delta}\right), \tag{14}$$

where the first equality follows from combining the result of Lemma 2 and Lemma 4 with the bound

$$\mathsf{PSReg}_{\mathcal{F}_4^{\mathsf{lin}}} = \mathcal{O}\left(Nd \log T + \frac{T}{N^2}\right);$$

the second equality follows by substituting $\varepsilon = \frac{1}{\sqrt{T}}$; the final equality follows by substituting $N = \left(\frac{T}{d \log T}\right)^{\frac{1}{3}}$. To bound $\mathbb{E}\left[\mathsf{SMCal}_{\mathcal{F}_1^{\mathsf{lin}},2}\right]$, we let $\mathcal{E}$ denote the event in (14). Then,

$$\mathbb{E}\left[\mathsf{SMCal}_{\mathcal{F}_1^{\mathsf{lin}},2}\right] = \mathbb{E}\left[\mathsf{SMCal}_{\mathcal{F}_1^{\mathsf{lin}},2}|\mathcal{E}\right] \cdot \mathbb{P}(\mathcal{E}) + \mathbb{E}\left[\mathsf{SMCal}_{\mathcal{F}_1^{\mathsf{lin}},2}|\bar{\mathcal{E}}\right] \cdot \mathbb{P}(\bar{\mathcal{E}})$$

$$= \mathcal{O}\left(T^{\frac{1}{3}} d^{\frac{2}{3}} (\log T)^{\frac{2}{3}} + \left(\frac{T}{d \log T}\right)^{\frac{1}{3}} \log \frac{1}{\delta} + \delta T\right)$$

$$= \mathcal{O}\left(T^{\frac{1}{3}} d^{\frac{2}{3}} (\log T)^{\frac{2}{3}}\right),$$

where the second equality follows by bounding $\mathbb{E}[\mathsf{SMCal}_{\mathcal{F}_1^{\mathsf{lin}},2}|\mathcal{E}]$ as per (14), $\mathbb{P}(\mathcal{E}) \leq 1$, $\mathbb{P}(\bar{\mathcal{E}}) \leq \delta$, and $\mathbb{E}[\mathsf{SMCal}_{\mathcal{F}_1^{\mathsf{lin}},2}|\bar{\mathcal{E}}] \leq T$ by definition; the second equality follows by choosing $\delta = 1/T$. This completes the proof. $\qquad\square$

**Corollary 1.** *There exists an efficient algorithm that achieves*

$$\mathsf{SMCal}_{\mathcal{F}_1^{\mathsf{lin}},1} = \mathcal{O}\left(T^{\frac{2}{3}} d^{\frac{1}{3}} (\log T)^{\frac{1}{3}} + T^{\frac{2}{3}} (d \log T)^{-\frac{1}{6}} \sqrt{\log \frac{1}{\delta}}\right)$$

*with probability at least $1 - \delta$. Furthermore, $\mathbb{E}\left[\mathsf{SMCal}_{\mathcal{F}_1^{\mathsf{lin}},1}\right] = \mathcal{O}\left(T^{\frac{2}{3}} d^{\frac{1}{3}} (\log T)^{\frac{1}{3}}\right)$, where the expectation is taken over the internal randomness of the algorithm.*

*Proof.* The high probability bound follows since $\mathsf{SMCal}_{\mathcal{F}_1^{\mathsf{lin}},1} \leq \sqrt{T \cdot \mathsf{SMCal}_{\mathcal{F}_1^{\mathsf{lin}},2}}$. The in-expectation bound is because $\mathbb{E}\left[\mathsf{SMCal}_{\mathcal{F}_1^{\mathsf{lin}},1}\right] \leq \sqrt{T \cdot \mathbb{E}\left[\mathsf{SMCal}_{\mathcal{F}_1^{\mathsf{lin}},2}\right]}$ by applying Jensen's inequality. This completes the proof. $\qquad\square$

## C   Bound on Swap Omniprediction and Contextual Swap regret

In this section, we derive substantially improved rates for (a) swap omniprediction for the class of bounded convex Lipschitz loss functions, and (b) contextual swap regret.

## C.1 Bound on swap omniprediction

Let $\mathcal{L}^{\mathsf{cvx}}$ denote the class of bounded (in $[-1,1]$) convex 1-Lipschitz loss functions, i.e., $\mathcal{L}^{\mathsf{cvx}}$ comprises of functions that are convex in $p$ for a fixed $y \in \{0,1\}$, $\ell(p,y) \in [-1,1]$, and $|\partial\ell(p,y)| \leq 1$ for all $p \in [0,1], y \in \{0,1\}$, where the subgradient is taken with respect to $p$. We first state a result that bounds the (swap) omniprediction error in terms of the (swap) multicalibration error. The following result holds for any hypothesis class $\mathcal{F}$; subsequently, we instantiate our result for an appropriate choice of $\mathcal{F}$.

**Lemma 7.** *(Garg et al., 2024, Theorem 4.1) Let $\mathcal{F} \subset [0,1]^{\mathcal{X}}$ be an arbitrary hypothesis class. We have* $\mathsf{SOmni}_{\mathcal{L}^{\mathsf{cvx}},\mathcal{F}} \leq 6 \cdot \mathsf{SMCal}_{\mathcal{F},1}$, $\mathsf{Omni}_{\mathcal{L}^{\mathsf{cvx}},\mathcal{F}} \leq 6 \cdot \mathsf{MCal}_{\mathcal{F},1}$.

Note that Lemma 7 does not immediately apply to the choice $\mathcal{F} = \mathcal{F}_1^{\mathsf{lin}}$ since the hypotheses in $\mathcal{F}_1^{\mathsf{lin}}$ can take negative values. To align with Lemma 7, we consider the hypothesis class $\mathcal{F}^{\mathsf{aff}} = \left\{ f_\theta(x) = \frac{1+\langle\theta,x\rangle}{2}; \theta \in \mathbb{R}^d \right\}$ and its restriction $\mathcal{F}_{\mathsf{res}}^{\mathsf{aff}} = \left\{ f_\theta(x) = \frac{1+\langle\theta,x\rangle}{2}; \|\theta\| \leq 1 \right\}$ instead. $\mathcal{F}^{\mathsf{aff}}$ satisfies Assumption 1 since

$$a f_\theta(x) + b = \frac{1}{2}\left( a + \frac{a\theta_1}{2} + 2b + a \langle x_{2:d}, \theta_{2:d} \rangle \right) = \frac{1}{2}\left( 1 + \langle \theta', x \rangle \right),$$

where $\theta' \in \mathbb{R}^d$ is such that $\theta'_1 = 2a + a\theta_1 + 4b - 2$ and $\theta'_i = a\theta_i$ for all $2 \leq i \leq d$. For this choice of $\mathcal{F}^{\mathsf{aff}}$, $\mathcal{F}_1^{\mathsf{aff}}$ is determined by the set $\Omega$ of all $\theta$'s that satisfy $\left| \frac{1+\langle\theta,x\rangle}{2} \right| \leq 1$ for all $x \in \mathcal{X}$, where recall that $\mathcal{X} = \{ x \in \mathbb{B}_2^d; x_1 = \frac{1}{2} \}$. Clearly, $\mathcal{F}_{\mathsf{res}}^{\mathsf{aff}} \subseteq \mathcal{F}_1^{\mathsf{aff}}$ by the Cauchy-Schwartz inequality. Furthermore, it is easy to verify that $\alpha_1 \cdot \mathbb{B}_2^d \subset \Omega \subset \alpha_2 \cdot \mathbb{B}_2^d$, where $\alpha_1, \alpha_2 = \Theta(1)$. Therefore, the entire analysis in Section 2 can be extended (with only a multiplicative change in the constants, which does not affect the final rate) to bound $\mathsf{SMCal}_{\mathcal{F}_1^{\mathsf{aff}},1}$, and thus $\mathsf{SMCal}_{\mathcal{F}_{\mathsf{res}}^{\mathsf{aff}},1}$, since $\mathcal{F}_{\mathsf{res}}^{\mathsf{aff}} \subseteq \mathcal{F}_1^{\mathsf{aff}}$. Since $\mathcal{F}_{\mathsf{res}}^{\mathsf{aff}} \subset [0,1]^{\mathcal{X}}$, we can finally use Lemma 7 to bound $\mathsf{SOmni}_{\mathcal{L}^{\mathsf{cvx}},\mathcal{F}_{\mathsf{res}}^{\mathsf{aff}}}$. We skip the exact derivations for the sake of brevity; however remark that the above discussion was implicitly skipped by Garg et al. (2024), who used the result of Lemma 7 to bound $\mathsf{SOmni}_{\mathcal{L}^{\mathsf{cvx}},\mathcal{F}_1^{\mathsf{lin}}}$.

Using Corollary 1 to bound $\mathsf{SMCal}_{\mathcal{F}_{\mathsf{res}}^{\mathsf{aff}},1}$ in Lemma 7, we obtain the following theorem.

**Theorem 2.** *There exists an efficient algorithm that achieves*

$$\mathsf{SOmni}_{\mathcal{L}^{\mathsf{cvx}},\mathcal{F}_{\mathsf{res}}^{\mathsf{aff}}} = \mathcal{O}\left( T^{\frac{2}{3}} d^{\frac{1}{3}} (\log T)^{\frac{1}{3}} + T^{\frac{2}{3}} (d \log T)^{-\frac{1}{6}} \sqrt{\log \frac{1}{\delta}} \right)$$

*with probability at least $1-\delta$. Furthermore, $\mathbb{E}\left[ \mathsf{SOmni}_{\mathcal{L}^{\mathsf{cvx}},\mathcal{F}_{\mathsf{res}}^{\mathsf{aff}}} \right] = \mathcal{O}\left( T^{\frac{2}{3}} d^{\frac{1}{3}} (\log T)^{\frac{1}{3}} \right)$, where the expectation is taken over the internal randomness of the algorithm.*

Theorem 2 significantly improves upon the $\tilde{\mathcal{O}}\left( T^{\frac{7}{8}} (d^2 \log \frac{1}{\delta})^{\frac{1}{4}} \right)$ high probability bound of Garg et al. (2024).

## C.2 Bound on the contextual swap regret

In this section, we derive an improved high probability bound on $\mathsf{SReg}_{\mathcal{F}_4^{\mathsf{lin}}}$. Similar to Lemma 1, we first obtain a high probability bound that relates $\mathsf{SReg}_{\mathcal{F}}$ and $\mathsf{PSReg}_{\mathcal{F}}$ for a finite hypothesis class.

**Lemma 8.** *Let $\mathcal{F} \subset [-1,1]^{\mathcal{X}}$ be a finite hypothesis class. For any algorithm $\mathcal{A}_{\mathsf{SReg}_{\mathcal{F}}}$ such that for each $t \in [T]$ the conditional distribution $\mathcal{P}_t$ is deterministic, with probability at least $1-\delta$ over $\mathcal{A}_{\mathsf{SReg}_{\mathcal{F}}}$'s predictions $p_1, \ldots, p_T$, we have*

$$\mathsf{SReg}_{\mathcal{F}} \leq \mathsf{PSReg}_{\mathcal{F}} + 8\sqrt{(N+1)T \log \frac{2(N+1)|\mathcal{F}|}{\delta}} + 8(N+1) \log \frac{2(N+1)|\mathcal{F}|}{\delta}.$$

*Proof.* The proof follows by an application of Freedman's inequality, similar to Lemma 1. Fix a $f \in \mathcal{F}, p \in \mathcal{Z}$ and define the martingale difference sequence $\{X_t\}_{t=1}^{T}$ as

$$X_t := (\mathbb{I}[p_t = p] - \mathcal{P}_t(p)) \cdot \left( (p - y_t)^2 - (f(x_t) - y_t)^2 \right).$$

Since $f \in [-1, 1]$, $|X_t| \leq 4$ for all $t \in [T]$. Fix a $\mu_p \in \left[0, \frac{1}{4}\right]$. Applying Lemma 6, we obtain

$$\left|\sum_{t=1}^{T} (\mathbb{I}[p_t = p] - \mathcal{P}_t(p)) \cdot \left((p - y_t)^2 - (f(x_t) - y_t)^2\right)\right| \leq 16\mu_p \sum_{t=1}^{T} \mathcal{P}_t(p) + \frac{1}{\mu_p} \log \frac{2}{\delta},$$

where the inequality is because the total conditional variance can be bounded as

$$\sum_{t=1}^{T} \mathbb{E}_t \left[X_t^2\right] = \sum_{t=1}^{T} \mathbb{E}_t \left[(\mathbb{I}[p_t = p] - \mathcal{P}_t(p))^2\right] \cdot \left((p - y_t)^2 - (f(x_t) - y_t)^2\right)^2$$

$$\leq 16 \sum_{t=1}^{T} \mathcal{P}_t(p)(1 - \mathcal{P}_t(p)),$$

and we drop the negative term. Taking a union bound over all $p \in \mathcal{Z}, f \in \mathcal{F}$, we obtain that with probability at least $1 - \delta$ (simultaneously over all $p \in \mathcal{Z}, f \in \mathcal{F}$),

$$\left|\sum_{t=1}^{T} (\mathbb{I}[p_t = p] - \mathcal{P}_t(p)) \cdot \left((p - y_t)^2 - (f(x_t) - y_t)^2\right)\right| \leq 16\mu_p \sum_{t=1}^{T} \mathcal{P}_t(p) + \frac{1}{\mu_p} \log \frac{2(N+1)|\mathcal{F}|}{\delta}.$$

Next, we minimize the bound above with respect to $\mu_p$. If $p \in \mathcal{Z}$ is such that $\sum_{t=1}^{T} \mathcal{P}_t(p) \geq \log \frac{2(N+1)|\mathcal{F}|}{\delta}$, the optimal choice of $\mu_p$ is $\frac{1}{4} \sqrt{\frac{\log \frac{2(N+1)}{\delta}}{\sum_{t=1}^{T} \mathcal{P}_t(p)}}$; otherwise, the optimal $\mu_p$ is $\frac{1}{4}$. Therefore, we define the set

$$\mathcal{I} := \left\{p \in \mathcal{Z} \text{ s.t } \sum_{t=1}^{T} \mathcal{P}_t(p) \geq \log \frac{2(N+1)|\mathcal{F}|}{\delta}\right\}$$

and bound the deviation as

$$\left|\sum_{t=1}^{T} (\mathbb{I}[p_t = p] - \mathcal{P}_t(p)) \cdot \left((p - y_t)^2 - (f(x_t) - y_t)^2\right)\right| \leq$$

$$\begin{cases} 8\sqrt{\left(\sum_{t=1}^{T} \mathcal{P}_t(p)\right) \log \frac{2(N+1)|\mathcal{F}|}{\delta}} & \text{if } p \in \mathcal{I}, \\ 4\left(\sum_{t=1}^{T} \mathcal{P}_t(p) + \log \frac{2(N+1)|\mathcal{F}|}{\delta}\right) & \text{otherwise.} \end{cases} \tag{15}$$

Equipped with (15), we can bound $\mathsf{SReg}_{\mathcal{F}}$ in the following manner:

$\mathsf{SReg}_{\mathcal{F}} =$

$$\sup_{\{f_p \in \mathcal{F}\}_{p \in \mathcal{Z}}} \sum_{p \in \mathcal{Z}} \sum_{t=1}^{T} \mathbb{I}[p_t = p] \left((p - y_t)^2 - (f_p(x_t) - y_t)^2\right)$$

$$= \sum_{p \in \mathcal{Z}} \sup_{f \in \mathcal{F}} \sum_{t=1}^{T} \mathbb{I}[p_t = p] \left((p - y_t)^2 - (f(x_t) - y_t)^2\right)$$

$$\leq \sum_{p \in \mathcal{Z}} \sup_{f \in \mathcal{F}} \sum_{t=1}^{T} (\mathbb{I}[p_t = p] - \mathcal{P}_t(p)) \cdot \left((p - y_t)^2 - (f(x_t) - y_t)^2\right) + \mathsf{PSReg}_{\mathcal{F}}$$

$$\leq 8 \sum_{p \in \mathcal{I}} \sqrt{\left(\sum_{t=1}^{T} \mathcal{P}_t(p)\right) \log \frac{2(N+1)|\mathcal{F}|}{\delta}} + 4 \sum_{p \in \bar{\mathcal{I}}} \left(\sum_{t=1}^{T} \mathcal{P}_t(p) + \log \frac{2(N+1)|\mathcal{F}|}{\delta}\right) + \mathsf{PSReg}_{\mathcal{F}}$$

$$\leq 8 \sum_{p \in \mathcal{I}} \sqrt{\left(\sum_{t=1}^{T} \mathcal{P}_t(p)\right) \log \frac{2(N+1)|\mathcal{F}|}{\delta}} + 8 |\bar{\mathcal{I}}| \log \frac{2(N+1)|\mathcal{F}|}{\delta} + \mathsf{PSReg}_{\mathcal{F}}$$

$$\leq 8 \sqrt{|\mathcal{I}| \left(\sum_{p \in \mathcal{I}} \sum_{t=1}^{T} \mathcal{P}_t(p)\right) \log \frac{2(N+1)|\mathcal{F}|}{\delta}} + 8 |\bar{\mathcal{I}}| \log \frac{2(N+1)|\mathcal{F}|}{\delta} + \mathsf{PSReg}_{\mathcal{F}}$$

$$\leq 8\sqrt{(N+1)T\log\frac{2(N+1)\,|\mathcal{F}|}{\delta}} + 8(N+1)\log\frac{2(N+1)\,|\mathcal{F}|}{\delta} + \mathsf{PSReg}_{\mathcal{F}},$$

where the first inequality follows from the sub-additivity of the supremum function, and by a similar reasoning as the first two equalities above, we have

$$\mathsf{PSReg}_{\mathcal{F}} = \sum_{p\in\mathcal{Z}}\sup_{f\in\mathcal{F}}\sum_{t=1}^{T}\mathcal{P}_t(p)\left((p-y_t)^2-(f(x_t)-y_t)^2\right);$$

the second inequality follows from (15); the third inequality follows since $\log\frac{2(N+1)|\mathcal{F}|}{\delta} > \sum_{t=1}^{T}\mathcal{P}_t(p)$ for all $p\in\bar{\mathcal{I}}$; the fourth inequality follows from the Cauchy-Schwartz inequality. This completes the proof. $\square$

Equipped with Lemma 8 and by a covering number-based argument, we bound $\mathsf{SReg}_{\mathcal{F}_4^{\mathsf{lin}}}$ in the following theorem.

**Theorem 3.** *There exists an efficient algorithm that achieves*

$$\mathsf{SReg}_{\mathcal{F}_4^{\mathsf{lin}}} = \mathcal{O}\left(T^{\frac{3}{5}}(d\log T)^{\frac{2}{5}} + T^{\frac{3}{5}}(d\log T)^{-\frac{1}{10}}\sqrt{\log\frac{1}{\delta}}\right)$$

*with probability at least $1-\delta$. Furthermore, $\mathbb{E}\left[\mathsf{SReg}_{\mathcal{F}_4^{\mathsf{lin}}}\right] = \mathcal{O}\left(T^{\frac{3}{5}}(d\log T)^{\frac{2}{5}}\right)$, where the expectation is taken over the internal randomness in the algorithm.*

*Proof.* First, we bound $\mathsf{SReg}_{\mathcal{F}_4^{\mathsf{lin}}}$ in terms of $\mathsf{SReg}_{\mathcal{S}_\varepsilon}$, where $\mathcal{S}_\varepsilon\subseteq\mathcal{F}_4^{\mathsf{lin}}$ is an $\varepsilon$-cover of $\mathcal{F}_4^{\mathsf{lin}}$. Let $f\in\mathcal{F}_4^{\mathsf{lin}}$ and $f_\varepsilon\in\mathcal{S}_\varepsilon$ be its representative. Then, for any $t\in[T]$, we have

$$(f_\varepsilon(x_t)-y_t)^2-(f(x_t)-y_t)^2 = (f_\varepsilon(x_t)-f(x_t))(f_\varepsilon(x_t)+f(x_t)-2y_t) \leq 6\varepsilon.$$

Therefore, $\mathsf{SReg}_{\mathcal{F}_4^{\mathsf{lin}}}$ can be bounded in terms of $\mathsf{SReg}_{\mathcal{S}_\varepsilon}$ as

$$\mathsf{SReg}_{\mathcal{F}_4^{\mathsf{lin}}} = \sum_{p\in\mathcal{Z}}\sup_{f\in\mathcal{F}_4^{\mathsf{lin}}}\sum_{t=1}^{T}\mathbb{I}[p_t=p]\left((p-y_t)^2-(f(x_t)-y_t)^2\right) \leq \mathsf{SReg}_{\mathcal{S}_\varepsilon} + 6\varepsilon T.$$

Since $|\mathcal{S}_\varepsilon|$ is finite (Proposition 1), using Lemma 8 to bound $\mathsf{SReg}_{\mathcal{S}_\varepsilon}$, we obtain

$$\mathsf{SReg}_{\mathcal{F}_4^{\mathsf{lin}}} \leq \mathsf{PSReg}_{\mathcal{S}_\varepsilon} + 8\sqrt{(N+1)T\log\frac{2(N+1)\,|\mathcal{S}_\varepsilon|}{\delta}} + 8(N+1)\log\frac{2(N+1)\,|\mathcal{S}_\varepsilon|}{\delta} + 6\varepsilon T$$

$$= \mathcal{O}\left(\mathsf{PSReg}_{\mathcal{F}_4^{\mathsf{lin}}} + \sqrt{NT\log\frac{N\,|\mathcal{S}_\varepsilon|}{\delta}} + N\log\frac{N\,|\mathcal{S}_\varepsilon|}{\delta} + \varepsilon T\right)$$

$$= \mathcal{O}\left(\frac{T}{N^2} + Nd\log T + \sqrt{NT\log\frac{N}{\delta}} + \sqrt{NdT\log T}\right)$$

$$= \mathcal{O}\left(T^{\frac{3}{5}}(d\log T)^{\frac{2}{5}} + T^{\frac{3}{5}}(d\log T)^{-\frac{1}{10}}\sqrt{\log\frac{1}{\delta}}\right)$$

with probability at least $1-\delta$. The first equality above follows since $\mathsf{PSReg}_{\mathcal{S}_\varepsilon}\leq\mathsf{PSReg}_{\mathcal{F}_4^{\mathsf{lin}}}$; the second equality follows by substituting $\varepsilon=\frac{1}{T}$ and bounding $|\mathcal{S}_\varepsilon|$ as per Proposition 1, dropping the lower order terms, and since $\mathsf{PSReg}_{\mathcal{F}_4^{\mathsf{lin}}} = \mathcal{O}\left(\frac{T}{N^2}+Nd\log T\right)$; the final equality follows by substituting $N=\left(\frac{T}{d\log T}\right)^{\frac{1}{5}}$. The in-expectation bound follows by repeating the exact same steps to bound $\mathbb{E}\left[\mathsf{SMCal}_{\mathcal{F}_1^{\mathsf{lin}},2}\right]$ in the proof of Theorem 1. This completes the proof. $\square$

For $\mathsf{SReg}_{\mathcal{F}_4^{\mathsf{lin}}}$, Garg et al. (2024) proposed an algorithm that achieves $\mathsf{SReg}_{\mathcal{F}_4^{\mathsf{lin}}} = \tilde{\mathcal{O}}\left(dT^{\frac{3}{4}}\sqrt{\log\frac{1}{\delta}}\right)$. Clearly, our result in Theorem 3 is strictly better, with an improved dependence in both $d, T$.

**Remark 1.** *Note that in Theorem 1 we set $N = \left(\frac{T}{d \log T}\right)^{\frac{1}{3}}$, which is different from the choice of $N$ in Theorem 3. Substituting the former value yields a leading dependence of $\tilde{\mathcal{O}}(T^{\frac{2}{3}})$ in the bound on $\mathsf{SReg}_{\mathcal{F}_4^{\mathsf{lin}}}$. Therefore, even if not the best achievable bound on $\mathsf{SReg}_{\mathcal{F}_4^{\mathsf{lin}}}$, the algorithm guaranteed by Theorem 1 achieves an improved dependence on $T$ compared to Garg et al. (2024)'s result.*

# D   From Online to Distributional

In this section, using our improved guarantees for contextual swap regret (Theorem 3), swap multicalibration (Theorem 1), and swap omniprediction (Theorem 2), we establish significantly improved sample complexity bounds for the corresponding distributional quantities. For swap omniprediction and swap agnostic learning, we shall perform an online-to-batch reduction using the corresponding online algorithm that achieves the improved guarantee. The sample complexity bound for swap multicalibration shall follow from that of swap agnostic learning. Before proceeding to the details, we first give formal definitions of the above notions in the distributional setting.

**Distributional (Swap) Multicalibration.**   For a bounded hypothesis class $\mathcal{F}$, the predictor $p$ is perfectly multicalibrated if $\sup_{f \in \mathcal{F}} \mathbb{E}[f(x) \cdot (y - v) \,|\, p(x) = v] = 0$ for each $v \in \mathsf{Range}(p)$. Since perfect multicalibration is both information theoretically and computationally infeasible, the above requirement is quantified via the objective of minimizing the multicalibration error, where the predictor $p$ has $\ell_q$-multicalibration error ($q \geq 1$) at most $\varepsilon$ if $\mathsf{DMCal}_{\mathcal{F},q} := \sup_{f \in \mathcal{F}} \mathbb{E}_v \left[|\mathbb{E}_{\mathcal{D}}\left[f(x) \cdot (y - v)|p(x) = v\right]|^q\right]$ satisfies $\mathsf{DMCal}_{\mathcal{F},q} \leq \varepsilon$. Motivated by the role of swap regret in online learning and to explore the interplay between multicalibration and omniprediction, (Gopalan et al., 2023b) introduced the notion of swap multicalibration, where the predictor $p$ has $\ell_q$-swap multicalibration error at most $\varepsilon$ if $\mathsf{DSMCal}_{\mathcal{F},q} := \mathbb{E}_v \left[\sup_{f \in \mathcal{F}} |\mathbb{E}_{\mathcal{D}}\left[f(x) \cdot (y - v) \,|\, p(x) = v\right]|^q\right] \leq \varepsilon$. Since $\mathsf{DMCal}_{\mathcal{F},q} \leq \mathsf{DSMCal}_{\mathcal{F},q}$, a swap-mutlicalibrated predictor is also multicalibrated.

**Distributional (Swap) Omniprediction.**   A predictor $p$ such that $\mathsf{DOmni}_{\mathcal{L},\mathcal{F}} := \sup_{\ell \in \mathcal{L}} \sup_{f \in \mathcal{F}} \mathbb{E}[\ell(k_\ell(p(x)), y) - \ell(f(x), y)] \leq \varepsilon$ is referred to as a $(\varepsilon, \mathcal{L}, \mathcal{F})$-omnipredictor. In a similar spirit to swap multicalibration, Gopalan et al. (2023b) introduced the notion of swap omniprediction, where the predictor is required to outperform the best hypothesis in $\mathcal{F}$ not just marginally but also when conditioned on the level sets of the predictor, even when the losses are indexed by the predictions themselves. In particular, the predictor $p$ has swap omniprediction error at most $\varepsilon$ if

$$\mathsf{DSOmni}_{\mathcal{L},\mathcal{F}} := \sup_{\{\ell_v \in \mathcal{L}, f_v \in \mathcal{F}\}_{v \in \mathcal{Z}}} \mathbb{E}_{v \sim \mathcal{D}_p}\left[\mathbb{E}\left[\ell_v(k_{\ell_v}(v), y) - \ell_v(f_v(x), y)|p(x) = v\right]\right] \leq \varepsilon. \quad (16)$$

Notably, omniprediction corresponds to a special case of swap omniprediction when the loss, comparator profiles are fixed and independent of $p \in \mathcal{Z}$. Therefore, we have the trivial relation $\mathsf{DOmni}_{\mathcal{L},\mathcal{F}} \leq \mathsf{DSOmni}_{\mathcal{L},\mathcal{F}}$.

**Swap Agnostic Learning.**   Swap agnostic learning is a special case of swap omniprediction when $\mathcal{L} = \{\ell\}$ and $\ell = (p - y)^2$, so that $k_\ell(p) = p$. We define the swap agnostic error as

$$\mathsf{SAErr}_{\mathcal{F}} := \sup_{\{f_v \in \mathcal{F}\}_{v \in \mathcal{Z}}} \mathbb{E}\left[(p(x) - y)^2 - (f_{p(x)}(x) - y)^2\right]. \quad (17)$$

## D.1   Sample complexity of swap omniprediction

We first derive the sample complexity of learning a $(\varepsilon, \mathcal{L}^{\mathsf{cvx}}, \mathcal{F}_{\mathsf{res}}^{\mathsf{aff}})$-swap omnipredictor. As already mentioned, we perform an online-to-batch reduction using our online algorithm in Theorem 2, which we refer to as $\mathcal{A}_{\mathsf{swap}}$ for brevity. The reduction proceeds in the following manner: given $T$ samples $(x_1, y_1), \ldots, (x_T, y_T)$ sampled i.i.d from $\mathcal{D}$, we feed the samples to $\mathcal{A}_{\mathsf{swap}}$ to obtain predictors $p_1, \ldots, p_T$, where $p_t : \mathcal{X} \to \mathcal{Z}$ for each $t \in [T]$. Subsequently, we sample a predictor $p$ from the uniform distribution $\pi$ over $p_1, \ldots, p_T$. To obtain the number of samples $T$ sufficient to drive the swap omniprediction error to be at most $\varepsilon$, we shall derive a concentration bound (tailored to the

choice of $\mathcal{F} = \mathcal{F}_{\text{res}}^{\text{aff}}, \mathcal{L} = \mathcal{L}^{\text{cvx}}$) that relates the distributional version of the swap omniprediction error with its online analogue, i.e., we bound the deviation $\Delta$ defined to be the supremum of

$$\sup_{\{(\ell_v, f_v) \in \mathcal{L} \times \mathcal{F}\}_{v \in \mathcal{Z}}} \left| \mathbb{E}_{(x,y)\sim\mathcal{D},\, p\sim\pi} \left[ \ell_{p(x)}(k_{\ell_{p(x)}}(p(x)), y) - \ell_{p(x)}(f_{p(x)}(x), y) \right] \right.$$

$$\left. - \frac{1}{T} \sum_{t=1}^{T} \left[ \ell_{p_t(x_t)}(k_{\ell_{p_t(x_t)}}(p_t(x_t)), y_t) - \ell_{p_t(x_t)}(f_{p_t(x_t)}(x_t), y_t) \right] \right|.$$

Since $\pi$ is the uniform mixture over $p_1, \ldots, p_T$, we have

$$\mathbb{E}_{(x,y)\sim\mathcal{D},p\sim\pi} \left[ \ell_{p(x)}(k_{\ell_{p(x)}}(p(x)), y) - \ell_{p(x)}(f_{p(x)}(x), y) \right] =$$

$$\frac{1}{T} \sum_{t=1}^{T} \mathbb{E}_{(x,y)\sim\mathcal{D}} \left[ \ell_{p_t(x)}(k_{\ell_{p_t(x)}}(p_t(x)), y) - \ell_{p_t(x)}(f_{p_t(x)}(x), y) \right].$$

Using the Triangle inequality and sub-additivity of the supremum function, we obtain $\Delta \leq \frac{1}{T}(\mathcal{T}_1 + \mathcal{T}_2)$, where $\mathcal{T}_1, \mathcal{T}_2$ are defined as

$$\mathcal{T}_1 := \sup_{\{\ell_v \in \mathcal{L}\}_{v \in \mathcal{Z}}} \left| \sum_{t=1}^{T} \ell_{p_t(x_t)}(k_{\ell_{p_t(x_t)}}(p_t(x_t)), y_t) - \sum_{t=1}^{T} \mathbb{E}_{(x,y)\sim\mathcal{D}} \left[ \ell_{p_t(x)}(k_{\ell_{p_t(x)}}(p_t(x)), y) \right] \right|,$$

$$\mathcal{T}_2 := \sup_{\{(\ell_v, f_v) \in \mathcal{L} \times \mathcal{F}\}_{v \in \mathcal{Z}}} \left| \sum_{t=1}^{T} \ell_{p_t(x_t)}(f_{p_t(x_t)}(x_t), y_t) - \sum_{t=1}^{T} \mathbb{E}_{(x,y)\sim\mathcal{D}} \left[ \ell_{p_t(x)}(f_{p_t(x)}(x), y) \right] \right|.$$

In the next two lemmas, we bound $\mathcal{T}_1, \mathcal{T}_2$.

**Lemma 9.** *For a $\delta \leq \frac{1}{T}$, with probability at least $1 - \delta$, we have $\mathcal{T}_1 = \mathcal{O}\left( \frac{T}{N} + \sqrt{NT \log \frac{N}{\delta}} \right)$.*

*Proof.* We begin by upper bounding $\mathcal{T}_1$ as

$$\mathcal{T}_1 =$$

$$\sup_{\{\ell_v \in \mathcal{L}\}_{v \in \mathcal{Z}}} \left| \sum_{v \in \mathcal{Z}} \sum_{t=1}^{T} \mathbb{I}[p_t(x_t) = v] \cdot \ell_v(k_{\ell_v}(v), y_t) - \mathbb{P}(p_t(x) = v) \cdot \mathbb{E}\left[ \ell_v(k_{\ell_v}(v), y) | p_t(x) = v \right] \right|$$

$$\leq \sum_{v \in \mathcal{Z}} \sup_{\ell \in \mathcal{L}} \left| \sum_{t=1}^{T} \mathbb{I}[p_t(x_t) = v] \cdot \ell(k_\ell(v), y_t) - \mathbb{P}(p_t(x) = v) \cdot \mathbb{E}\left[ \ell(k_\ell(v), y) | p_t(x) = v \right] \right|$$

$$\leq \sum_{v \in \mathcal{Z}} \sup_{\ell \in \mathcal{L}^{\text{proper}}} \left| \sum_{t=1}^{T} \mathbb{I}[p_t(x_t) = v] \cdot \ell(v, y_t) - \mathbb{P}(p_t(x) = v) \cdot \mathbb{E}\left[ \ell(v, y) | p_t(x) = v \right] \right|, \tag{18}$$

where the second inequality follows since the loss $\tilde{\ell}(p, y)$ defined as $\tilde{\ell}(p, y) = \ell(k_\ell(p), y)$ is proper; we replace the $\sup_{\tilde{\ell} \in \mathcal{L}^{\text{proper}}}$ with $\sup_{\ell \in \mathcal{L}^{\text{proper}}}$. It follows from (Kleinberg et al., 2023, Theorem 8) that there exists a basis for proper losses in terms of V-shaped losses $\ell_v(p, y) = (v - y) \cdot \text{sign}(p - v)$, i.e.,

$$\ell(p, y) = \int_0^1 \mu_\ell(v) \cdot \ell_v(p, y) dv,$$

where $\mu_\ell : [0, 1] \to \mathbb{R}_{\geq 0}$ and $\int_0^1 \mu_\ell(v) dv \leq 2$. To avoid overloading the usage of $v$ for both $\ell_v(p, y)$ and $v \in \mathcal{Z}$, we replace the $v$ in (18) with $p$ for all the subsequent steps. Furthermore, as shown in (Okoroafor et al., 2025, Lemma 6.4), for each V-shaped loss $\ell_v$, setting $v' = \frac{1}{N}\lceil Nv \rceil \in \mathcal{Z}$ ensures the following bound for all $p \in \mathcal{Z}, y \in \{0, 1\}$:

$$|\ell_v(p, y) - \ell_{v'}(p, y)| = |(v - y) \cdot \text{sign}(p - v) - (v' - y) \cdot \text{sign}(p - v')|$$

$$= |(v - v') \cdot \text{sign}(p - v)| \leq \frac{1}{N}, \tag{19}$$

where the second equality is because for all $p \in \mathcal{Z}$, we have $\text{sign}(p - v) = \text{sign}(p - v')$. Using this to bound $\mathcal{T}_1$ further, we obtain

$$\mathcal{T}_1 \leq$$

$$\sum_{p \in \mathcal{Z}} \sup_{\ell \in \mathcal{L}^{\text{proper}}} \left| \sum_{t=1}^{T} \mathbb{I}[p_t(x_t) = p] \int_0^1 \mu_\ell(v) \cdot \ell_v(p, y_t) dv - \mathbb{P}(p_t(x) = p) \int_0^1 \mu_\ell(v) \cdot \mathbb{E}[\ell_v(p, y)|p_t(x) = p] dv \right|$$

$$= \sum_{p \in \mathcal{Z}} \sup_{\ell \in \mathcal{L}^{\text{proper}}} \left| \int_0^1 \mu_\ell(v) \left( \sum_{t=1}^{T} \mathbb{I}[p_t(x_t) = p] \cdot \ell_v(p, y_t) - \mathbb{P}(p_t(x) = p) \cdot \mathbb{E}[\ell_v(p, y)|p_t(x) = p] \right) dv \right|$$

$$\leq \sum_{p \in \mathcal{Z}} \sup_{\ell \in \mathcal{L}^{\text{proper}}} \int_0^1 \mu_\ell(v) \left| \sum_{t=1}^{T} \mathbb{I}[p_t(x_t) = p] \cdot \ell_v(p, y_t) - \mathbb{P}(p_t(x) = p) \cdot \mathbb{E}[\ell_v(p, y)|p_t(x) = p] \right| dv.$$

In the next step, we bound the term inside the absolute value. It follows from (19) and the Triangle inequality that the term can be bounded by

$$\frac{1}{N} \sum_{t=1}^{T} \mathbb{I}[p_t(x_t) = p] + \frac{1}{N} \sum_{t=1}^{T} \mathbb{P}(p_t(x) = p) +$$

$$\left| \sum_{t=1}^{T} \mathbb{I}[p_t(x_t) = p] \cdot \ell_{v'}(p, y_t) - \mathbb{P}(p_t(x) = p) \cdot \mathbb{E}[\ell_{v'}(p, y)|p_t(x) = p] \right|.$$

Fix a $v' \in \mathcal{Z}, p \in \mathcal{Z}$. Observe that the sequence $X_1, \ldots, X_T$ defined as

$$X_t := \mathbb{I}[p_t(x_t) = p] \cdot \ell_{v'}(p, y_t) - \mathbb{P}(p_t(x) = p) \cdot \mathbb{E}[\ell_{v'}(p, y)|p_t(x) = p]$$

is a martingale difference sequence with $|X_t| \leq 2$ for all $t \in [T]$. Furthermore, the cumulative conditional variance can be bounded by

$$\sum_{t=1}^{T} \mathbb{E}_t[X_t^2] \leq \sum_{t=1}^{T} \mathbb{P}(p_t(x) = p) \cdot (\mathbb{E}[\ell_{v'}(p, y)|p_t(x)])^2 \leq \sum_{t=1}^{T} \mathbb{P}(p_t(x) = p).$$

Fix a $\mu \in [0, \frac{1}{2}]$. By Lemma 6, we have

$$\left| \sum_{t=1}^{T} X_t \right| \leq \mu \sum_{t=1}^{T} \mathbb{P}(p_t(x) = p) + \frac{1}{\mu} \log \frac{2}{\delta}.$$

Since $\sum_{t=1}^{T} \mathbb{P}(p_t(x) = p)$ is a random variable, the optimal $\mu = \frac{1}{2} \min \left( 1, \sqrt{\frac{\log \frac{2}{\delta}}{\sum_{t=1}^{T} \mathbb{P}(p_t(x)=p)}} \right)$ is also random. Therefore, we cannot merely substitute the optimal $\mu$ (similar to our proofs in Lemmas 1 and 8). However, note that the optimal choice of $\mu \in \left[ \frac{1}{2} \sqrt{\frac{\log \frac{2}{\delta}}{T}}, \frac{1}{2} \right]$. For the subsequent steps, for simplicity in the analysis we assume that there exists a $n \in \mathbb{Z}_{\geq 0}$ such that $\sqrt{\frac{\log \frac{2}{\delta}}{T}} = \frac{1}{2^n}$, and partition the interval $\mathcal{I} = \left[ \frac{1}{2} \sqrt{\frac{\log \frac{2}{\delta}}{T}}, \frac{1}{2} \right]$ as $\mathcal{I} = \mathcal{I}_n \cup \cdots \cup \mathcal{I}_1 \cup \mathcal{I}_0$, where $\mathcal{I}_k = \left[ \frac{1}{2^{k+1}}, \frac{1}{2^k} \right)$ for all $k \in [n]$ and $\mathcal{I}_0 = \{ \frac{1}{2} \}$. Each interval corresponds to a condition on $\sum_{t=1}^{T} \mathbb{P}(p_t(x) = p)$ for which the optimal $\mu$ lies within that interval. In particular, for each $k \in [n]$, the interval $\mathcal{I}_k$ shall correspond to the condition that

$$\frac{1}{2^k} \leq \sqrt{\frac{\log \frac{2(n+1)}{\delta}}{\sum_{t=1}^{T} \mathbb{P}(p_t(x) = p)}} < \frac{1}{2^{k-1}} \equiv 4^{k-1} \log \frac{2(n+1)}{\delta} < \sum_{t=1}^{T} \mathbb{P}(p_t(x) = p) \leq 4^k \log \frac{2(n+1)}{\delta}.$$

(20)

However, $\mathcal{I}_0$ shall represent the condition that $0 \leq \sum_{t=1}^{T} \mathbb{P}(p_t(x) = p) \leq \log \frac{2(n+1)}{\delta}$. For each interval $\mathcal{I}_k$, we associate a parameter $\mu_k = \frac{1}{2^{k+1}}$. Applying Lemma 6 and taking a union bound over

all $k \in \{0, \ldots, n\}$, we obtain

$$\left| \sum_{t=1}^{T} X_t \right| \leq \mu_k \sum_{t=1}^{T} \mathbb{P}(p_t(x) = p) + \frac{1}{\mu_k} \log \frac{2(n+1)}{\delta}$$

with probability at least $1 - \delta$ (simultaneously for all $k$). Next, we prove an uniform upper bound on $\left| \sum_{t=1}^{T} X_t \right|$ by analyzing the bound for each interval. Towards this end, let $k \in [n]$ be such that (20) holds. Then,

$$\left| \sum_{t=1}^{T} X_t \right| \leq \frac{1}{2^{k+1}} \sum_{t=1}^{T} \mathbb{P}(p_t(x) = p) + 2^{k+1} \log \frac{2(n+1)}{\delta} \leq \frac{9}{2} \sqrt{\left( \sum_{t=1}^{T} \mathbb{P}(p_t(x) = p) \right) \log \frac{2(n+1)}{\delta}},$$

where the second inequality follows from (20). Note that the choice of $\mu_k = \frac{1}{2^{k+1}}$ is not necessarily optimal, however, since the optimal $\mu \in \left[ \frac{1}{2^{k+1}}, \frac{1}{2^k} \right)$ for $\mathcal{I}_k$, the bound on $\left| \sum_{t=1}^{T} X_t \right|$ obtained above is only worse than the optimal by a constant factor. Similarly, for $k = 0$, we have

$$\left| \sum_{t=1}^{T} X_t \right| \leq \frac{1}{2} \sum_{t=1}^{T} \mathbb{P}(p_t(x) = p) + 2 \log \frac{2(n+1)}{\delta} \leq \frac{5}{2} \log \frac{2(n+1)}{\delta}.$$

Combining both bounds, we have shown that

$$\left| \sum_{t=1}^{T} X_t \right| \leq \frac{9}{2} \sqrt{\left( \sum_{t=1}^{T} \mathbb{P}(p_t(x) = p) \right) \log \frac{2(n+1)}{\delta}} + \frac{5}{2} \log \frac{2(n+1)}{\delta}$$

$$= \mathcal{O}\left( \sqrt{\left( \sum_{t=1}^{T} \mathbb{P}(p_t(x) = p) \right) \log \frac{1}{\delta}} + \log \frac{1}{\delta} \right),$$

where the second equality follows since $n = \mathcal{O}(\log T)$ and $\log \frac{n+1}{\delta} = \mathcal{O}(\log \frac{1}{\delta})$ since $\delta \leq \frac{1}{T}$. Taking a union bound over all $v' \in \mathcal{Z}, p \in \mathcal{Z}$, and substituting back to the bound on $\mathcal{T}_1$, we obtain

$$\mathcal{T}_1 \leq$$

$$2 \sum_{p \in \mathcal{Z}} \frac{1}{N} \sum_{t=1}^{T} \mathbb{I}[p_t(x_t) = p] + \frac{1}{N} \sum_{t=1}^{T} \mathbb{P}(p_t(x) = p) + \mathcal{O}\left( \sqrt{\sum_{t=1}^{T} \mathbb{P}(p_t(x) = p) \log \frac{N}{\delta}} + \log \frac{N}{\delta} \right)$$

$$= \mathcal{O}\left( \frac{T}{N} + \sqrt{N \sum_{p \in \mathcal{Z}} \sum_{t=1}^{T} \mathbb{P}(p_t(x) = p) \log \frac{N}{\delta} + N \log \frac{N}{\delta}} \right) = \mathcal{O}\left( \frac{T}{N} + \sqrt{NT \log \frac{N}{\delta}} \right),$$

where the first equality follows from the Cauchy-Schwartz inequality. This completes the proof. $\square$

For $\mathcal{T}_2$, we specifically tailor our analysis to the case $\mathcal{L} = \mathcal{L}^{\mathsf{cvx}}$ and $\mathcal{F} = \mathcal{F}_{\mathsf{res}}^{\mathsf{aff}}$. Since $\mathcal{L}^{\mathsf{cvx}}$ is infinite, we cannot merely apply Freedman's inequality and take a union bound over all $\ell \in \mathcal{L}$. Furthermore, $\mathcal{L}^{\mathsf{cvx}}$ does not have a finite-sized cover with respect to the $\ell_\infty$ metric (Guntuboyina and Sen, 2012), thereby rendering our covering number-based arguments futile. However, a recent result (Lemma 10) due to Gopalan et al. (2024) gives a tight (up to logarithmic terms) bound on the approximate rank of convex functions. Therefore, to obtain a high probability bound for $\mathcal{T}_2$, we shall express the supremum over $\ell \in \mathcal{L}^{\mathsf{cvx}}$ in terms of a supremum over the elements of the basis and subsequently apply Freedman's inequality to bound the latter quantity.

We first define a notion of approximate basis, following Gopalan et al. (2023b); Okoroafor et al. (2025).

**Definition 1** (Okoroafor et al. (2025)). *Let $\Gamma$ be a set and $\mathcal{F} \subseteq [-1, 1]^\Gamma$. A set $\mathcal{G} \subset [-1, 1]^\Gamma$ is an $\varepsilon > 0$ approximate basis for $\mathcal{F}$ with sparsity $s$ and coefficient norm $\lambda$, if for every $h \in \mathcal{F}$, there exists a finite subset $\{g_1, \ldots, g_s\} \subseteq \mathcal{G}$ and coefficients $c_1, \ldots, c_s \in [-1, 1]$ satisfying*

$$\left| h(x) - \sum_{i=1}^{s} c_i g_i(x) \right| \leq \varepsilon \text{ for all } x \in \Gamma \text{ and } \sum_{i=1}^{s} |c_i| \leq \lambda.$$

In the special case when $\mathcal{G}$ itself has $s$ elements, we say $\mathcal{G}$ is a finite $\varepsilon$-basis for $\mathcal{F}$ of size $s$ with coefficient norm $\lambda$.

**Lemma 10** (Gopalan et al. (2024)). *For all $\varepsilon > 0$, $\mathcal{L}^{\mathsf{cvx}}$ admits a finite $\varepsilon$-basis of size $\mathcal{O}\left(\frac{\log^{\frac{4}{3}}(\frac{1}{\varepsilon})}{\varepsilon^{\frac{2}{3}}}\right)$ with coefficient norm $2$.*

In the following result, we bound $\mathcal{T}_2$ when $\mathcal{F}$ is finite. Subsequently, we bound $\mathcal{T}_2$ for $\mathcal{F}_{\mathsf{res}}^{\mathsf{aff}}$ by a covering number-based argument.

**Lemma 11.** *Let $\mathcal{F} \subset [-1,1]^{\mathcal{X}}$ be a finite hypothesis class. For a $\delta \leq \frac{1}{T}$, with probability at least $1 - \delta$, we have $\mathcal{T}_2 = \mathcal{O}\left(\frac{T}{N} + \sqrt{NT \log \frac{N|\mathcal{F}|}{\delta}}\right)$.*

*Proof.* Observe that $\ell(p, y)$ can be written as $\ell(p, y) = (1 - y) \cdot \ell(p, 0) + y \cdot \ell(p, 1)$. We begin by bounding $\mathcal{T}_2$ as

$$\mathcal{T}_2 =$$

$$\sup_{\{(\ell_v, f_v) \in \mathcal{L}^{\mathsf{cvx}} \times \mathcal{F}\}_{v \in \mathcal{Z}}} \left| \sum_{p \in \mathcal{Z}} \sum_{t=1}^{T} \mathbb{I}[p_t(x_t) = v] \cdot \ell_v(f_v(x_t), y_t) - \mathbb{P}(p_t(x) = v) \cdot \mathbb{E}[\ell_v(f_v(x), y)|p_t(x) = v] \right|$$

$$\leq \sum_{p \in \mathcal{Z}} \sup_{(\ell, f) \in \mathcal{L}^{\mathsf{cvx}} \times \mathcal{F}} \left| \sum_{t=1}^{T} \mathbb{I}[p_t(x_t) = v] \cdot \ell(f(x_t), y_t) - \mathbb{P}(p_t(x) = v) \cdot \mathbb{E}[\ell(f(x), y)|p_t(x) = v] \right|$$

$$\leq \sum_{p \in \mathcal{Z}} \sup_{(\ell, f) \in \mathcal{L}^{\mathsf{cvx}} \times \mathcal{F}} \left| \sum_{t=1}^{T} (1 - y_t) \cdot \mathbb{I}[p_t(x_t) = v] \cdot \ell(f(x_t), 0) - \mathbb{P}(p_t(x) = v) \cdot \mathbb{E}\left[(1 - y) \cdot \ell(f(x), 0)|p_t(x) = v\right] \right|$$

$$+ \sum_{p \in \mathcal{Z}} \sup_{(\ell, f) \in \mathcal{L}^{\mathsf{cvx}} \times \mathcal{F}} \left| \sum_{t=1}^{T} y_t \cdot \mathbb{I}[p_t(x_t) = v] \cdot \ell(f(x_t), 1) - \mathbb{P}(p_t(x) = v) \cdot \mathbb{E}\left[y \cdot \ell(f(x), 1)|p_t(x) = v\right] \right|.$$

The two terms above can be bounded in an exactly similar manner. For the sake of brevity, we only provide details for bounding the second term. We begin by bounding the absolute value accompanying the second term. Since the function $\ell(p, 1)$ is convex in $p$, it admits a finite $\varepsilon$-basis $\mathcal{G}$ with coefficient norm $2$ (Lemma 10), i.e., there exist functions $g_1, \ldots, g_{|\mathcal{G}|}$ such that for each $\ell \in \mathcal{L}^{\mathsf{cvx}}$ there exist constants $c_1(\ell), \ldots, c_{|\mathcal{G}|}(\ell) \in [-1, 1]$ satisfying $\sum_{i=1}^{|\mathcal{G}|} |c_i(\ell)| \leq 2$ and $\left| \ell(p, 1) - \sum_{i=1}^{|\mathcal{G}|} c_i(\ell) g_i(p) \right| \leq \varepsilon$ for all $p \in [0, 1]$. Bounding $\ell(f(x_t), 1)$ in terms of its basis representation and applying the Triangle inequality, we obtain the following upper bound on the absolute value accompanying the second term

$$\left| \sum_{t=1}^{T} y_t \cdot \mathbb{I}[p_t(x_t) = v] \cdot \left( \sum_{i=1}^{|\mathcal{G}|} c_i(\ell) g_i(f(x_t)) \right) - \mathbb{P}(p_t(x) = v) \cdot \left( \sum_{i=1}^{|\mathcal{G}|} \mathbb{E}\left[y \cdot c_i(\ell) g_i(f(x))|p_t(x) = v\right] \right) \right|$$

$$+ \varepsilon \sum_{t=1}^{T} y_t \cdot \mathbb{I}[p_t(x_t) = v] + \varepsilon \sum_{t=1}^{T} \mathbb{P}(p_t(x) = v)$$

which can be further bounded by

$$\left( \sum_{i=1}^{|\mathcal{G}|} |c_i(\ell)| \right) \sup_{g \in \mathcal{G}} \left| \sum_{t=1}^{T} y_t \cdot \mathbb{I}[p_t(x_t) = v] \cdot g(f(x_t)) - \mathbb{P}(p_t(x) = v) \cdot \mathbb{E}\left[y \cdot g(f(x))|p_t(x) = v\right] \right| +$$

$$\varepsilon \sum_{t=1}^{T} y_t \cdot \mathbb{I}[p_t(x_t) = v] + \varepsilon \sum_{t=1}^{T} \mathbb{P}(p_t(x) = v).$$

Therefore, the following expression upper bounds the second term:

$$\mathcal{O}\left( \sum_{v \in \mathcal{Z}} \sup_{g \in \mathcal{G}, f \in \mathcal{F}} \left| \sum_{t=1}^{T} y_t \cdot \mathbb{I}[p_t(x_t) = v] \cdot g(f(x_t)) - \mathbb{P}(p_t(x) = v) \cdot \mathbb{E}\left[y \cdot g(f(x))|p_t(x) = v\right] \right| + \varepsilon T \right).$$

Fix a $g \in \mathcal{G}, f \in \mathcal{F}, v \in \mathcal{Z}$ and define the martingale difference sequence $X_1, \ldots, X_T$, where

$$X_t := y_t \cdot \mathbb{I}[p_t(x_t) = v] \cdot g(f(x_t)) - \mathbb{P}(p_t(x) = v) \cdot \mathbb{E}[y \cdot g(f(x))|p_t(x) = v].$$

Repeating a similar analysis as in the proof of Lemma 9, we obtain,

$$\left| \sum_{t=1}^{T} X_t \right| = \mathcal{O}\left( \sqrt{\left( \sum_{t=1}^{T} \mathbb{P}(p_t(x) = v) \right) \log \frac{1}{\delta}} + \log \frac{1}{\delta} \right).$$

Taking a union bound over all $g \in \mathcal{G}, f \in \mathcal{F}, v \in \mathcal{Z}$, we obtain that with probability at least $1 - \delta$,

$$\left| \sum_{t=1}^{T} y_t \cdot \mathbb{I}[p_t(x_t) = v] \cdot g(f(x_t)) - \mathbb{P}(p_t(x) = v) \cdot \mathbb{E}[y \cdot g(f(x))|p_t(x) = v] \right| =$$

$$\mathcal{O}\left( \sqrt{\left( \sum_{t=1}^{T} \mathbb{P}(p_t(x) = v) \right) \log \frac{N |\mathcal{F}| |\mathcal{G}|}{\delta}} + \log \frac{N |\mathcal{F}| |\mathcal{G}|}{\delta} \right).$$

Combining everything, we have shown that the second term can be bounded by

$$\mathcal{O}\left( \sum_{p \in \mathcal{Z}} \sqrt{\left( \sum_{t=1}^{T} \mathbb{P}(p_t(x) = v) \right) \log \frac{N |\mathcal{F}| |\mathcal{G}|}{\delta}} + N \log \frac{N |\mathcal{F}| |\mathcal{G}|}{\delta} + \varepsilon T \right) =$$

$$\mathcal{O}\left( \frac{T}{N} + \sqrt{NT \log \frac{N |\mathcal{F}|}{\delta}} \right)$$

on choosing $\varepsilon = \frac{1}{N}$, bounding $|\mathcal{G}|$ as per Lemma 10, and using the Cauchy-Schwartz inequality. Repeating the steps above, we obtain the same bound for the first term. Adding both the bounds yields the desired result. $\square$

Using the result of Lemma 11 and by a covering number-based argument, we bound $\mathcal{T}_2$ for $\mathcal{F} = \mathcal{F}_{\text{res}}^{\text{aff}}$ in the following lemma.

**Lemma 12.** *When $\mathcal{F} = \mathcal{F}_{\text{res}}^{\text{aff}}$, for a $\delta \le \frac{1}{T}$, with probability at least $1 - \delta$, we have $\mathcal{T}_2 = \mathcal{O}\left( \frac{T}{N} + \sqrt{dNT \log \frac{N}{\delta}} \right)$.*

*Proof.* Let $\mathcal{C}_\varepsilon$ be an $\varepsilon$-cover for $\mathcal{F}_{\text{res}}^{\text{aff}}$. For a $f \in \mathcal{F}_{\text{res}}^{\text{aff}}$, let $f_\varepsilon$ be the representative of $f$. By the 1-Lipschitzness of $\ell$, we have

$$|\ell(f(x), y) - \ell(f_\varepsilon(x), y)| \le |f(x) - f_\varepsilon(x)| \le \varepsilon.$$

Therefore, $\mathcal{T}_2$ can be bounded as

$\mathcal{T}_2$

$$\le \sum_{p \in \mathcal{Z}} \sup_{(\ell, f) \in \mathcal{L}^{\text{cvx}} \times \mathcal{F}} \left| \sum_{t=1}^{T} \mathbb{I}[p_t(x_t) = v] \cdot \ell(f(x_t), y_t) - \mathbb{P}(p_t(x) = v) \cdot \mathbb{E}[\ell(f(x), y)|p_t(x) = v] \right|$$

$$\le \sum_{p \in \mathcal{Z}} \sup_{(\ell, f) \in \mathcal{L}^{\text{cvx}} \times \mathcal{C}_\varepsilon} \left| \sum_{t=1}^{T} \mathbb{I}[p_t(x_t) = v] \cdot \ell(f(x_t), y_t) - \mathbb{P}(p_t(x) = v) \cdot \mathbb{E}[\ell(f(x), y)|p_t(x) = v] \right| + 2\varepsilon T$$

$$= \mathcal{O}\left( \frac{T}{N} + \sqrt{NT \log \frac{N |\mathcal{C}_\varepsilon|}{\delta}} + \varepsilon T \right) = \mathcal{O}\left( \frac{T}{N} + \sqrt{dNT \log \frac{N}{\delta}} \right)$$

on choosing $\varepsilon = \frac{1}{N}$ and bounding $|\mathcal{C}_\varepsilon|$ as per Proposition 1. This completes the proof. $\square$

Combining the result of Lemmas 9 and 12, we have the following high probability bound on $\Delta$ — the deviation between the distributional and online swap omniprediction errors: $\Delta = \mathcal{O}\left( \frac{1}{N} + \sqrt{\frac{dN}{T} \log \frac{N}{\delta}} \right)$. Combining this with a bound on the online swap omniprediction error (Theorem 2), we obtain the following theorem, which bounds the swap omniprediction error of the predictor learned via the online-to-batch reduction.

**Theorem 4.** *With probability at least $1-\delta$, the randomized predictor $p$ learned via the online-to-batch reduction satisfies*

$$\mathsf{SOmni}_{\mathcal{L}^{\text{cvx}}, \mathcal{F}^{\text{aff}}_{\text{res}}} = \tilde{\mathcal{O}}\left( \left(\frac{d}{T}\right)^{\frac{1}{3}} + \left(\frac{d}{T}\right)^{\frac{1}{3}} \sqrt{\log \frac{1}{\delta}} \right).$$

*Consequently, $\tilde{\Omega}(d\varepsilon^{-3})$ samples are sufficient for $p$ to achieve a swap omniprediction error at most $\varepsilon$.*

*Proof.* The swap omniprediction error of the predictor can be bounded as

$$\Delta + \frac{\mathsf{SMCal}_{\mathcal{F}^{\text{lin}}_1, 1}}{T} \leq \Delta + \sqrt{\frac{\mathsf{SMCal}_{\mathcal{F}^{\text{lin}}_1, 2}}{T}} = \mathcal{O}\left( \frac{1}{N} + \sqrt{\frac{Nd}{T} \log \frac{N}{\delta}} \right).$$

Setting $N = \left(\frac{T}{d}\right)^{\frac{1}{3}}$, we obtain the desired result. $\qquad\square$

## D.2 Sample complexity of swap agnostic learning

For the squared loss, the result of the previous section already gives a $\tilde{\mathcal{O}}(d\varepsilon^{-3})$ sample complexity for swap agnostic learning. However, in this section, we improve the bound to $\tilde{\mathcal{O}}(d\varepsilon^{-2.5})$. Similar to Section D.1, we perform an online-to-batch reduction using our online algorithm in Theorem 3, which we refer to as $\mathcal{A}_{\text{swap}-\text{ag}}$ for brevity. Given $T$ instances $(x_1, y_1), \ldots, (x_T, y_T)$ that are sampled i.i.d from $\mathcal{D}$, we feed the instances to $\mathcal{A}_{\text{swap}-\text{ag}}$ to obtain predictors $p_1, \ldots, p_T$. Subsequently, the predictor $p$ is sampled from the uniform distribution $\pi$ over $p_1, \ldots, p_T$.

To derive the improved sample complexity rate, we show an improved $\mathcal{O}\left( \frac{1}{N^2} + \sqrt{\frac{dN}{T} \log \frac{N}{\delta}} \right)$ bound on the deviation between the swap agnostic error and the contextual swap regret. Notably, our bound on the deviation is stronger than the $\mathcal{O}\left( \frac{1}{N} + \sqrt{\frac{dN}{T} \log \frac{N}{\delta}} \right)$ bound obtained in Section D.1. Comparing the two bounds, in the latter, we incur a $\frac{1}{N}$ term due to an approximation of proper losses in terms of V-shaped losses $\ell_v(p, y) = (v - y) \cdot \text{sign}(p - v)$, and subsequently, we replace the supremum over $v \in [0, 1]$ with $v' \in \mathcal{Z}$, which incurs an additive $\frac{1}{N}$ error. However, for the squared loss, this step is no longer needed since we can directly apply Freedman's inequality. This enables us to get an improved $\frac{1}{N^2}$ dependence. Finally, on combining this with the $\tilde{\mathcal{O}}(T^{\frac{3}{5}})$ bound on the contextual swap regret (which is better than the $\tilde{\mathcal{O}}(T^{\frac{2}{3}})$ bound on the online swap omniprediction error), we obtain the improvement from $\varepsilon^{-3}$ to $\varepsilon^{-2.5}$, thereby saving a $\varepsilon^{-0.5}$ factor. To formalize the above discussion, we define the deviation as

$$\Delta := \sup_{\{f_v \in \mathcal{F}\}_{v \in \mathcal{Z}}} \left| \mathbb{E}_{(x,y)\sim\mathcal{D}, p\sim\pi} \left[ (p(x) - y)^2 - (f_{p(x)}(x) - y)^2 \right] - \right.$$
$$\left. \frac{1}{T} \sum_{t=1}^{T} \left( (p_t(x_t) - y_t)^2 - (f_{p_t(x_t)}(x_t) - y_t)^2 \right) \right|.$$

Since $p$ is sampled from $\pi$, we have

$$\mathbb{E}_{(x,y)\sim\mathcal{D}, p\sim\pi} \left[ (p(x) - y)^2 - (f_{p(x)}(x) - y)^2 \right] =$$
$$\frac{1}{T} \sum_{t=1}^{T} \mathbb{E}_{(x,y)\sim\mathcal{D}} \left[ (p_t(x) - y)^2 - (f_{p_t(x)}(x) - y)^2 \right].$$

The expectation on the right hand side of the equation above can be rewritten as $\sum_{v \in \mathcal{Z}} \mathbb{P}(p_t(x) = v) \cdot \mathbb{E}\left[ (v - y)^2 - (f_v(x) - y)^2 | p_t(x) = v \right]$. Therefore,

$$\Delta \cdot T = \sup_{\{f_v \in \mathcal{F}\}_{v \in \mathcal{Z}}} \left| \sum_{v \in \mathcal{Z}} \left( \sum_{t=1}^{T} \mathbb{P}(p_t(x) = v) \cdot \mathbb{E}[(v - y)^2 - (f_v(x) - y)^2 | p_t(x) = v] - \right. \right.$$
$$\left. \left. \mathbb{I}[p_t(x_t) = v] \cdot ((v - y_t)^2 - (f_v(x_t) - y_t)^2) \right) \right|$$

$$\leq \sum_{v \in \mathcal{Z}} \sup_{f \in \mathcal{F}} \left| \sum_{t=1}^{T} \mathbb{P}(p_t(x) = v) \cdot \mathbb{E}\left[(v-y)^2 - (f(x)-y)^2 | p_t(x) = v\right] - \right.$$

$$\left. \mathbb{I}[p_t(x_t) = v] \cdot ((v-y_t)^2 - (f(x_t)-y_t)^2) \right|, \tag{21}$$

where the inequality follows by Triangle inequality and sub-additivity of the supremum function. In the next lemma, we bound $\Delta$ when $\mathcal{F}$ is finite. Subsequently, we bound $\Delta$ for $\mathcal{F} = \mathcal{F}_4^{\mathsf{lin}}$ by a covering-number based argument.

**Lemma 13.** *Let $\mathcal{F} \subset [-1,1]^{\mathcal{X}}$ be a finite hypothesis class. For a $\delta \leq \frac{1}{T}$, with probability at least $1 - \delta$, we have $\Delta = \mathcal{O}\left(\sqrt{\frac{N}{T} \log \frac{N|\mathcal{F}|}{\delta}}\right)$.*

*Proof.* Fix a $v \in \mathcal{Z}, f \in \mathcal{F}$, and consider the sequence $X_1, \ldots, X_T$, where $X_t$ is defined as

$$X_t := \mathbb{P}(p_t(x) = v) \cdot \mathbb{E}\left[(v-y)^2 - (f(x)-y)^2 | p_t(x) = v\right] - \\ \mathbb{I}[p_t(x_t) = v] \cdot \left((v-y_t)^2 - (f(x_t)-y_t)^2\right).$$

Observe that the above sequence is a martingale difference sequence. Furthermore, since $f \in [-1,1]$, we have $|X_t| \leq 8$ for all $t \in [T]$. Applying Lemma 6, we obtain $\left|\sum_{t=1}^{T} X_t\right| \leq \mu V_X + \frac{1}{\mu} \log \frac{2}{\delta}$ with probability at least $1 - \delta$, where $\mu \in [0, \frac{1}{8}]$ is fixed and the cumulative conditional variance $V_X$ can be bounded as

$$V_X \leq \sum_{t=1}^{T} \mathbb{P}(p_t(x) = v) \cdot \left(\mathbb{E}\left[(v-y)^2 - (f(x)-y)^2 | p_t(x) = v\right]\right)^2 \leq 16 \sum_{t=1}^{T} \mathbb{P}(p_t(x) = v).$$

Repeating a similar analysis as in the proof of Lemma 9, we obtain

$$\left|\sum_{t=1}^{T} X_t\right| = \mathcal{O}\left(\sqrt{\left(\sum_{t=1}^{T} \mathbb{P}(p_t(x) = v)\right) \log \frac{1}{\delta}} + \log \frac{1}{\delta}\right).$$

Taking a union bound over all $v \in \mathcal{Z}, f \in \mathcal{F}$, we obtain that with probability at least $1 - \delta$

$$\sup_{f \in \mathcal{F}} \left| \sum_{t=1}^{T} \mathbb{P}(p_t(x) = v) \cdot \mathbb{E}\left[(v-y)^2 - (f(x)-y)^2 | p_t(x) = v\right] - \right.$$

$$\left. \mathbb{I}[p_t(x_t) = v] \cdot \left((v-y_t)^2 - (f(x_t)-y_t)^2\right) \right| =$$

$$\mathcal{O}\left(\sqrt{\left(\sum_{t=1}^{T} \mathbb{P}(p_t(x) = v)\right) \log \frac{N|\mathcal{F}|}{\delta}} + \log \frac{N|\mathcal{F}|}{\delta}\right) \tag{22}$$

holds simultaneously for all $v \in \mathcal{Z}$. Therefore, we can finally upper bound $\Delta$ as

$$\Delta = \mathcal{O}\left(\frac{1}{T}\sqrt{N\left(\sum_{t=1}^{T}\sum_{v \in \mathcal{Z}} \mathbb{P}(p_t(x) = v)\right) \log \frac{N|\mathcal{F}|}{\delta}} + \frac{N}{T} \log \frac{N|\mathcal{F}|}{\delta}\right) = \mathcal{O}\left(\sqrt{\frac{N}{T} \log \frac{N|\mathcal{F}|}{\delta}}\right),$$

where the first equality follows from (22) and the Cauchy-Schwartz inequality, while the second equality follows by dropping the lower order term. This completes the proof. $\qquad\square$

Using the result of Lemma 13 and by a covering number-based argument, we bound $\mathcal{T}_2$ for $\mathcal{F} = \mathcal{F}_4^{\mathsf{lin}}$ in the following lemma.

**Lemma 14.** *When $\mathcal{F} = \mathcal{F}_4^{\mathsf{lin}}$, for a $\delta \leq \frac{1}{T}$, with probability at least $1 - \delta$, we have $\Delta = \mathcal{O}\left(\frac{1}{N^2} + \sqrt{\frac{dN}{T} \log \frac{N}{\delta}}\right)$.*

*Proof.* Let $\ell$ be the squared loss and $\mathcal{C}_\varepsilon$ be an $\varepsilon$-cover for $\mathcal{F}_4^{\mathsf{lin}}$. Observe that the squared loss is 6-Lipschitz in $p$ for $p \in [-1, 1]$. For a $f \in \mathcal{F}$, letting $f_\varepsilon$ be the representative of $f$, we have $|\ell(f(x), y) - \ell(f_\varepsilon(x), y)| \leq 6\,|f(x) - f_\varepsilon(x)| \leq 6\varepsilon$ for all $x \in \mathcal{X}, y \in \{0, 1\}$. It then follows from (21) that

$$
\Delta \leq \frac{1}{T} \sum_{v \in \mathcal{Z}} \sup_{f \in \mathcal{F}} \left| \sum_{t=1}^{T} \mathbb{P}(p_t(x) = v) \cdot \mathbb{E}\left[\ell(v, y) - \ell(f(x), y) | p_t(x) = v\right] - \right.
$$
$$
\left. \mathbb{I}[p_t(x) = v] \cdot (\ell(v, y_t) - \ell(f(x_t), y_t)) \right|
$$

$$
\leq \frac{1}{T} \sum_{v \in \mathcal{Z}} \sup_{f \in \mathcal{C}_\varepsilon} \left| \sum_{t=1}^{T} \mathbb{P}(p_t(x) = v) \cdot \mathbb{E}\left[\ell(v, y) - \ell(f(x), y) | p_t(x) = v\right] - \right.
$$
$$
\left. \mathbb{I}[p_t(x) = v] \cdot (\ell(v, y_t) - \ell(f(x_t), y_t)) \right|
$$

$$
+ \frac{6\varepsilon}{T} \sum_{t=1}^{T} \sum_{v \in \mathcal{Z}} \mathbb{P}(p_t(x) = v) + \frac{6\varepsilon}{T} \sum_{t=1}^{T} \sum_{v \in \mathcal{Z}} \mathbb{I}[p_t(x) = v]
$$
$$
= \mathcal{O}\left( \varepsilon + \sqrt{\frac{N}{T} \log \frac{N\,|\mathcal{C}_\varepsilon|}{\delta}} \right) = \mathcal{O}\left( \frac{1}{N^2} + \sqrt{\frac{dN}{T} \log \frac{N}{\delta}} \right),
$$

where the second inequality follows by bounding $\ell(f(x), y)$ in terms of $\ell(f_\varepsilon(x), y)$; the first equality follows from Lemma 13, while the final equality follows by choosing $\varepsilon = \frac{1}{N^2}$ and bounding $|\mathcal{C}_\varepsilon|$ as per Proposition 1. This completes the proof. □

Combining the result of Lemma 14 with the bound on $\mathsf{SReg}_{\mathcal{F}_4^{\mathsf{lin}}}$ (Theorem 3), we bound the expected swap agnostic error incurred by $p$.

**Theorem 5.** *With probability at least $1-\delta$, the randomized predictor $p$ learned via the online-to-batch reduction satisfies*

$$
\mathsf{SAErr}_{\mathcal{F}_4^{\mathsf{lin}}} = \tilde{\mathcal{O}}\left( \left(\frac{d}{T}\right)^{\frac{2}{5}} + \left(\frac{d}{T}\right)^{\frac{2}{5}} \sqrt{\log \frac{1}{\delta}} \right).
$$

*Consequently, $\tilde{\Omega}(d\varepsilon^{-2.5})$ samples are sufficient for $p$ to be achieve a swap agnostic error at most $\varepsilon$.*

*Proof.* The expected swap agnostic error incurred by $p$ can be bounded by

$$
\Delta + \frac{\mathsf{SReg}_{\mathcal{F}_4^{\mathsf{lin}}}}{T} = \mathcal{O}\left( \frac{1}{N^2} + \sqrt{\frac{dN}{T} \log \frac{N}{\delta}} \right) = \tilde{\mathcal{O}}\left( \left(\frac{d}{T}\right)^{\frac{2}{5}} + \left(\frac{d}{T}\right)^{\frac{2}{5}} \sqrt{\log \frac{1}{\delta}} \right),
$$

on choosing $N = \left(\frac{T}{d}\right)^{\frac{1}{5}}$. Note that in the first equality above, we have used the bound

$$
\mathsf{SReg}_{\mathcal{F}_4^{\mathsf{lin}}} = \left( \frac{T}{N^2} + Nd \log T + \sqrt{dNT \log \frac{N}{\delta}} \right)
$$

which follows from Theorem 3. This completes the proof. □

### D.3 Sample complexity of swap multicalibration

In this section, we establish $\tilde{\mathcal{O}}(\varepsilon^{-5}), \tilde{\mathcal{O}}(\varepsilon^{-2.5})$ sample complexities for learning $\ell_1, \ell_2$-swap multicalibrated predictors respectively. We first establish the sample complexity of $\ell_2$-swap multicalibration by using a characterization of swap multicalibration in terms of swap agnostic learning. Subsequently, we utilize the sample complexity result derived from the previous section.

**Lemma 15.** *(Globus-Harris et al., 2023, Theorem 3.2) Fix a predictor $p$ and assume that there exists a $v \in \mathsf{Range}(p)$, $f \in \mathcal{F}_1$ such that $\mathbb{E}[f(x)(y-v)|p(x) = v] \geq \alpha$, for some $\alpha > 0$. Then, there exists $f' \in \mathcal{F}_4$ such that*

$$\mathbb{E}\left[(p(x) - y)^2 - (f'(x) - y)^2|p(x) = v\right] \geq \alpha^2.$$

The above result was concurrently obtained by Globus-Harris et al. (2023) and Gopalan et al. (2023b), and subsequently extended to the online setting by Garg et al. (2024) (see also Lemma 3). Using Lemma 15, we establish the following result that relates the $\ell_2$-swap multicalibration error and the swap agnostic error.

**Lemma 16.** *Fix a distribution $\pi$ over predictors, and assume that there exists $\alpha > 0$ such that*

$$\sup_{\{f_v \in \mathcal{F}_4\}_{v \in \mathcal{Z}}} \mathbb{E}_{(x,y)\sim\mathcal{D}, p\sim\pi}\left[(p(x) - y)^2 - (f_{p(x)}(x) - y)^2\right] \leq \alpha.$$

*Then,*

$$\sup_{\{f_v \in \mathcal{F}_1\}_{v \in \mathcal{Z}}} \mathbb{E}_{p\sim\pi}\mathbb{E}_v\left[\mathbb{E}\left[f_v(x) \cdot (y - v)|p(x) = v\right]^2\right] \leq \alpha.$$

*Proof.* The proof follows a similar approach to that of Lemma 4. Assume on the contrary that

$$\sup_{\{f_v \in \mathcal{F}_1\}_{v \in \mathcal{Z}}} \mathbb{E}_{p\sim\pi}\mathbb{E}_v\left[\mathbb{E}\left[f_v(x) \cdot (y - v)|p(x) = v\right]^2\right] > \alpha.$$

Then, there exists a comparator profile $\{f_v \in \mathcal{F}_1\}_{v \in \mathcal{Z}}$ such that

$$\mathbb{E}_{p\sim\pi}\left[\sum_{v\in\mathcal{Z}} \mathbb{P}(p(x) = v) \cdot \mathbb{E}[f_v(x)(y - v)|p(x) = v]^2\right] > \alpha.$$

For simplicity, define $\alpha_v := \mathbb{P}(p(x) = v) \cdot \mathbb{E}[f_v(x)(y - v)|p(x) = v]^2$. Then, for all $v \in \mathcal{Z}$, we have

$$|\mathbb{E}[f_v(x)(y - v)|p(x) = v]| = \sqrt{\frac{\alpha_v}{\mathbb{P}(p(x) = v)}}.$$

Clearly, there exists a function $f_v^\star$ which is either $f_v$ or $-f_v$ such that $\mathbb{E}[f_v^\star(x)(y - v)|p(x) = v] = \sqrt{\frac{\alpha_v}{\mathbb{P}(p(x)=v)}}$. Furthermore, $f_v^\star \in \mathcal{F}_1$ since $\mathcal{F}$ satisfies Assumption 1. As per Lemma 15, for each $v \in \mathcal{Z}$ there exists a $f_v' \in \mathcal{F}_4$ such that

$$\mathbb{P}(p(x) = v) \cdot \mathbb{E}[(p(x) - y)^2 - (f_v'(x) - y)^2|p(x) = v] \geq \alpha_v$$

for all $v \in \mathcal{Z}$. Summing over $v \in \mathcal{Z}$, we obtain $\mathbb{E}_v\mathbb{E}\left[(p(x) - y)^2 - (f_v'(x) - y)^2|p(x) = v\right] \geq \sum_{v\in\mathcal{Z}} \alpha_v$. Taking expectation over $p \sim \pi$, we obtain

$$\mathbb{E}_{p\sim\pi}\left[\mathbb{E}_v\mathbb{E}\left[(p(x) - y)^2 - (f_v'(x) - y)^2|p(x) = v\right]\right] \geq \mathbb{E}_{p\sim\pi}\left[\sum_{v\in\mathcal{Z}} \alpha_v\right] > \alpha$$

which contradicts the assumption

$$\sup_{\{f_v \in \mathcal{F}_4\}_{v \in \mathcal{Z}}} \mathbb{E}_{(x,y)\sim\mathcal{D}, p\sim\pi}\left[(p(x) - y)^2 - (f_{p(x)}(x) - y)^2\right] \leq \alpha.$$

This completes the proof. $\qquad\square$

Since $\gtrsim \varepsilon^{-2.5}$ samples are sufficient to achieve a swap agnostic error at most $\varepsilon$ (as per Theorem 5), we obtain a $\tilde{\mathcal{O}}(\varepsilon^{-2.5})$ sample complexity of learning a $\ell_2$-swap multicalibrated predictor with error at most $\varepsilon$. Therefore, we have the main result of this section.

**Theorem 6.** *With probability at least $1 - \delta$, the randomized predictor $p$ (Section D.2) learned via the online-to-batch reduction satisfies*

$$\mathsf{DSMCal}_{\mathcal{F}_1^{\mathsf{lin}}, 2} = \tilde{\mathcal{O}}\left(\left(\frac{d}{T}\right)^{\frac{2}{5}} + \left(\frac{d}{T}\right)^{\frac{2}{5}}\sqrt{\log\frac{1}{\delta}}\right),$$

$$\mathsf{DSMCal}_{\mathcal{F}_1^{\mathsf{lin}}, 1} = \tilde{\mathcal{O}}\left(\left(\frac{d}{T}\right)^{\frac{1}{5}} + \left(\frac{d}{T}\right)^{\frac{1}{5}}\left(\log\frac{1}{\delta}\right)^{\frac{1}{4}}\right).$$

*Consequently, $\tilde{\Omega}(d\varepsilon^{-2.5}), \tilde{\Omega}(d\varepsilon^{-5})$ samples are sufficient to achieve $\ell_2, \ell_1$-swap multicalibration errors at most $\varepsilon$, respectively.*

