# OpenReview forum: "Improved Bounds for Swap Multicalibration and Swap Omniprediction"
_NeurIPS.cc/2025/Conference — NeurIPS 2025 spotlight_

### Official Review · Reviewer_v3FW · 2025-06-24

**Clarity:** 2
**Significance:** 3
**Originality:** 2
**Rating:** 4
**Confidence:** 1

**Summary:**

This paper studies multicalibration and omniprediction problems in distributional and online settings. In the online setting, they show an $O(T^{1/3})$ $\ell_2$-swap multicalibration error, resolving an open question posed by Garg et al. The algorithm, which is based on Blum-Mansour reduction, is extended to significantly improve error and sample complexity bounds on several related notions of contextual swap regrets, multicalibration and omniprediction errors.

**Questions:**

Please see the questions in the "Strengths and Weaknesses" section above.

**Ethical Concerns:**

["NO or VERY MINOR ethics concerns only"]

**Final Justification:**

I keep my original positive score (4) since the authors have adequately addressed my questions.

**Limitations:**

Yes

**Quality:**

2

**Strengths And Weaknesses:**

Disclaimer: I am not familiar with the topic of the paper.

## Assessment:
This paper makes significant contribution in improving a large number of existing error and sample complexity bounds. In most cases, the improvements are orders of magnitude better than existing results (e.g. Theorem 1, Corollary 1 and Theorem 4).

The two new notions on pseudo-swap multicalibration and pseudo contextual swap regret, as well as the reduction from the former to the latter, seem fundamentally important as they ultimately lead to improved $\ell_2$-swap multicalibration error.

In addition to novel ideas and improved bounds, it appears that the paper also introduce new analysis techniques as stated in line 134: "By
performing a careful martingale analysis using Freedman’s inequality on a geometric partition of the interval [0, 1], we finally establish the desired concentration bound." However, this new martingale analysis is not made clear in the main text.

In terms of weaknesses, I do not find anything significant. For a theoretical paper with many upper bounds, it is beneficial to have a formal argument on the sort of lower bounds one can obtain. However, it is understandable that lower bound proofs might take an entirely different paper.

## Questions:
1. Can the authors clarify where the "careful martingale analysis using Freedman’s inequality on a geometric partition of the interval [0, 1]" is? I did look at Appendix D.1, but I cannot find any new technical lemma on this. Is this hidden in the proof of Lemma 9? In any cases, could you briefly describe the nature of this new analysis?
2. I cannot find the discussion on the computational complexity of your algorithms. In particular, I am concerned whether or not your algorithms have to explicitly construct a cover in Proposition 1.
3. How robust are your results with respect to the particular finite discretization of $[0,1]$ line 76? (This might be a stupid question as I am not familiar with this setting).
4. While I suppose the paper is sufficiently clear for expert reviewers who are familiar with swap regret and multicalibration, some parts of it should be improved for better clarity so that even people outside of the domain can understand. For example, I am unable to grasp what $\mathcal{P}_t$ in line 154 is. Where does it come from, for example is it something that the algorithm computes in time step $t$?
5. Can you clarify what it means for $\mathcal{P}_t$ to be deterministic?
6. Similarly, Algorithm 1 is not self-contained as it is written. Are $p_{t,i}$ and $ALG_i$ input to Algorithm 1?

---

> ### Author Rebuttal · Authors · 2025-07-26
>
> We sincerely thank the reviewer for their comments and time in evaluating our paper. Below, we respond to all your questions and comments.
>
> > For a theoretical paper with many upper bounds, it is beneficial to have a formal argument on the sort of lower bounds one can obtain. However, it is understandable that lower bound proofs might take an entirely different paper.
>
> Regarding lower bounds, we would like to mention that Appendix A does provide some lower bounds for online $\ell_{1}, \ell_{2}$-Calibration (particularly, $\text{Cal}_1, \text{Cal}_2$ as defined in lines 167-168). In summary, proving lower bounds for online calibration is a significantly challenging problem and obtaining the minimax bound is still open, with recent breakthroughs establishing that $\tilde{\Omega}(T^{0.54389}) \le \mathbb{E}[\text{Cal}_1] \le \tilde{\mathcal{O}}(T^{\frac{2}{3} - \varepsilon})$ (for a sufficiently small $\varepsilon > 0$) and $\tilde{\Omega}(T^{0.08778}) \le \mathbb{E}[\text{Cal}_2] \le \tilde{\mathcal{O}}(T^{\frac{1}{3}})$ (Dagan et al., Fishelson et al.). Based on the above results, we infer the following:
>
> (a) Since contextual learning is harder than the non-contextual variant, and for the squared loss, the non-contextual swap regret is equivalent to $\text{Cal}_2$, the lower bound for $\text{Cal}_2$ also applies to $\text{SReg}$ (contextual swap regret).
>
> (b) Since (swap) multicalibration is stronger than calibration, the same lower bounds also apply to $\text{MCal}, \text{SMCal}$.
>
> (c) For swap omniprediction, Garg et al. proved that for a loss class that comprises of convex Lipschitz functions, against a hypothesis class $\mathcal{F}$ that consists of the constant $0, 1$ functions, we have $\Omega(\mathbb{E}[\text{Cal}_1]) \le \mathbb{E}[\text{SOmni}]$.
>
> > Can the authors clarify where the "careful martingale analysis using Freedman’s inequality on a geometric partition of the interval [0, 1]" is? I did look at Appendix D.1, but I cannot find any new technical lemma on this. Is this hidden in the proof of Lemma 9? In any case, could you briefly describe the nature of this new analysis?
>
> Yes, the analysis is hidden in the proof of Lemma 9. Recall that Freedman's inequality (Lemma 6; Line 604) states that for a martingale difference sequence $X_1, \dots, X_T$, where $|X_t| \le B$ for all $t \in [T]$, we have $|\sum_{t = 1} ^ {T} X_t| \le \mu V_X + \frac{1}{\mu} \log \frac{2}{\delta}$ with probability at least $1 - \delta$, where $V_X = \sum_{t = 1} ^ {T} \mathbb{E}_t[X_t ^ {2}]$, $\mu \in [0, \frac{1}{B}]$ is a fixed constant, and $\delta \in (0, 1)$. When $V_X$ is a deterministic quantity, the upper bound in the display above can be minimized with respect to $\mu \in [0, \frac{1}{B}]$; the optimal $\mu^\star = \min\left(\frac{1}{B}, \sqrt{\frac{\log \frac{2}{\delta}}{V_X}}\right)$. However, when $V_X$ is random (which is indeed the case in Lemma 9), we cannot merely substitute $\mu^\star$ in the bound above since $\mu^\star$ is random. To circumvent this difficulty, we partition the feasible interval for $\mu$ geometrically (e.g., when $B = 2$, we partition to intervals of the form $[\frac{1}{2^{k + 1}}, \frac{1}{2^{k}}]$), apply Freedman's inequality within each interval, and subsequently take a union bound. Notably, each interval corresponds to a condition on $V_X$ for which the optimal $\mu$ falls within that interval, and for that interval substituting $\mu = \frac{1}{2^{k}}$ or $\mu = \frac{1}{2^{k + 1}}$ leads to only a lower-order blow-up in the quantity $\mu V_X + \frac{1}{\mu} \log \frac{2}{\delta}$ compared to substituting $\mu^\star$.
>
> > I cannot find the discussion on the computational complexity of your algorithms. In particular, I am concerned whether or not your algorithms have to explicitly construct a cover in Proposition 1.
>
> No, our algorithm does not construct a cover $\mathcal{C}_\varepsilon$. We only utilize Proposition 1 in the analysis (particularly Lemma 2) in two ways: (a) the $\mathcal{O}(Nd\log \frac{1}{\varepsilon})$ term is due to the bound on the size of the cover; (b) we upper bound $\text{PSMCal}$ for the cover by $\text{PSMCal}$ for the linear class. As can be observed from (b) above, instead of minimizing the pseudo swap multicalibration error over the cover, we instead minimize it over the linear class (since the cover is chosen as a subset of the linear class (ref. Proposition 1)). Regarding the overall computation cost, our algorithm in Section 2.3 operates over the ball $4 \cdot \mathbb{B}_2^d$ and all our subroutines (BM reduction, ONS, Randomized rounding) can be implemented efficiently over the ball.
>
> > How robust are your results with respect to the particular finite discretization of $[0,1]$ line 76?
>
> The question of robustness with regard to the discretization scheme is generally not considered in online calibration since almost always the uniform discretization scheme (which arguably is the simplest discretization) yields the desired result. However, depending on the curvature of the loss function, or the problem itself a non-uniform discretization scheme might be required, e.g., a non-uniform discretization ($z_i = \sin ^ {2} \frac{\pi i}{2 N}$ for all $i = 1, \dots, N - 1$) scheme was used in Luo et al (2025) to minimize the swap regret for the log loss.
>
> Luo, H., Senapati, S., \& Sharan, V. (2025). Simultaneous swap regret minimization via kl-calibration. arXiv preprint arXiv:2502.16387.
>
> > While I suppose the paper is sufficiently clear for expert reviewers who are familiar with swap regret and multicalibration, some parts of it should be improved for better clarity so that even people outside of the domain can understand. For example, I am unable to grasp what $\mathcal{P}_t$ in line 154 is. Where does it come from, for example, is it something that the algorithm computes in time step $t$?
>
> We apologize for the lack of clarity for the more general audience, and this is something that we shall work on for the subsequent revisions. $\mathcal{P}_t$ represents a probability distribution over the set $\mathcal{Z} = \{0, \frac{1}{N}, \dots, \frac{N - 1}{N}, 1\}$. The elements of $\mathcal{Z}$ represent predictions $(p_t)$ that our algorithm makes. At each time $t \in [T]$, our algorithm computes $\mathcal{P}_t$ and predicts $p_t$ by sampling according to $\mathcal{P}_t$, i.e., $p_t \sim \mathcal{P}_t$.
>
> > Can you clarify what it means for $\mathcal{P}_t$ to be deterministic?
>
> Recall that $\mathcal{P}_t$ represents a probability distribution over the set $\mathcal{Z} = $ {$0, \frac{1}{N}, \dots, \frac{N - 1}{N}, 1$}, i.e., $\mathcal{P}_t \in \Delta(\mathcal{Z})$, where we have used the notation $\Delta(\Omega)$ to represent the simplex over $\Omega$. $\mathcal{P}_t$ being deterministic means that the algorithm does not sample $\mathcal{P}_t \sim \mathcal{D}$, where $\mathcal{D} \in \Delta(\Delta(\mathcal{Z}))$. Note that this is not necessarily true for an adaptive adversary, since the adversary can include its own randomness and make $\mathcal{P}_t$ random, despite of the algorithm not randomizing over $\mathcal{P}_t$; however, since we consider an oblivious adversary throughout the paper, we can abide by the above interpretation.
>
> > Similarly, Algorithm 1 is not self-contained as it is written. Are $p_{t,i}$ and $ALG_i$ input to Algorithm 1?
>
> Algorithm 1 invokes $ALG_i$ as a subroutine, and $p_{t,i}$ is an input to $ALG_i$ (therefore, Algorithm 1 as well). We apologize if things were not clear and shall work on revising in the subsequent versions.
>
> Should you have any more questions, do not hesitate to reach out. If our responses have addressed your concerns, we would be grateful if you would consider updating your assessment accordingly.

---

### Official Review · Reviewer_59sM · 2025-07-02

**Clarity:** 4
**Significance:** 3
**Originality:** 3
**Rating:** 5
**Confidence:** 4

**Summary:**

This paper provides an algorithm and improved upper bounds for the problems of (swap) omniprediction and multicalibration in the online and (by an online-to-batch reduction) the offline batch settings. In particular, the authors employ an alternate proof strategy to solve an open problem proposed by Garg et al. on the possibility of achieving $O(\sqrt{T})$ $\ell_2$-multicalibration error against bounded linear functions. The authors' main technical result is an algorithm for achieving $\ell_2$-swap multicalibration in $O(T^{1/3})$. This bound also implies significant rates for multicalibration, swap omniprediction for convex Lipschitz functions, and, accordingly, improved sample complexity in the offline analogues. The main argument, to my understanding, is as follows:

1. The authors introduce two notions *pseudo swap multicalibration* and *pseudo swap contextual regret* that avoid dealing with averaging over the Learner's predictions in the online game, and a reduction exists from swap multicalibration to pseudo swap multicalibration (simply via a covering argument on the class of linear functions)
2. The authors reduce pseudo swap multicalibration to pseudo contextual swap regret via properties of the linear function class
3. **main technical contribution:** the authors develop an algorithm for pseudo contextual swap regret using a reduction to the classical external to swap regret reduction of Blum and Mansour and an improved rounding scheme. This reduction requires an external regret algorithm to exist, which the authors instantiate as simply Online Newton Step for a particular linear function class.

The upper bound obtained by this argument then implies improved results in online and offline (swap) omniprediction and $\ell_1$-multicalibration through improvements of more standard arguments.

**Questions:**

1. Have these pseudo notions previously appeared in the literature? It seems that the argument hinges around the construction of these notions, and they do point towards a major difficulty in multicalibration-style regret bounds -- the conditioning on the $p_t$s.

2. Is the bounded linear functional necessary in this argument because of the covering and reduction to $P_t$ (pseudo notions)? The argument seems specialized once we go to pseudo notions because of the ease of constructing a cover. To this note, would more general function classes need new techniques entirely?

3. I'm curious as to the efficiency compared to the previous algorithm of Garg et al? To my understanding, the Blum/Mansour reduction is efficient because it just employs $N$ external regret algorithms (where $N$ is from the rounding). Even a qualitative comparison of the "efficiency" compared to the previous algorithm would be interesting, just to note the algorithmic differences of the approaches.

**Ethical Concerns:**

["NO or VERY MINOR ethics concerns only"]

**Final Justification:**

I keep my original positive score.

**Limitations:**

Yes (pure theory paper).

**Paper Formatting Concerns:**

It seems that the authors attached the optional Appendix to the nine content pages by mistake (but the checklist is still there).

**Quality:**

3

**Strengths And Weaknesses:**

I am quite positive on this paper -- the results are clearly presented with an eye towards giving a nice proof sketch and overview of the entire argument for the reader, the techniques themselves are combined in a novel way, and the authors provide good contextualization with the rest of the literature on omniprediction and multicalibration. The only weakness that the authors might address is to possibly provide a bit more intuition on why these particular techniques result in the improved bounds -- although it seems clear to me that the improvements come from the reduction to the *pseudo* notions, a comparison to why the pseudo notions combined with the standard Blum/Mansour external to swap regret reduction and improved rounding scheme give a key algorithmic improvement to the prior work of Garg et al. I understand that the rates go down, but it would be very nice to also have the insight on *why* the pseudo notion is "easier to optimize" in this particular context. A couple sentences on that would improve an already very good paper.

## Strengths

**Originality:** To my knowledge, the authors' techniques in proving the main improved upper bound on $\ell_2$-swap multicalibration is novel, and, in particular, the key introduction of *pseudo* swap multicalibration and *pseudo* contextual swap regret is novel. In fact, I am interested in seeing if similar notions have cropped up in the literature (see "Questions"), as the move from having to condition using the empirical conditional distribution of the $p_t$ to just dealing with the true distributions at each step certainly simplify the "conditioning" problem that arises when dealing with these multicalibration notions in the online setting, and I'd be interested in seeing if this kind of strategy has been employed before. To my knowledge, this seems like a new move.

**Quality:** The results of the paper themselves improve bounds for several open questions, and the authors "use the full force" of their main derived bound to fully address the implications downstream for ominprediction and multicalibration by also working out the details for the offline setting, which I appreciated. It answers an existing open question about a specific faster rate for $\tilde{O}(\sqrt{T})$ for the specific problem of swap multicalibration in the affirmative, and the new algorithm that results in this upper bound uses the authors' pseudo notions effectively.

**Clarity:** The paper itself is very well-written and the results are presented in a clear manner, with intuitive proof sketches and a logical flow to the sections. The provided diagram is helpful in understanding the big picture of the argument, and the authors do not waste space on more routine details. I had little trouble following along with the paper (though I am quite familiar with the multicalibration/omniprediction subfield, so others' mileage may vary).

**Significance:** I believe that the results the authors provide are significant enough to warrant publication and answer an existing open question. The reduction to the pseudo notions also give us insight on a particular difficulty in multicalibration/related problems in regret minimization, particularly their swap variants.

## Weaknesses
The weaknesses I see in the work are minor.

The first pertains to motivation: the authors might address is to possibly provide a bit more intuition on why these particular techniques result in the improved bounds -- although it seems clear to me that the improvements come from the reduction to the *pseudo* notions, a comparison to why the pseudo notions combined with their algorithm nets an improvement.

Another weakness one might argue that the results themselves are too narrow -- that the result for just the class of bounded linear functions doesn't achieve the scope of previous multicalibration results. I don't really see this as a big weakness, as the existing open question asked if a faster rate was possible even for bounded linear functions, and that was not known. Although it seems to me that more general function classes may need different techniques entirely (it is unclear to me how to extend dealing with the "pseudo" notions for more complex function classes), the technical contribution of the authors for bounded linear functions does add knowledge to the subfield, in my opinion.

---

> ### Author Rebuttal · Authors · 2025-07-27
>
> We sincerely thank the reviewer for their comments and time in evaluating our paper. Below, we provide responses to all your questions and comments.
>
> > Have these pseudo notions previously appeared in the literature? It seems that the argument hinges around the construction of these notions, and they do point towards a major difficulty in multicalibration-style regret bounds -- the conditioning on the $p_{t}$'s.
>
> To the best of our knowledge, the work by Fishelson et al. [8] was the first to introduce the notion of pseudo swap regret (referred to in their paper as full swap regret) and pseudo calibration. Their motivation was to minimize the pseudo swap regret for a broad class of loss functions (e.g., strongly convex, Lipschitz) in the high-dimensional setting. Following up on their work, Luo et al. [7] considered online calibration of binary outcomes and proposed a new calibration measure called KL-Calibration (along with an accompanying notion pseudo KL-Calibration), which they showed is equivalent to the swap regret (resp. pseudo swap regret) of the log loss. As mentioned in the manuscript, our introduction of pseudo contextual swap regret and pseudo swap multicalibration is motivated from the above works.
>
> Moreover, pseudo notions have previously appeared in the literature in several other contexts. For example, a recent line of work on UCalibration (Kleinberg et al. [24], Luo et al. [25]) also considered (pseudo) U-Calibration, where the U-Calibration error ($\text{UCal}$) for a family $\mathcal{L}$ of loss functions is defined as $\text{UCal} \coloneqq \mathbb{E}\left[\sup_{\ell \in \mathcal{L}} \text{Reg}^\ell\right]$, where $\text{Reg}^\ell$ represents the external regret incurred by the forecaster for a loss function $\ell \in \mathcal{L}$, where as the pseudo U-Calibration error is defined as $\text{PUCal} \coloneqq \sup_{\ell \in \mathcal{L}} \mathbb{E}\left[\text{Reg} ^ \ell\right]$. A similar notion of pseudo regret has also previously appeared in the literature on multi-armed bandits (ref. the book by Bubeck and Cesa-Bianchi).
>
> Bubeck, S., & Cesa-Bianchi, N. (2012). Regret analysis of stochastic and nonstochastic multi-armed bandit problems. Foundations and Trends® in Machine Learning, 5(1), 1-122.
>
> > Is the bounded linear functional necessary in this argument because of the covering and reduction to
>  (pseudo notions)? The argument seems specialized once we go to pseudo notions because of the ease of constructing a cover. To this note, would more general function classes need new techniques entirely?
>
> As you pointed out, two crucial steps that invoke specific properties of the linear class are: (a) the existence of a cover , which allows us to bound $\text{SMCal}$ for the linear class in terms of $\text{SMCal}$ for the cover (in Lemma 2, we relate the latter to the pseudo swap multicalibration error $\text{PSMCal}$ for the cover via the concentration bound in Lemma 1; the $\mathcal{O}(N)$ term in Lemma 1 is needed to obtain $\tilde{\mathcal{O}}(T^{\frac{1}{3}})$ style bounds); (b) an explicit algorithm (ONS) that minimizes the scaled external regret $$\mathbb{E}\left[\sup_{f \in \mathcal{F}} \sum_{t = 1} ^ {T} \alpha_{t} (w_{t} - y_{t}) ^ {2} - \alpha_{t}(f(x_{t}) - y_{t}) ^ {2}\right]$$ against the linear class. For an arbitrary $\mathcal{F}$ which is closed under affine transformations (we require the closure property for the reduction from (pseudo) swap multicalibration to (pseudo) contextual swap regret (Lemma 3)), we suspect that Rademacher complexity arguments for obtaining fast rates for online learning of the squared loss can be generalized for minimizing the scaled external regret in the display above. However, for an arbitrary $\mathcal{F}$ which does not admit a finite cover, we are not aware of any techniques to generalize and prove an equivalent of Lemma 2. While it is certainly possible to derive an equivalent of Lemma 1 for an arbitrary $\mathcal{F}$ using results from the theory of empirical processes (in particular, following the chaining technique in Block et al., we bound the deviation of a martingale difference sequence $Z_{1}, \dots, Z_{T}$), we do not hope to retrieve the $\mathcal{O}(N)$ term appearing in Lemma 1 since the proof of Lemma 1 heavily relies on controlling the variance of the associated martingale difference sequence via Freedman's inequality. We are not familiar with whether similar results that bound the expected deviation of the supremum of a martingale difference sequence with the variance exist.
>
> In summary, following our framework, generalizing the $\tilde{\mathcal{O}}(T^{\frac{1}{3}})$ bound for swap multicalibration for the linear class to an arbitrary $\mathcal{F}$ seems quite convoluted and requires significant heavy lifting.
>
> Block, A., Dagan, Y., \& Rakhlin, A. (2021, July). Majorizing measures, sequential complexities, and online learning. In Conference on Learning Theory (pp. 587-590). PMLR.
>
> > I'm curious as to the efficiency compared to the previous algorithm of Garg et al? To my understanding, the Blum/Mansour reduction is efficient because it just employs external regret algorithms (where $N$ is from the rounding). Even a qualitative comparison of the "efficiency" compared to the previous algorithm would be interesting, just to note the algorithmic differences of the approaches.}
>
> For minimizing the contextual swap regret of the squared loss against the linear class, the algorithm of Garg et al. invokes the reduction from swap regret to external regret as proposed by Ito [10] and Azoury and Warmuth's algorithm for minimizing external regret against the linear class. In contrast, for minimizing pseudo contextual swap regret, we utilize the Blum-Mansour reduction and the ONS algorithm for minimizing the scaled external regret against the linear class. Qualitatively, both the Blum-Mansour reduction and the reduction by Ito require fixed-point computations, and roughly speaking, the AW algorithm requires similar computation as ONS, thereby rendering our computation cost similar to that of Garg et al. Notably, both our algorithm and that of Garg et al. are efficient due to the efficiency of the subroutines that they invoke.
>
>  However, Garg et al. achieve $\tilde{\mathcal{O}}(T^{\frac{3}{4}})$ $\ell_{2}$-swap multicalibration error, whereas we achieve $\tilde{\mathcal{O}}(T^{\frac{1}{3}})$ $\ell_{2}$-swap multicalibration error (we also achieve a superior dependence on the dimensionality $d$), which is orders of magnitude better. In a restricted setting  ($|\mathcal{F}|$ is finite), Garg et al. also proposed an algorithm that achieves $\tilde{\mathcal{O}}(\sqrt{T})$ $\ell_{2}$-multicalibration error. However, the above algorithm is inefficient since it has running time proportional to $|\mathcal{F}|$.
>
>  Azoury, K. S., \& Warmuth, M. K. (2001). Relative loss bounds for on-line density estimation with the exponential family of distributions. Machine learning, 43(3), 211-246.
>
> We hope that our responses answer your questions. Should you have any more questions, please feel free to reach out.

---

> > ### Comment · Reviewer_59sM · 2025-08-06
> >
> > Thank you to the authors for providing these comprehensive answers to my questions! I have decided to keep my score.

---

### Official Review · Reviewer_bikt · 2025-07-16

**Clarity:** 4
**Significance:** 3
**Originality:** 3
**Rating:** 5
**Confidence:** 3

**Summary:**

This paper derives improved online rates and sample complexity rates for swap omniprediction.
  Swap omniprediction is a strengthening of omniprediction (learn a predictor that is close enough to Bayes optimal that it can be post-processed to be near-optimal for any loss) that was introduced by GKR24, who motivated the concept by noting that while multicalibration > omniprediction, swap multicalibration = swap omniprediction.
  The authors obtain these improved rates by instead first deriving faster rates for a 'pseudo' variant of swap multicalibration that allows them to reduce to a more tractable problem of pseudo contextual swap regret.

 The main proofs are for the online setting, with the distributional sample complexity rates following from an online-to-batch reduction. The main intuition of the paper (which is nicely summarized in Figure 1) is that instead of obtaining swap multicalibration rates by reducing to swap regret minimization, the authors instead reduce to pseudo swap regret minimization. One way to understand what this detour through pseudo calibration measures is doing is that its effectively separating the analysis of the variance caused by the forecaster's random forecasts with the actual learning component.

  The interesting thing (at least to me, who has not followed the recent work here closely) is that reducing from swap multicalibration to pseudo-swap multicalibration costs such little overhead: $O(N)$ . This is done in Lemma 1 which shows SMCal_{ F, 2} leq O(N log (|F| N / delta)) + 6 PSMCal_{ F,2} with the overhead term being negligible for the types of discreitizations we normally consider, e.g. N=T^1/3. The analysis is an adaptation of Freedman's inequality, which is currently not really talked about in the main text.

  The pseudo-multicalibration rates are obtained by reducing to pseudo contextual swap regret. This happens in Lemma 3 (through lemma 4). Specifically: if there is pseudo-multicalibration error, then that error can be captured by a linear competitor meaning that there must be some pseudo contextual swap regret. So, pseudo swap contextual regret for the affine closure of functions in F always upper bounds pseudo multicalibration for F.  The other step of the reduction, going from swap multicalibration to swap omniprediction is standard.

The other chunk of the paper is actually obtaining good rates for pseudo swap regret minimization. The algorithm follows the usual Blum-Mansour algorithm for minimizing swap regret where you instantiate multiple algorithms, one for each action, and then act according to a stationary distribution over the algorithms. In this case, we use the BM algorithm with actions corresponding to discretized level sets and online newton step for external regret minimizers (because squared loss is exp-concave). The algorithm design and analysis follow a pretty standard recipe, with the main interesting component being the observation that relaxing to "pseudo" swap regret means that instead of the BM reduction getting noisy feedback it gets deterministic feedback, and so one doesn't need to add in an error term for stochastic approximation.

**Questions:**

I'm not very familiar with the swap omniprediction literature; is there a fundamental eps^-3-eps^-2 separation for swap vs normal omniprediction that's known?

Prior offline swap omniprediction rates cited in Table 1 are attributed to [2, 13] rather than [5]: is there a challenge with online-to-batch adaptation of the rates of [1]?

Why do these detours into pseudo calibration measures arise with "swap" objectives whereas e.g. [6]'s optimal rates for omniprediction do not appear to need them?

**Ethical Concerns:**

["NO or VERY MINOR ethics concerns only"]

**Final Justification:**

I maintain my original positive review.

**Limitations:**

Yes

**Quality:**

4

**Strengths And Weaknesses:**

The paper is well-written with concrete improvements in theoretical results (improved rates for swap omniprediction) and a clean conceptual contribution: when trying to get rates for swap calibration/omniprediction/agnostic-learning, it's often easier to mentally separate out how one analyzes the stochastic approximation errors from forecast randomness—reducing to pseudo calibration and pseudo swap regret is a nice way to think about how to do this.

I don't have any particular concerns with weaknesses in presentation or results. I did not check the proofs, which are deferred entirely to the appendix, carefully though the results and steps seem reasonable.

---

> ### Author Rebuttal · Authors · 2025-07-26
>
> We sincerely thank the reviewer for their comments and time in evaluating our paper. Below, we provide responses to all your questions.
>
> > I'm not very familiar with the swap omniprediction literature; is there a fundamental $\varepsilon^{-3}-\varepsilon^{-2}$ separation for swap vs normal omniprediction that's known?
>
>  We are not aware of such a separation in the offline setting. However, in the online setting, Garg et al. [1] established that for a loss class of convex Lipschitz functions, against a hypothesis class $\mathcal{F}$ that comprises the constant $0, 1$ functions, we have $\tilde{\Omega}(T^{0.54389}) \le \mathbb{E}[\text{SOmni}]$, where the $\tilde{\Omega}(T^{0.54389})$ lower bound is due to a breakthrough result by Dagan et al. [19], Qiao and Valiant [18]. The lower bound on swap omniprediction also implies that an online-to-batch conversion for swap omniprediction cannot achieve $\varepsilon ^ {-2}$ sample complexity, in general. For omniprediction, Garg et al. [1], Okoroafor et al. [6] established that $\tilde{\mathcal{O}}(\sqrt{T})$ omniprediction error is achievable, therefore establishing a strict separation between the above notions in the online setting.
>
> > Prior offline swap omniprediction rates cited in Table 1 are attributed to [2, 13] rather than [5]: is there a challenge with online-to-batch adaptation of the rates of [1]?
>
> [5] established an equivalence between swap multicalibration and swap omniprediction, and also showed that the multicalibration algorithm of [2] is swap multicalibrated; therefore, we attributed the bound to [2], which had an explicit sample complexity bound. Moreover, a characterization of swap multicalibration in terms of swap agnostic learning was concurrently obtained by [13], who also derived an explicit sample complexity for swap agnostic learning; therefore, we chose to refer to [13]. However, we accept your suggestion and shall additionally refer to [5] for the result regarding swap omniprediction. For the second part of your question, a standard online-to-batch adaptation (using Azuma-Hoeffding's inequality) of the rates of [1] shall yield a $\tilde{\mathcal{O}}(\varepsilon^{-8})$ sample complexity for swap omniprediction for the class of convex Lipschitz functions against the class $\mathcal{F}_{\text{aff}}^{\text{res}}$ of scaled and shifted linear functions. Throughout the paper, we abstain from referring to this result since it was not mentioned in [1]; we instead tabulate the rates that were explicitly documented in the literature.
>
> > Why do these detours into pseudo calibration measures arise with "swap" objectives whereas e.g. [6]'s optimal rates for omniprediction do not appear to need them?
>
> Excellent question. The relationship between swap regret and calibration has been known since long, starting with the seminal work of Foster and Vohra [9]. It is known from several papers that $\ell_{2}$-Calibration ($\text{Cal}_{2}$) is equivalent to the swap regret of the squared loss. While this opens up a natural direction to developing algorithms for calibration, it also poses an immediate challenge --- minimizing swap regret is considerably challenging than minimizing external regret. Along the direction of minimizing swap regret, the well-known Blum-Mansour [11] reduction works for the non-stochastic component of learning, i.e., it bounds what we refer in the paper to as pseudo swap regret, which is related to pseudo calibration. While there exist other reductions, e.g., Ito et al., Dagan et al., that reduce swap regret minimization to external regret minimization, they either lead to suboptimal dependence on $T$ (Dagan et al.) or involve further randomization in the algorithm (Ito et al.), rendering their adaptation to minimizing the swap regret of the squared loss quite non-trivial. In summary, the relationship between calibration and swap regret, and the fact that minimizing pseudo swap regret (thus, pseudo calibration) is easier than swap regret, is a possible explanation of the detour.
>
> Ito, S. (2020). A tight lower bound and efficient reduction for swap regret. Advances in Neural Information Processing Systems, 33, 18550-18559.
>
> Dagan, Y., Daskalakis, C., Fishelson, M., \& Golowich, N. (2024, June). From external to swap regret 2.0: An efficient reduction for large action spaces. In Proceedings of the 56th Annual ACM Symposium on Theory of Computing (pp. 1216-1222).
>
> We hope that our responses answer your questions. Should you have any more questions, please feel free to reach out.

---

> > ### Author Response · Authors · 2025-08-07
> >
> > Dear Reviewer,
> >
> > Since the discussion period is coming to an end soon, we would like to know if you have any further questions, or if our answers have addressed your questions.
> >
> > Best, Authors

---

> > ### Comment · Reviewer_bikt · 2025-08-07
> >
> > Thanks for the response–I'll maintain my original positive review

---

### Official Review · Reviewer_wMot · 2025-07-21

**Clarity:** 1
**Significance:** 3
**Originality:** 2
**Rating:** 5
**Confidence:** 1

**Summary:**

This paper gives novel bounds on the multicalibration and ominprediction errors in the both the adversarial (online) and distributional settings. The proven error bounds in the adversarial setting are as follows:
* swap multi-calibration regret: $T^{1/3}$ (improved from $T^{3/4}$)
* swap omniprediction regret: $T^{2/3}$ (improved from $T^{7/8}$)
Similar improvements have also been established in the distributional setting, with an online-to-batch conversion. The key idea is to connect to a particular notion of a pseudo-regret (where some of the randomness is removed) via a martingale concentration argument.

**Questions:**

* What does the notation $\mathbb{R}^X$ mean? Is this the set of functions $X \to \mathbb{R}$?
* The connections between multi-calibration and fairness are unclear. It would be good to dedicate some space to that aspect to appeal to a broader audience.
* Are any lower bounds known for any of these problems? What about simplifications to known textbook-type problems?
* I'm unable to parse $\rho_{p, f}$ and the meaning of the definition in eq. (1). This needs some interpretation.m\

**Ethical Concerns:**

["NO or VERY MINOR ethics concerns only"]

**Final Justification:**

I'm updating my score with the understanding that the authors will revise the paper to make it more broadly accessible (with interpretations, examples, etc.) and less mathematically dense.

**Quality:**

3

**Strengths And Weaknesses:**

Strengths:
* This paper gives significantly improved bounds across the board over prior works on swap multi-calibration and, via established reductions, on swap omniprediction. Both of these are previously studied problems in the literature.
* While the techniques and individual steps appear to be known or inspired from very similar ones, the improved bounds are obtained by carefully putting together various pieces.
* The paper seems well-polished and I did not see any typos.

**Weaknesses**:

I found the paper nearly impossible to parse. As somebody who is familiar with learning theory but never previously heard about multi-calibration or omniprediction (or swap contextual regret, or similar terms), I expected to be able to at least understand the definitions and the significance of the results. While the paper reads like something that researchers knowledgeable in this (apparently niche) area would appreciate, I did not even understand the basic definitions. Further, it tries to pack a lot of math into a small number of pages, making the text incredibly dense as well.

I especially did not understand the definition of multi-calibration. While this is an established notion and has a prior literature, I believe the onus is on the authors to provide enough intuition about the definitions and significance of the results (at least in the appendix). Similarly, I do not understand where the "swap" versions come from or why they are interesting.

It would have been nice to see more special cases/connections as to why these bounds are interesting, such as connections to textbook examples that a reader such as myself would appreciate. For example, the introduction mentions connections to fairness: that would be a good setting to expand upon.

---

> ### Author Rebuttal · Authors · 2025-07-26
>
> We sincerely thank the reviewer for their comments and time in evaluating our paper. We are sorry to hear that the reviewer found our paper difficult to parse. We apologize for the same and shall revise the paper to make it appealing to the broader audience, keeping in mind several suggestions given by the reviewer. Below, we respond to all your questions and comments, along with an (additional) explanation of multicalibration, since the reviewer was unable to understand the concrete definition.
>
> > What does the notation $\mathbb{R}^{\mathcal{X}}$ mean? Is this the set of functions  $\mathcal{X} \to \mathbb{R}$?
>
> Yes, this is a standard notation used in set theory.
>
> > Are any lower bounds known for any of these problems? What about simplifications to known textbook-type problems?
>
> Regarding lower bounds, we would like to mention that Appendix A does provide some lower bounds for online $\ell_1, \ell_2$-Calibration (particularly, $\text{Cal}_1, \text{Cal}_2$ as defined in lines 167-168). In summary, proving lower bounds for online calibration is a significantly challenging problem and obtaining the minimax bound is still open, with recent breakthroughs establishing that $\tilde{\Omega}(T^{0.54389}) \le \mathbb{E}[\text{Cal}_1] \le \tilde{\mathcal{O}}(T^{\frac{2}{3} - \varepsilon})$ (for a sufficiently small $\varepsilon > 0$) and $\tilde{\Omega}(T^{0.08778}) \le \mathbb{E}[\text{Cal}_2] \le \tilde{\mathcal{O}}(T^{\frac{1}{3}})$ (Qiao and Valiant, Dagan et al., Fishelson et al.). Based on the above results, we infer the following:
>
> (a) Since contextual learning is harder than the non-contextual variant, and for the squared loss, the non-contextual swap regret is equivalent to $\text{Cal}_{2}$, the lower bound for $\text{Cal}_2$ also applies to $\text{SReg}$ (contextual swap regret).
>
> (b) Since (swap) multicalibration is stronger than calibration, the same lower bounds also apply to $\text{MCal}, \text{SMCal}$.
>
> (c) For swap omniprediction, Garg et al. proved that for a loss class that comprises of convex Lipschitz functions, against a hypothesis class $\mathcal{F}$ that consists of the constant $0, 1$ functions, $\Omega(\mathbb{E}[\text{Cal}_1]) \le \mathbb{E}[\text{SOmni}]$.
>
>
> Regarding simplifications to known textbook-type problems, we would like to mention that some reductions do exist; however, the reductions and the reduced problems are far from deserving a textbook treatment, e.g., a lower bound for $\text{Cal}_{1}$ was proved by Qiao and Valiant by reducing to the sign preservation game and an improved lower bound was proved by Dagan et al. by reducing to the sign preservation game with reuse.
>
> > I'm unable to parse and the meaning of the definition in eq.(1). This needs some interpretation.
>
>  $\rho_{p, f}$ represents the average correlation between $f$ and the residual error sequence $\{y_1 - p_1, \dots, y_{T} - p_{T}\}$ conditioned on the rounds that the forecaster predicts $p_{t} = p$. Noting the requirement for multicalibration in the distributional setting, i.e., $\mathbb{E}[f(x) \cdot (y - p(x)) | p(x) = v] = 0$ for all $f \in \mathcal{F}, v \in \text{Range}(p)$, the quantity $\rho_{p, f}$ is an analogue to $\mathbb{E}[f(x) \cdot (y - p(x)) | p(x) = v]$, adapted to the online setting. Similarly, the normalized term $\frac{1}{T}\sum_{t = 1} ^ {T} \mathbb{I}[p_{t} = p]$ in Equation (1) (normalization skipped in the manuscript) corresponds to the average frequency of the prediction $p$ and is an analogue to the quantity $\mathbb{P}(p(x) = v)$. With this intuition, the quantity $\sum_{t = 1} ^ {T} \mathbb{I}[p_{t} = p] \sup_{f \in \mathcal{F}} |\rho_{p, f}| ^ {q}$ corresponds to the maximum correlation (weighted by the number of rounds the prediction made is $p$) achieved by an $f \in \mathcal{F}$ for the prediction $p_{t} = p$. The cumulative sum across all predictions corresponds to the quantity $\text{SMCal}$ (we allow higher order $q$-th moments for $\rho_{p, f}$ to consider the full generality). Intuitively, a small value for $\text{SMCal}$ represents that no hypothesis in $\mathcal{F}$ is able to detect correlation with the residual sequence $y_{1} - p_{1}, \dots, y_{T} - p_{T}$ when conditioned on the level sets.
>
> > The connections between multi-calibration and fairness are unclear. It would be good to dedicate some space to that aspect to appeal to a broader audience.
>
>  Thank you for pointing this out. We shall clarify this in the subsequent revisions. Intuitively, since multicalibration is defined at the scale of partitions of the instance space (the predictions are conditionally unbiased for the every set $s \subseteq \mathcal{X}$ in a collection of subsets $\mathcal{S}$ of the input space $\mathcal{X}$), it allows fairness guarantees for every subset $s$ of $\mathcal{S}$ simultaneously, e.g., $\mathcal{X}$ could represent the population of a province, the set $\mathcal{S}$ can be several possibly overlapping demographic groups within that province, and $p$ can be the probability that an individual is affected by a particular disease. On the contrary, calibration is only a marginal requirement and cannot address such finer fairness concerns. For further discussion regarding the fairness implications of multicalibration, we refer to Obermeyer et al.
>
>  Obermeyer, Z., Powers, B., Vogeli, C., & Mullainathan, S. (2019). Dissecting racial bias in an algorithm used to manage the health of populations. Science, 366(6464), 447-453.
>
> > I especially did not understand the definition of multi-calibration.
>
> We begin with defining calibration in the distributional setting. Consider a binary classification problem with instance space $\mathcal{X}$, label space $\mathcal{Y}$ be {$0, 1$}, $\mathcal{D}$ be an unknown distribution over $\mathcal{X} \times \mathcal{Y}$, and $p^\star(x) \coloneqq \mathbb{P}(y = 1 | x)$ be the Bayes optimal predictor. Let $p: \mathcal{X} \to [0, 1]$ be a predictor and $\text{Range}(p)$ denote the range of $p$ (set of all values that $p$ predicts). The predictor $p$ is perfectly calibrated if for all $v \in \text{Range}(p)$, we have $\mathbb{E}[y | p(x) = v] = v$, i.e., the predictions of $p$ are correct conditioned on its level sets. With this background, multicalibration can be interpreted as a refinement of calibration to partitions of the instance space $\mathcal{X}$. Particularly, given a collection $\mathcal{S}$ of potentially overlapping subsets of $\mathcal{X}$, a predictor $p$ is perfectly multicalibrated if for all subsets $s \in \mathcal{S}$ and $v \in \text{Range}(p)$, we have $\mathbb{E}[y| p(x) = v, x \in s] = v$. As can be noted from the definition, the Bayes optimal predictor $p^\star$ is multicalibrated. For both computational and information-theoretic reasons, obtaining the Bayes optimal predictor is considered impossible; therefore, the problem of multicalibration asks to efficiently find a predictor $p$ such that $\mathbb{E}[y| p(x) = v, x \in s] \approx v$ for all $v \in \text{Range}(p), s \in \mathcal{S}$. Note that the above definition of multicalibration can be made more general. Specifically, since each subset $s$ of $\mathcal{X}$ can be described via a function $g: \mathcal{X} \to$ {$0, 1$} ($1$ if $x \in s$ and $0$ otherwise), the set $\mathcal{S}$ can be equivalently described by a collection $\mathcal{G}$ of set-membership functions. The above definition can be equivalently written as $\mathbb{E}[y| p(x) = v, g(x) = 1] = v \equiv \mathbb{E}[g(x) \cdot (y - p(x))|p(x) = v] = 0$. For an arbitrary real-valued hypothesis class $\mathcal{F} \subset \mathbb{R}^{\mathcal{X}}$, multicalibration requires $\mathbb{E}[f(x) \cdot (y - p(x)) | p(x) = v] = 0$ for all $f \in \mathcal{F}, v \in \text{Range}(p)$. Since the quantity $\mathbb{E}[f(x) \cdot (y - p(x))]$ corresponds to the correlation between $f$ and the residual $y - p(x)$, multicalibration for an arbitrary real-valued hypothesis class can be interpreted as the condition that the hypotheses in $\mathcal{F}$ cannot find any correlation with the residual sequence $y - p(x)$ when conditioned on the level sets of $p$.
>
> Should you have any more questions, do not hesitate to reach out. If our responses have addressed your concerns, we would be grateful if you would consider updating your assessment accordingly.

---

> > ### Comment · Reviewer_wMot · 2025-08-02
> > **Thank you for the clarifications**
> >
> > Thank you for the detailed response. I'll try to get another reading of in and get back in case of further questions.

---

> > > ### Author Response · Authors · 2025-08-07
> > >
> > > Dear Reviewer,
> > >
> > > Since the discussion period is coming to an end soon, and we haven’t heard back from you following up on your response, we would like to know if you have any further questions, or if our responses have answered your questions.
> > >
> > >
> > > Best,
> > > Authors

---

> > > > ### Comment · Reviewer_wMot · 2025-08-07
> > > > **Response**
> > > >
> > > > No further questions. I have revised my score. Thank you!

---

### Decision · Program_Chairs · 2025-09-17

**Decision:**

Accept (spotlight)

**Comment:**

This paper provides an algorithm and improved upper bounds for the problems of (swap) omniprediction and multicalibration in the online and (by an online-to-batch reduction) the offline batch settings. It has made several important theoretical contributions, including resolving an open problem in SODA and unifying calibration, swap regret, and omniprediction under a pseudo framework. Thus, I would recommend its acceptance with a spotlight.